# Implementation of a brittle sea-ice rheology in an Eulerian, finite-difference, C-grid modeling framework: Impact on the simulated deformation of sea-ice in the Arctic

Laurent Brodeau[1], Pierre Rampal[1], Einar Ólason[2], and Véronique Dansereau[3]

[1]IGE/CNRS, Grenoble, France.
[2]NERSC, Bergen, Norway.
[3]ISTERRE/CNRS, Grenoble, France.

**Correspondence:** Laurent Brodeau (laurent.brodeau@univ-grenoble-alpes.fr)

**Abstract.** We have implemented the Brittle Bingham-Maxwell sea-ice rheology (BBM) into SI3, the sea-ice component of NEMO. After discussing the numerical aspects and requirements that are specific to the implementation of a brittle rheology in the Eulerian, finite-difference, Arakawa C-grid framework, we detail the approach we have used. This approach relies on the introduction of an additional set of prognostic stress tensor components, sea-ice damage, and sea-ice velocity vector, following a grid-point arrangement that expands the C-grid into the Arakawa E-grid. The newly-implemented BBM rheology is first assessed by means of a set of idealized SI3 simulations at different spatial resolutions. Then, sea-ice deformation rates obtained from simulations of the Arctic at a 1/4°spatial resolution, performed with the coupled ocean/sea-ice setup of NEMO, are assessed against satellite observations. For all these simulations, results obtained with the default current workhorse setup of SI3 are provided to serve as a reference. Our results show that using a brittle-type of rheology, such as BBM, allows SI3 to simulate the highly-localized deformation pattern of sea-ice, as well as its scaling properties, from the scale of the model's computational grid to the basin scale.

## 1 Introduction

Sea-ice is a critical physical component of the climate system as it directly impacts the ocean and the atmosphere, both at the local and the regional scale (Vihma, 2014; IPCC, 2022). In polar regions, the sea-ice cover modulates the radiative and turbulent exchanges of heat, freshwater, gas, and momentum between the ocean and the atmosphere (*e.g.* Taylor et al., 2018, for a review). At the local scale, these fluxes are predominately controlled by the heterogeneity of the sea-ice thickness, which itself is governed by the sea-ice dynamics and the associated formation of leads and ridges. This stresses the relevance of accurately representing sea-ice dynamics in simulations of the coupled/multi-component earth system, such as regional and global climate simulations, and even for short-term sea-ice predictions.

The dynamical behavior of sea-ice is controlled by processes that interact and evolve over a wide range of spatial and temporal scales. This multi-scale nature of sea-ice physics is fascinating and has triggered the curiosity of geophysicists since the early 70's (Coon et al., 1974). More recently, scientific interest in sea-ice dynamics has grown significantly due to the

dramatic retreat and thinning of the Arctic sea-ice cover. In addition, the abundance of new observations of sea-ice kinematics, recorded by both *in-situ* instruments (*e.g.* the buoy trajectories of the International Arctic Buoy Program, https://iabp.apl.uw.edu/data.html) and satellites (*e.g.* the ice trajectories from the RADARSAT Geophysical Processor System, Kwok et al., 1998), has the potential to foster additional advancements in sea-ice modeling.

The dynamics of sea-ice is complex. For instance, Rampal et al. (2008); Weiss et al. (2009) showed that the statistical properties of sea-ice deformation are characterized by a coupled space-time multifractal scaling invariance, similar to what is observed for the deformation of the Earth's crust (Kagan and Jackson, 1991; Marsan and Weiss, 2010). The spatial and temporal scaling properties of sea-ice deformation and their coupling provide evidence for the strong heterogeneity and intermittency that characterizes sea-ice dynamics (Rampal et al., 2008).

Reproducing the discontinuous nature of sea-ice – related to the presence of fractures and leads – in continuous sea-ice models, as well as the complexity of the spatial patterns and temporal evolution of these features, poses a fundamental and major challenge (*e.g.* Bouchat et al., 2022; Hutter et al., 2022). As an effort to tackle this challenge, Dansereau et al. (2016) introduced the Maxwell-Elasto-Brittle rheology (MEB). MEB was implemented into neXtSIM – a large-scale dynamical-thermodynamical *Lagrangian* finite element sea-ice model (Rampal et al., 2016) – to evaluate the performance of this new rheology in a realistic Pan-Arctic simulation. Wintertime sea-ice deformations simulated by MEB in neXtSIM have been first evaluated statistically against satellite observations, in terms of PDFs and scaling invariance properties, in Rampal et al. (2016) and Rampal et al. (2019), and later in the two companion papers of Bouchat et al. (2022) and Hutter et al. (2022), showing satisfying results.

Recently, Ólason et al. (2022) introduced the Brittle Bingham Maxwell rheology (BBM) as an effort to address the incomplete treatment of the convergence of highly damaged sea-ice in MEB. This deficiency of MEB results in the unrealistic representation of the ice thickness after a couple of years of model integration. Indeed, recent realistic BBM-driven multi-decadal simulations performed with neXtSIM have shown to reproduce well (i) the scaling properties of sea-ice deformation from the model grid cell up to the scale of the Arctic basin, and (ii) the thickness pattern of the sea-ice cover, when compared to observations (Ólason et al., 2022; Boutin et al., 2023).

Yet, performing coupled ocean/sea-ice or earth-system CMIP-like simulations with neXtSIM in a numerically-efficient manner remains challenging. Because, the numerical coupling of neXtSIM to a third-party – generally *Eulerian* – GCM component requires the implementation of a relatively inefficient *Lagrangian-Eulerian* coupling strategy. Furthermore, the weak scalability capabilities of neXtSIM when run in parallel on more than a few tenths of processors, and/or at spatial resolutions below typically 10 km, has been shown to substantially hinder the scalability of coupled setups (Samaké et al., 2017). Thus, the implementation of BBM into an *Eulerian* CMIP-class sea-ice model, such as SI3 of NEMO, has the potential to significantly benefit the sea-ice, ocean, and climate modeling communities. First, it will facilitate the assessment of the sensitivity of the simulated sea-ice dynamics to the type of rheology used, in a modeling framework that these communities are familiar with. And second, the good scalability capabilities of NEMO (Tintó Prims et al., 2019) will allow to perform realistic kilometer-scale simulations that use a brittle rheology.

As of today, a few efforts have been made to implement MEB in sea-ice models comparable to SI3 in terms of discretization method and grid, such as the MIT general circulation model (Losch et al., 2010), or LIM, the former sea-ice component of the NEMO modeling system (Rousset et al., 2015). And more recently, Plante et al. (2020) have successfully implemented MEB in the McGill sea-ice model (Tremblay and Mysak, 1997; Lemieux et al., 2008, 2014). Overall, the work of these modeling groups have highlighted some challenging aspects that are specific to the implementation of a brittle rheology in a realistic *Eulerian* model that uses the finite-difference method on a staggered grid. As suggested by the work of Plante et al. (2020), when discretized on the *Arakawa* C-grid (Arakawa and Lamb, 1977), the same grid as used by SI3 (Vancoppenolle et al., 2023), brittle rheologies seem to be more prone to numerical instabilities than their viscous-plastic counterparts. In particular, they report that the stability of their MEB implementation is sensitive to the resort to spatial averaging, an interpolation technique that is traditionally used to relocate certain fields between the staggered points of the grid. Moreover, the need to advect the stress tensor, specific to brittle rheologies, poses another challenge when using the C-grid, because it demands the advection of a scalar field, namely the shear element of the stress tensor, that is defined at the corner-points of the grid cell.

In this paper, we propose a new discretization approach adapted to the numerical implementation of a brittle rheology in an *Eulerian* finite-difference-, C-grid-based sea-ice model. We describe how we have implemented this new approach into SI3, based on the expansion of the C-grid into an *Arakawa* E-grid.

As a first validation step of our BBM implementation, we discuss SI3 results obtained using the idealized test-case setup of Mehlmann et al. (2021), at different horizontal resolutions. Then, as the second step, we compare the sea-ice deformations obtained from realistic coupled ocean/sea-ice simulations of the Pan-Arctic, against those constructed from satellite observations. To serve as a reference, the results of simulations that use the default workhorse setup of SI3 (based on the aEVP rheology of Kimmritz et al., 2016) are also included in both validation steps.

This paper is organized as follows. In section 2, we summarize the equations of the sea-ice dynamics model, discuss the aspects in which the numerical implementation of a brittle rheology may differ from that of a non-brittle viscous-plastic one, and detail the numerical aspects specific to our implementation of BBM into SI3. In section 3, we describe the technical aspects of our SI3 simulations and discuss the results obtained with both the idealized and Pan-Arctic configurations. In section 4, we discuss some numerical aspects of our implementation and some limitations of the BBM rheology. Our conclusions are summarized in section 5. A detailed nomenclature relating the acronyms and symbols used throughout the paper is outlined in Appendix A.

## 2   Model and implementation

### 2.1   Governing equations and constitutive law

The two-dimensional, vertically-integrated, momentum equation for sea-ice reads

$$m \, \partial_t \vec{u} = \vec{\nabla} \cdot (h\boldsymbol{\sigma}) + A \, (\vec{\tau}_a + \vec{\tau}) - m f \, \vec{k} \times \vec{u} - m g \vec{\nabla} H, \tag{1}$$

where $m$ is the mass of ice and snow per unit area, $\vec{u}$ is the ice velocity vector, $h$ is the ice thickness, $\boldsymbol{\sigma}$ is the internal stress tensor, $A$ is the sea-ice fraction, $\vec{\tau}_a$ is the wind stress vector, $\vec{\tau}$ is the ocean current stress, $f$ is the *Coriolis* frequency, $\vec{k}$ is the vertical unit vector, $g$ is the acceleration of gravity, and $H$ is the sea surface height.

In the two-dimensional (plane stresses) case, the stress tensor is written as

$$\boldsymbol{\sigma} = \begin{pmatrix} \sigma_{11} & \sigma_{12} \\ \sigma_{12} & \sigma_{22} \end{pmatrix}. \tag{2}$$

In general, a constitutive law relates $\boldsymbol{\sigma}$ to the strain-rate tensor $\dot{\varepsilon}$, defined as follows:

$$\dot{\boldsymbol{\varepsilon}} = \begin{pmatrix} \dot{\varepsilon}_{11} & \dot{\varepsilon}_{12} \\ \dot{\varepsilon}_{12} & \dot{\varepsilon}_{22} \end{pmatrix} \equiv \begin{pmatrix} \partial_x u & \frac{1}{2}(\partial_y u + \partial_x v) \\ \frac{1}{2}(\partial_y u + \partial_x v) & \partial_y v \end{pmatrix}. \tag{3}$$

As derived by Ólason et al. (2022) (their Eq. 20), the BBM constitutive model yields

$$\partial_t \underline{\boldsymbol{\sigma}} = E\,\underline{\boldsymbol{K}} \cdot \dot{\underline{\boldsymbol{\varepsilon}}} - \underline{\boldsymbol{\sigma}}\frac{1}{\lambda}\left(1 + \tilde{P} + \frac{\lambda}{1-d}\,\partial_t d\right)$$
$$\text{with } \underline{\boldsymbol{K}} = \frac{1}{1-\nu^2}\begin{pmatrix} 1 & \nu & 0 \\ \nu & 1 & 0 \\ 0 & 0 & 1-\nu \end{pmatrix}, \tag{4}$$

where $E$ and $\lambda$ are the elastic modulus and apparent viscous relaxation time of the ice, $\boldsymbol{K}$ is the elastic stiffness tensor, $\tilde{P}$ is a term introduced to prevent excessive ridging (see below), and $d$ is the damage scalar: a variable that represents the density of fractures in the ice at the subgrid-scale. The underbar notation indicates that the tensors are expressed in their *Voigt* form. In a way similar to that of the sea-ice concentration, $A$, the damage modulates $E$ and $\lambda$ as

$$E = E_0(1-d)\,e^{-C(1-A)}, \tag{5}$$
$$\lambda = \lambda_0\Big[(1-d)e^{-C(1-A)}\Big]^{\alpha-1}, \tag{6}$$

where $C$ is the compaction parameter constant and $\alpha$ is a constant exponent greater than $1$. $\alpha$ fulfills the physical constraint that the relaxation time for the stress also decreases as damage increases, and re-increases as the ice heals (*i.e.* damage decreases); because the material respectively loses and recovers the memory of reversible deformations (Dansereau et al., 2016).

The BBM constitutive equation (4) only differs from that of MEB through the inclusion of the term $\tilde{P}$: a threshold between reversible and permanent deformation regimes. As noted by Ólason et al. (2022), the inclusion of this term prevents the excessive convergence that is occurring in MEB simulations lasting longer than a season. For convergent stresses in the range $-P_{\max} < \sigma_I < 0$, the deformation is elastic; otherwise, it is visco-elastic. Ólason et al. (2022) interpret this threshold as the maximum pressure the ice can withstand before ridging. They consequently choose to let the ridging threshold, $P_{\max}$, be proportional to the thickness to the power $3/2$, following Hopkins (1998) and depend exponentially on the concentration,

following Hibler (1979), *i.e.*

$$\tilde{P} = \begin{cases} 0 & \text{if } \sigma_I > 0 \\ -1 & \text{if } -P_{\max} < \sigma_I < 0 \\ \dfrac{P_{\max}}{\sigma_I} & \text{if } \sigma_I < -P_{\max} \end{cases}, \tag{7}$$

where $\sigma_I$ is the (isotropic) normal stress and $P_{\max}$ is the ridging threshold defined as

$$P_{\max} = P_0 \left( \frac{h}{h_0} \right)^{3/2} e^{-C(1-A)}. \tag{8}$$

We follow Dansereau et al. (2016) and Ólason et al. (2022) in using a two-step approach to solve equation 4. As the first step, an initial estimate of $\underline{\sigma}$, noted $\underline{\sigma}^{(i)}$, is calculated assuming no change in damage:

$$\partial_t \underline{\sigma}^{(i)} = E\, \boldsymbol{K} \cdot \underline{\dot{\varepsilon}} - \underline{\sigma} \frac{1}{\lambda} \left( 1 + \tilde{P} \right). \tag{9}$$

Then, as the second step, the following test and adjustment are performed on the state of stress : if $\underline{\sigma}^{(i)}$ is locally overcritical, *i.e.* located outside of the *Mohr-Coulomb* damage criterion (Fig. 1), an increment in ice damage, $d_{\mathrm{crit}}$, is applied such that

$$\underline{\sigma} = d_{\mathrm{crit}}\, \underline{\sigma}^{(i)}, \tag{10}$$

where $\underline{\sigma}^{(i)}$ is the local value of the overcritical stress, and $\underline{\sigma}$ is the corresponding post-failure (*i.e.* post-damage) stress. As discussed in Dansereau et al. (2016); Plante and Tremblay (2021), $d_{\mathrm{crit}}$ is used to scale overcritical stresses back towards the *Mohr-Coulomb* damage criterion, assuming viscous relaxation to be negligible during the (comparatively very fast) damage process. The associated temporal evolution of the damage and adjustment of the stress state is given by

$$\partial_t d = \frac{1 - d_{\mathrm{crit}}}{t_d} (1 - d), \tag{11}$$

$$\partial_t \underline{\sigma} = -\frac{1 - d_{\mathrm{crit}}}{t_d} \underline{\sigma}^{(i)}, \tag{12}$$

where $t_d$ is a characteristic time scale for damage propagation. In the BBM framework, $d_{\mathrm{crit}}$ is expressed as follows:

$$d_{\mathrm{crit}} = \begin{cases} \dfrac{c}{\sigma_{II}^{(i)} + \mu \sigma_{I}^{(i)}} & \text{if } \sigma_I^{(i)} >= -N \\ \dfrac{-N}{\sigma_I^{(i)}} & \text{otherwise} \end{cases}. \tag{13}$$

where $c$ is the cohesion, and $\mu$ is the friction coefficient. The threshold $N$ is used to prevent any numerical instability at very high normal stresses and is set large enough not to impact the solution noticeably.

As suggested by Rampal et al. (2016), a slow restoring process is applied to the damage to account for the healing of ice under refreezing conditions. The rate of decrease of the damage associated with this refreezing is taken proportional to $\Delta T_h$, the temperature difference between basal and surface ice:

$$\partial_t d = -\frac{\Delta T_h}{k_{th}}, \tag{14}$$

where $k_{th}$ is the healing constant. This process can be decoupled from equation (11) due to the large separation of time scales between the healing and damaging processes.

## 2.2 Numerical implementation: brittle versus viscous-plastic rheologies

To understand the extent to which the numerical implementation of a brittle rheology differs from that of a viscous-plastic rheology, let us first review the main differences between these rheologies and their respective classical numerical implementation.

First, the elasto-visco-brittle family of rheologies (MEB, BBM, Dansereau et al., 2016; Ólason et al., 2022) considers unfragmented sea-ice as an elastic and damageable solid. Fragmented sea-ice is a visco-elastic material in which irreversible deformations dissipate the stresses. As opposed to the viscous-plastic frameworks, elasticity is therefore a physical and non-negligible component of the model. It is modulated by the level of damage, $d$, which keeps the memory of the state of fragmentation of the sea-ice cover. The combination of elasticity and damage, even if treated in an isotropic manner, naturally simulates a strong anisotropy and localization of the deformation, down to the nominal spatial and temporal scale (*i.e.* the grid resolution and time-step of the model, respectively, Dansereau et al., 2016; Weiss and Dansereau, 2017; Rampal et al., 2019; Ólason et al., 2022). Therefore, all the mechanically-related fields, such as damage, concentration, thickness and velocity, tend to exhibit very sharp gradients.

Second, in the BBM (as in the MEB) framework, a two-fold approach is used to linearize the system of equations and solve the coupled constitutive and damage evolution equations: (i) an initial estimate, in which stress components are updated based on the constitutive law (Eq. 9), (ii) a damage step in which the *Mohr-Coulomb* test is performed, resulting in a potential adjustment of local overcritical stresses and associated increase in local damage (Fig. 1, Eq. 11 & 12). In viscous-plastic rheologies, which do not incorporate damage, no such two-fold approach is necessary to solve the system of dynamical equations.

A third and major difference between the two types of model is that in brittle models, the stress tensor $\boldsymbol{\sigma}$ is a prognostic variable, while in viscous-plastic models, it is a diagnostic variable. This implies that the implementation of a brittle rheology in an *Eulerian* framework, as opposed to that of a viscous-plastic rheology, should, in practice, consider the advection of $\boldsymbol{\sigma}$, along with other – typically scalar – tracers (see section 2.4). One could argue that, based on a scale analysis, the advection terms of the stress tensor components are somewhat negligible, and that it is therefore acceptable to simply omit these terms (similarly to what is done for the ice velocity in the momentum equations). While these terms are indeed very small, we think that it is important to include them in the *Eulerian* implementation of a brittle model. Because in this type of model, the damage tracer and the stress tensor are inherently bound by a strong interdependence. This interdependence is the consequence of $E$ and $\lambda$ being a function of $d$ (Eq. 5, 6) in the estimate of $\boldsymbol{\sigma}^{(\mathrm{i})}$ (Eq. 9). And the damage, as a tracer that can live on for days, if not weeks, depending on the temperature conditions, has to be advected with the ice velocity, like any other tracer. Therefore, we think that the advection of the stress tensor is necessary to preserve the full spatial consistency between the damage tracer and the internal stress state; in particular in the case of simulations longer than a few days that involve significant sea-ice displacements.

Finally, note that in their numerical implementation of BBM, Ólason et al. (2022) chose to solve the dynamics explicitly using a time-step sufficiently small to account for the propagation of damage in the ice in a physically realistic manner. We follow the same approach in our implementation of BBM into SI3. Typically, this implies using a time-step a few hundred times

smaller (hereafter referred to as dynamical time-step, $\Delta t$) than that used for the thermodynamics and the advection (hereafter referred to as the advective time-step, $\Delta T$). This is implemented by means of a *time-splitting* approach. $N_s$, the number of ($\Delta t$-long) integrations to perform during one advective time-step ($\Delta T$), is imposed by $\Delta T$ and $\Delta x$, the horizontal resolution of the grid:

$$\frac{\Delta x}{\Delta t} > 2C_E \Rightarrow N_s > 2C_E \frac{\Delta T}{\Delta x}, \quad \text{with } N_s \equiv \frac{\Delta T}{\Delta t} \quad \text{and } C_E = \sqrt{\frac{E}{2(1+\nu)\rho_i}}, \tag{15}$$

where $C_E$ is the propagation speed of an elastic shear wave and $\rho_i$ is the density of ice. Note that in practice, if $\Delta T$ is already constrained by $\Delta x$, as in NEMO, the choice of $N_s$ becomes somewhat independent of that of the spatial resolution at which the model is run.

## 2.3 Numerical implementation of the BBM rheology

SI3 uses curvilinear coordinates on a fixed *Eulerian* mesh, and the spatial discretization is achieved by means of the finite-difference method (FD) on the *Arakawa* C-grid (Arakawa and Lamb, 1977). The use of the C-grid is justified based on numerical and practical grounds, as it ensures the exact collocation of ocean and sea-ice horizontal velocity components, which simplifies the coupling with the ocean component of NEMO, and prevents interpolation-related errors as well as extra computational load.

As shown in Fig. 2.a, on the C-grid, tracers are defined at the cell centers, hereafter referred to as the T-point; while the $x$ and $y$ components of vectors are defined at the center of the right-hand and upper edges of each cell, respectively (hereafter U-point and V-point). The point located at the upper-right corner of each cell, known as the vorticity point, is referred to as the F-point. In the literature, this vorticity point is sometimes located at the bottom-left corner of the cell, and is sometimes referred to as the Z-point (Losch et al., 2010; Plante et al., 2020). The U- and V-points may also be located at the left-hand and lower edges of the cell, in which case the F-point is located at the bottom-left corner of the cell (*e.g.* Losch et al., 2010).

Regardless of the type of rheology considered, the main challenge posed by the C-grid is a consequence of the discretized FD expressions of the elements of the strain-rate tensor $\dot{\varepsilon}$ being staggered in space, with the trace elements $\dot{\varepsilon}_{11}$ and $\dot{\varepsilon}_{22}$ defined at the T-point, and the shear rate $\dot{\varepsilon}_{12}$ defined at the F-point. Based on the constitutive law (Eq. 3), the same applies to the stress tensor $\sigma$. This staggering between the diagonal and off-diagonal elements of $\sigma$ is appropriate when considering the discretization of the momentum equation (Eq. 1), because the discretized elements of the vector divergence of $\sigma$ are then defined where they are needed: namely at U- and V-points. However, this staggering becomes an issue whenever the parameterization of the constitutive law requires $\dot{\varepsilon}_{12}$ or $\sigma_{12}$ to be known at a T-point. This is the case, for instance, for the expression of the $\Delta$ parameter in EVP models, or that of the second stress invariant $\sigma_{\text{II}}$ in MEB and BBM (*i.e.* Eq.13), as they require $\dot{\varepsilon}_{12}$ and $\sigma_{12}$, respectively, to be known at T-points. Moreover, in brittle rheologies, a value of $d$ is required not only at the T-point, but also at the F-point in order to estimate $\sigma_{12}^{(i)}$ (Eq. 9). On the C-grid, a common way to interpolate a scalar defined at F-points onto T-points is to simply use the average of this scalar on the four surrounding F- points, and conversely to interpolate from T- to F-points. In the aEVP implementation of SI3 (Kimmritz et al., 2016), the problem posed by the staggering of tensor elements is overcome by using this averaging approach to interpolate the square of the shear rate $\dot{\varepsilon}_{12}$ from F- to T-points . Later on,

the term $P/\Delta$ is also interpolated from T- to F-points in order to estimate $\sigma_{12}$. In their implementation of MEB, Plante et al. (2020) also use this approach to interpolate the damage tracer at F-points. However, they report that using the same approach to estimate $\sigma_{12}$, and hence $\sigma_{\text{II}}$, at T-points when performing the *Mohr-Coulomb* test (Eq.13), results in checkerboard instabilities. The solution they propose to prevent the occurrence of these instabilities is to introduce an additional $\sigma_{12}$ that is defined at T-points. This additional $\sigma_{12}$ is updated at each time step using – as an increment – the average of the four $\sigma_{12}$ increments

computed at the surrounding F-points.

Note that based on the strong interdependence between the internal stress and the damage in brittle rheologies (section 2.2), and the highly-localized nature of the damage, we think that the use of the averaging technique to estimate $d$ at the corner points of the C-grid cells should be avoided if possible. Indeed, by using such a technique, $\sigma_{12}^{(i)}$, as opposed to $\sigma_{11}^{(i)}$ and $\sigma_{22}^{(i)}$, is updated using values of $E$ and $\lambda$ defined by a value of $d$ that might poorly represent the actual value that the rheology, together with

*Mohr-Coulomb* test, would have produced at this location. Because the four-point-average of a variable such as the damage, that is highly heterogeneous in space, even at the grid-point-scale, cannot provide a very accurate and reliable estimate of the local value.

Finally, with the C-grid, the implementation of the advection of $\sigma_{12}$ (F-point) in a way consistent (in terms of the advection scheme used) with that used for $\sigma_{11}$ and $\sigma_{22}$ (T-point) is somewhat challenging. That is because the advection of a scalar

defined at the F-point, using the same scheme as that used for the advection of scalars at T-points, requires the existence of a $u$ and a $v$ at V- and U-points, respectively.

These later considerations have prompted us to consider the use of a new spatial discretization approach for the implementation of BBM on the C-grid.

### 2.3.1  The E-grid approach

To avoid the interpolation of the damage and the stress components between the center and the corner points of the grid cell, and allow the consistent advection of all the components of the stress tensor, an additional sea-ice velocity vector, noted $(\hat{u}, \hat{v})$, is introduced. As shown in figure 2.b, the x-component of this additional velocity, $\hat{u}$, is defined at V-points, while its y-component, $\hat{v}$, is defined at U-points. Similarly, the damage tracer is also duplicated, with an additional occurrence at the upper-right corners of the grid cell, *i.e.* at F-points. This grid staggering arrangement corresponds to that of the *Arakawa* E-grid

(Arakawa and Lamb, 1977; Janjić, 1984; Maier-Reimer et al., 1993).

As suggested by Fig. 3.b, the E-grid can be seen as a superposition of two C-grids, in which the cell center of the additional C-grid coincides with the upper right corner of the original C-grid. For convenience, we will refer to these two grids as F-centric (additional) and T-centric (original), respectively.

In order to minimize the number of modifications and rewriting in the SI3 code, the idea was to restrict the use of this *E-*

*augmented* C-grid to the rheology/dynamics module only. The rest of the code, which includes the thermodynamics, remains unmodified and relies entirely on the standard C-grid. As such, only rheology-specific tracers are defined in the E-grid fashion, *i.e.* at both T- and F-points. In our case, this applies only to the ice damage $d$ and components of the internal stress tensor (even though components of a tensor cannot be considered exactly as tracers when it comes to the advection, see section 2.4).

However, global tracers, such as ice concentration and thickness, which are updated within the thermodynamics module, remain
defined at the T-point only. Consequently, these tracers are interpolated at the F-point within the rheology module whenever
needed.

To summarize, in the proposed rheology-specific *E-augmented* C-grid approach, as shown in figure 3, the conventional C-grid model variables are augmented with: (i) the u-velocity component at V-points and v-velocity component at U-points, (ii) the ice damage, $\sigma_{11}$ and $\sigma_{22}$ at F-points, and (iii) $\sigma_{12}$ at T-points. This approach implies that most of the equations related to
245 the dynamics, including constitutive and momentum equations, as well as the advection, have to be solved on both the T- and F-centric grids. As detailed in Appendix B, the exact same discretization and numerical schemes can be used on both grids, with only the indices of the velocity components on the F-centric grid requiring particular attention: $\hat{u}_{i+1,j}$ and $\hat{v}_{i,j+1}$ have to be used as the counterparts of $u_{i,j}$ and $v_{i,j}$ on the T-centric grid (Fig.3.b). This is true for the computation of the strain-rate tensors (B2.1), constitutive equation (B2.2), momentum equation (B3), divergence of the stress tensor (B3.1), advection, etc.
At this stage it is important to note that the doubling of the number of computational points implied by the transition to the E-grid, in no way relates to an increase of the spatial resolution of the original C-grid. Because the FD discretization of spatial derivatives on the E-grid (see Appendix B) still relies on the same local spatial increment, *i.e.* $\Delta x$, as that of the original C-grid, regardless of the sub-grid considered (T- or F-centric).

### 2.3.2 The separation of solutions and how it is restrained

With the *E-augmented C-grid* approach, all rheology-specific prognostic variables are defined at the points where their value is required, and no interpolation is needed to solve the equations. It does, however, result in an apparent over-determination, which allows the T- and F-centric solutions to evolve somewhat independently from one another. This separation of solutions rapidly degenerates into unrealistically noisy solutions as the spatial consistency of the fields between the two grids deteriorates.

This problem of grid separation has been known since the early adoption of the E-grid by the community (Arakawa, 1972;
Mesinger, 1973; Janjić, 1974; Janjić and Mesinger, 1984; Mesinger and Popovic, 2010). It is mostly discussed in the context of the shallow-water equations, and is is often referred to as "(short) gravity wave decoupling" or "lattice separation". Various treatments and methods have been proposed, from filtering approaches to more advanced ones such as the introduction of auxiliary velocity points, midway between the neighboring tracer points (Mesinger, 1973; Janjić, 1974; Mesinger and Popovic, 2010). Interestingly, the E-grid was used in the Hamburg Large-Scale Geostrophic (LSG) model (Maier-Reimer et al., 1993) in
order to achieve more accurate geostrophic balance, while retaining some advantages of the C-grid such as the straightforward discretization of the divergence. In their model, the problem of grid separation, already limited due to the use of a monthly time-step, was overcome through adding horizontal viscosity and diffusion. Recently, Konor and Randall (2018) also mentioned the need to introduce a "horizontal mixing process" to avoid the "separation of solutions" when using the E-grid.

The cause of the separation of the two solutions resides in the weak coupling between the two grids, as they only exchange
very little information. Specifically, in our case, the only exchange of information between the T- and F-centric grids occurs via the shear stress $\sigma_{12}$ and the ice velocity vector. The estimate of $\sigma_{12}^{(i)}$ of the T-centric grid (at F-point), based on equation 9, uses $\hat{E}$, $\hat{\lambda}$ and $\hat{\tilde{P}}$ of the F-centric grid (at F-point), and conversely for $\hat{\sigma}_{12}^{(i)}$. Similarly, the correction of $\sigma_{12}^{(i)}$ in equation 12, if

occurring, uses $\hat{d}_{\mathrm{crit}}$ and $\hat{t}_d$. For the velocity, the exchange of information occurs in the *Coriolis* term of equation 1, and through the advection of $\sigma_{12}$ via $\hat{u}, \hat{v}$ (and that of $\hat{\sigma}_{12}$ via $u, v$). As suggested by results discussed later in this section, this exchange of information is not sufficient enough to prevent the decoupling of the solutions between the two grids. Hence, a numerical treatment is required to constrain the T- and F-centric solutions to remain spatially consistent with one another.

During the early phase of our development, we considered, implemented, and tested a variety of such treatments. As of now, only one has proven able to prevent the grid separation issue without leading to noisy and/or unrealistic solutions. This treatment, which operates sequentially on the T- and F-centric stress tensors at the dynamical time-step level, is hereafter referred to as the *cross-nudging*. It consists in nudging each vertically-integrated component of the T-centric stress tensor $\boldsymbol{\sigma}$ towards its F-centric counterpart (tensor $\hat{\boldsymbol{\sigma}}$) interpolated at the relevant point under even time-step integrations, and conversely under odd time-step integrations. This is written as follows:

$$
\begin{pmatrix} \sigma_{11} \\ \sigma_{22} \\ \sigma_{12} \end{pmatrix} = \begin{pmatrix} \sigma_{11} \\ \sigma_{22} \\ \sigma_{12} \end{pmatrix} - \frac{\gamma_{cn}}{N_s} \begin{pmatrix} \sigma_{11} - \bar{\hat{\boldsymbol{\sigma}}}^{\scriptscriptstyle T}_{\mathbf{11}}/h \\ \sigma_{22} - \bar{\hat{\boldsymbol{\sigma}}}^{\scriptscriptstyle T}_{\mathbf{22}}/h \\ \sigma_{12} - \bar{\hat{\boldsymbol{\sigma}}}^{\scriptscriptstyle F}_{\mathbf{12}}/\bar{h}^{\scriptscriptstyle F} \end{pmatrix} \qquad \text{(even dynamical time-step )}
$$

$$
\begin{pmatrix} \hat{\sigma}_{11} \\ \hat{\sigma}_{22} \\ \hat{\sigma}_{12} \end{pmatrix} = \begin{pmatrix} \hat{\sigma}_{11} \\ \hat{\sigma}_{22} \\ \hat{\sigma}_{12} \end{pmatrix} - \frac{\gamma_{cn}}{N_s} \begin{pmatrix} \hat{\sigma}_{11} - \bar{\boldsymbol{\sigma}}^{\scriptscriptstyle F}_{\mathbf{11}}/\bar{h}^{\scriptscriptstyle F} \\ \hat{\sigma}_{22} - \bar{\boldsymbol{\sigma}}^{\scriptscriptstyle F}_{\mathbf{22}}/\bar{h}^{\scriptscriptstyle F} \\ \hat{\sigma}_{12} - \bar{\boldsymbol{\sigma}}^{\scriptscriptstyle T}_{\mathbf{12}}/h \end{pmatrix} \qquad \text{(odd dynamical time-step )}
$$

$$(16)$$

where $\gamma_{cn}$ is the cross-nudging coefficient, $N_s$ is the time-splitting parameter (Eq. 15), the bar notation denotes the spatial interpolation from F- to T-points or T- to F-points (see Eq. A1 in Appendix A3), and stress components in bold are vertically-integrated (Appendix A4). Each of the two tensors is "corrected" $N_s/2$ times during the course of one advective time-step $\Delta T$. Similarly to what is used for the term of the divergence of the stress tensor in the equation of momentum (Eq. 1), it is important to consider the vertically-integrated stresses, *i.e.* $h\boldsymbol{\sigma}$, when applying the cross-nudging. Because the physical quantity of interest, that has to be compared between neighboring cells, is the force per horizontal unit length rather than the force per unit area (vertical section). Doing so prevents the introduction of errors that stem from a strong thickness discrepancy, between the center- and the corner-points of the grid cell considered, in regions with abrupt gradients in ice thickness. Note that since our current implementation uses a thickness at F-points ($\bar{h}^{\scriptscriptstyle F}$) that is the average of the thickness at the four surrounding T-points, these errors would remain relatively small in any case.

The form of the term that modulates the nudging intensity, *i.e.* $\gamma_{cn}/N_s$, ensures that the level of cross-nudging undergone by the two tensors under one $\Delta T$ is primarily controlled by $\gamma_{cn}$ and remains somewhat independent of the choice of $N_s$.

We chose to apply the cross-nudging (CN) on $\boldsymbol{\sigma}^{(\mathrm{i})}$ (Eq. 9), before any potential upcoming correction is applied following the *Mohr-Coulomb* test (Eq.12). Here too, we justify our choice based on the strong damage-stress interdependence (section 2.2). If we apply the CN on stresses that have undergone the *Mohr-Coulomb*-related correction, then we may propagate, to neighbor points, stress values that have been corrected, without propagating the associated increase in damage. In that case, one could consider applying the CN on the damage as well, but as discussed in section 2.3, smoothing the damage field is something we want to avoid.

The 4-point spatial interpolation used in the cross-nudging inevitably results in the introduction of a smoothing of the solution in space. As such, $\gamma_{cn}$, is chosen to achieve the best compromise between the smoothing and the coupling of the T- and F-centric solutions. We have performed sensitivity tests with our Pan-Arctic setup and we conclude, relying exclusively on visual assessment of the simulated fields, that the right compromise is achieved when $\gamma_{cn}$ is typically of the order of 1, with $\gamma_{cn} = 1$ being the value used in our experiments. As illustrated in figure 4, with a value below 1, the solutions becomes increasingly noisy as $\gamma_{cn}$ approaches zero. In particular, the damage field tends to exhibit strongly unrealistic straight-line features of high damage that are aligned along the x- or y-axis of the grid. Our results suggest that values of $\gamma_{cn}$ above typically 2 lead to an excessive smoothing of the solutions (as shown for example for $\gamma_{cn} = 10$ in figure 4.f). The value of $\gamma_{cn}$ appropriate for a given model setup is likely to be dependent on different factors that we have not identified yet. As such we can only recommend potential users of our implementation to consider $\gamma_{cn}$ as a tuning parameter that should be adjusted for a given setup. However, simulations that we have performed at spatial resolutions of 1, 2, 4 and 10 km with the idealized test-case discussed in section 3.1 (not shown), suggest that $\gamma_{cn}$ is only weakly, if not, influenced by the spatial resolution at which the model is run (values between typically 0.5 and 2 consistently yielding what we refer to as the best compromise).

## 2.4 Horizontal advection

In neXtSIM, the *Lagrangian* finite-element model used by Ólason et al. (2022), the advection occurs implicitly at each advective time-step (also corresponding to the thermodynamics time-step) through the ice-velocity-driven displacement of the mesh elements. As such, the rate of change of a prognostic scalar $\phi$ is $\dot{\phi} \equiv \partial_t \phi$. In the present *Eulerian* context, however, the term relative to the horizontal advection has to be considered so that the rate of change of $\phi$ is now $\partial_t \phi + U \partial_x \phi + V \partial_y \phi$. In our implementation, as pointed out by Ólason et al. (2022), this advection term is computed and added to the trend of the prognostic scalar considered every advective time-step. Thus, the sea-ice velocity vector $U, V$ we consider for the advection, at the advective time-step level, is the mean of the $N_s$ successive velocity vectors $(u, v)$ calculated under one time-splitting instance. $U, V$ can also be seen as the sum of the $N_s$ successive displacement vectors, hence the total displacement vector during one advective time-step, divided by the advective time-step.

We use the *second-order-moments-conserving* advection scheme of Prather (1986) available in SI3 to advect the damage and the components of the stress tensors (considered as scalar for now, see section 2.4.1). Technically, the damage and stress tensor components defined at the T-point ($d$, $\sigma_{11}$, $\sigma_{22}$ and $\hat{\sigma}_{12}$) are advected using $U$ and $V$ defined at U- and V-points, respectively. Their F-point counterparts ($\hat{d}^F$, $\hat{\sigma}_{11}$, $\hat{\sigma}_{22}$ and $\sigma_{12}$) are advected using $\hat{U}$ and $\hat{V}$ defined at V- and U-points, respectively. In practice, the exact same implementation of the advection scheme can be used to perform the advection at T- and F-points; the only difference being that for the advection of F-point scalars, the spatial indexing of the velocity components is staggered by 1 cell. Namely, $\hat{U}_{i+1,j}$ and $\hat{V}_{i,j+1}$ have to be used in place of $U_{i,j}$ and $V_{i,j}$ (Fig. 3.b).

As it is commonly done in sea-ice models, and justified by a scale analysis of the momentum equation, the term for the advection of momentum is neglected.

### 2.4.1 Advection of the internal stress tensor

In the *Eulerian* framework, the rate of change of a second-rank tensor must introduce additional terms to the material time derivative in order for the dynamics of the tensor to remain independent of the frame of reference (Oldroyd, 1950; Larson, 1988; Hinch and Harlen, 2021; Stone et al., 2023). These terms account for the effects of rotation and deformation of the medium on the evolution of the stress tensor, and are gathered here in a symmetric tensor $\dot{L}$:

$$\dot{\boldsymbol{\sigma}} \equiv \frac{D\boldsymbol{\sigma}}{Dt} + \dot{L} \equiv \partial_t \boldsymbol{\sigma} + (\vec{U} \cdot \vec{\nabla})\boldsymbol{\sigma} + \dot{L} \tag{17}$$

As stressed by Snoeijer et al. (2020), one faces a "a somewhat unpleasant ambiguity" as two different formulations exist for $\dot{L}$. Both formulations are equally-valid in terms of frame invariance, and so is any linear combination of the two. The first formulation yields the so-called upper-convected time derivative of $\boldsymbol{\sigma}$, noted $\overset{\triangledown}{\boldsymbol{\sigma}}$,

$$\overset{\triangledown}{\boldsymbol{L}} = -(\vec{\nabla} \otimes \vec{U})^{\top}\boldsymbol{\sigma} - \boldsymbol{\sigma}(\vec{\nabla} \otimes \vec{U}), \tag{18}$$

which, in component form, simplifies into

$$\overset{\triangledown}{L}_{11} = -2(\dot{\varepsilon}_{11}\,\sigma_{11} + \partial_y U\,\sigma_{12}),$$
$$\overset{\triangledown}{L}_{22} = -2(\dot{\varepsilon}_{22}\,\sigma_{22} + \partial_x V\,\sigma_{12}), \tag{19}$$
$$\overset{\triangledown}{L}_{12} = -(\dot{\varepsilon}_{11} + \dot{\varepsilon}_{22})\,\sigma_{12} - \partial_x V\,\sigma_{11} - \partial_y U\,\sigma_{22}.$$

The second formulation yields the lower-convected time derivative, noted $\overset{\triangle}{\boldsymbol{\sigma}}$,

$$\overset{\triangle}{\boldsymbol{L}} = \boldsymbol{\sigma}(\vec{\nabla} \otimes \vec{U})^{\top} + (\vec{\nabla} \otimes \vec{U})\boldsymbol{\sigma}, \tag{20}$$

with

$$\overset{\triangle}{L}_{11} = 2(\dot{\varepsilon}_{11}\,\sigma_{11} + \partial_x V\,\sigma_{12}),$$
$$\overset{\triangle}{L}_{22} = 2(\dot{\varepsilon}_{22}\,\sigma_{22} + \partial_y U\,\sigma_{12}), \tag{21}$$
$$\overset{\triangle}{L}_{12} = (\dot{\varepsilon}_{11} + \dot{\varepsilon}_{22})\,\sigma_{12} + \partial_y U\,\sigma_{11} + \partial_x V\,\sigma_{22}.$$

These formulations of $\dot{L}$ are straightforward to implement in the model as they only involve multiplications between the components of tensors $\dot{\varepsilon}$ and $\boldsymbol{\sigma}$, which are all defined at both T- and F-points with the E-grid. Therefore, we have implemented both formulations in SI3. In our implementation, the standard advection trend for each tensor component, corresponding to the term $(\vec{U} \cdot \vec{\nabla})\boldsymbol{\sigma}$ in equation 17, is computed using the identical scheme as that used for regular scalar fields. The tensor-specific advection trend, $\dot{L}$, is computed according to equation 19 or 21. These two contributions are computed independently from one another, using stress values that have not been updated yet by the advection process.

We have chosen to use the upper-convected formulation in both the idealized and Pan-Arctic simulations presented in this paper. This choice is purely arbitrary and is not based on scientific considerations of any kind. It relies solely on the fact that

the upper-convected formulation has been favored in the literature since *Oldroyd*, who introduced both formulations in his 1950 paper, found that his model would only realistically represent the flow around a rotating rod when using this formulation, as opposed to the lower-convected one (Hinch and Harlen, 2021). Nevertheless, two twin simulations of our reference BBM simulation (see section 3.2) have been run, one using the traditional material derivative (*i.e.* $\dot{L} = 0$), and the second the lower-convected formulation. All the diagnostics and deformation statistics discussed later in this paper have been performed on these two additional simulations and no significant differences have been identified between the three options (PDFs of the total deformation for the reference and additional simulations are provided in figure C.3 in Appendix C as an example).

Further work, involving for example the design of new idealized test-cases, should be conducted to address this ambivalence and help identify which time-derivative formulation (or combination of them, such as the *Gordon-Schowalter* time-derivative discussed by Dansereau et al., 2016) is best adapted to sea-ice rheology.

## 2.5 Construction of observed and simulated *Lagrangian* sea-ice deformations

Our assessment of the NEMO Pan-Arctic simulations relies on a multiscale statistical analysis of sea-ice deformation rates constructed using observed and simulated *Lagrangian* sea-ice trajectories, during winter 1996-1997. Observed trajectories are taken from the RGPS (RADARSAT Geophysical Processor System Lagrangian trajectories) dataset of Kwok et al. (1998), while simulated trajectories are generated from the *Eulerian* sea-ice velocities of SI3 by means of sea-ice particle tracker software.

The preprocessing and computing approach we use to construct sea-ice deformations out of the raw RGPS *Lagrangian* trajectories is very similar to that used by Ólason et al. (2022), the main difference being that it relies on the tracking of quadrangles rather than triangles. To construct the SI3-derived synthetic version of these deformations, the tracking software seeds the identical points as those involved in the definition of the quadrangles selected for computing the RGPS deformation, respecting their initial position in space and time. These points are then tracked for about three days, using the hourly-averaged *Eulerian* sea-ice velocities of SI3; the exact tracking duration used being that of the time interval between the two consecutive positions of the corresponding RGPS point.

The period of interest, chosen to match that of the production segment of the two simulations, *i.e.* December 15[th] 1996 to April 20[th] 1997 (see Section 3.2), is divided into 3-day long bins, which corresponds to the nominal time resolution of the RGPS dataset.

As the first step of our selection process, for each 3-day bin, an initial subset of the RGPS points is selected. Each point of this initial subset must satisfy the following requirements:

– the point has at least one position that occurs within the time interval of the bin; this position, or the earliest-occurring one if more than one occurrence, is selected and referred to as position #1

– position #1 is located at least 100 km away from the nearest coastline

- the point has at least one upcoming position that occurs 3 days after position #1, with a tolerated deviation of ± 6 hours, referred to as position #2 (in the event of more than one position satisfying this requirement, the position yielding the time interval the closest to 3 days is selected)

### 2.5.1 Quadrangulation of selected trajectories

As the second step, a *Delaunay* triangulation is performed on this initial subset of points at position #1. Triangles with an area smaller than 25%, or larger than 75% of the nominal area of the quadrangles to be constructed (*i.e.* the square of the spatial scale under consideration), or with an angle below 5°or above 160°, are excluded. Neighboring pairs of remaining triangles are then merged into quadrangles in order to transform the triangular mesh into a quadrangular mesh.

*Aspiring quadrangles* at position #2 are constructed by simply considering the exact same respective sets of 4 points as those defining quadrangles at position #1.

Then, as the third and final step of the selection process, only points that define quadrangles that satisfy the following requirements, at both position #1 and position #2, are retained:

- the square root of the area of the quadrangle falls within a ±12.5% range of agreement with the horizontal scale under consideration

- the time position of each of the four points defining the vertices of the quadrangle should not differ from that of any of the other three points by more than 60 s

- the thresholds for the minimum and maximum angles allowed are 40°and 140°, respectively

### 2.5.2 Computation of deformation rates based on the quadrangles

For all quadrangles selected in a given 3-day bin, strain-rates are computed based on position #1 and position #2 of the quadrangle, using the line-integral approximations (see *e.g.* Lindsay and Stern, 2003, equations 10-14).

Similarly to what is used as a $\Delta t$ to estimate velocities from displacements when computing the deformation rates, the actual time location (*i.e.* date) assigned to each deformation rate is not that of the center of the 3-day bin considered. Instead, we assign the time that corresponds to the center of the time interval defined by position #1 and position #2 of each quadrangle. Spatial location of the deformation rates corresponds to the barycenter of the 4 vertices of the quadrangle considered at the center of this same time interval.

### 2.5.3 Construction of the simulated *Lagrangian* sea-ice trajectories

To save computer resources, only the points from which valid RGPS deformation estimates were computed are retained.

Each of these points is seeded using the same initialization date and location (bilinear interpolation) as its RGPS counterpart. It is then tracked during the same time interval of about 3 days (± 6 h) that separates the two consecutive records of the RGPS point considered. The tracking software uses a time-step of 1 h and feeds on the hourly-averaged simulated sea-ice velocities

of the SI3 experiments. Note that only the conventional C-grid velocities $u, v$ of the T-centric cell are used to track the points ($\hat{u}$ and $\hat{v}$, available in SI3-BBM, are not used).

## 3    Model Evaluation

We use the version 4.2.2 of the NEMO modeling system (Madec et al., 2022) as the basis for the development of the BBM rheology code extension, and to perform both the idealized and coupled Pan-Arctic simulations to be assessed. Since version 4, the default sea-ice component of NEMO is SI3 (Vancoppenolle et al., 2023).

### 3.1    Idealized simulations

To provide a first qualitative evaluation of the behavior of our BBM implementation, SI3 simulations were run on the idealized test-case setup introduced by Mehlmann et al. (2021); including simulations using the default aEVP-driven SI3 setup for reference purposes. This test-case, defined on a 512 km wide square domain, simulates a cyclone traveling in the north-eastward direction over a thin layer of ice ($h \simeq 0.3$ m) that floats on an anticyclonically-circulating ocean. This test-case is well suited to illustrate the influence of the grid-discretization on rheology-related processes such as the representation of LKFs (Danilov et al., 2022, 2024), which makes it particularly relevant to our study. We use the identical setup and parameter values as defined in Mehlmann et al. (2021) (see the *Code and data availability* section to access the SI3 namelists and forcing files). SI3 is run in standalone mode using SAS, the *stand-alone-surface* module of NEMO. In SAS mode, SI3 uses a prescribed surface ocean state (current, height, temperature and salinity) instead of being coupled to the ocean component of NEMO as in our Pan-Arctic simulations (section 3.2).

The results of this test case, for both the aEVP and BBM rheology, are shown in figure 5. First we note that for the SI3 implementation of aEVP, the deformation fields obtained are in a qualitative agreement with those of Mehlmann et al. (2021) (see for instance their figure 7). Results obtained with our BBM implementation, appear somewhat very different from those obtained with aEVP. We note the presence of a circular network of LKFs, that contrast, by their arrangement, with the "spider-web-like" arrangement of the LKFs in the aEVP solution. In the BBM-driven simulation these LKFs are also simulated in the 10 km setup (Fig. 5.h). The spatial pattern of the LKFs, particularly those accommodating the highest deformation, also look qualitatively different: apparently longer and with circular and concentric shapes with respect to the center of the forcing cyclone in the case of BBM, shorter and in radial alignment with respect to the forcing cyclone in the aEVP case.

We also find that the background deformation field is close to zero in the BBM solution, except along the LKFs, whereas the deformation looks more homogeneous in space in the aEVP solution. This can also be seen on the respective PDFs (Fig. 5.c,f,i) that exhibit different shapes and heavier tails in the BBM solution. As shown in figure C.1, we have verified that the aEVP solution is not too significantly impacted by the number of iterations used in the aEVP solver of SI3 by conducting the same aEVP experiments with a $N_{\text{EVP}} = 1000$ instead of $N_{\text{EVP}} = 100$.

Finally, we note that the solutions do not contain any apparent numerical instabilities or noise, neither for aEVP nor BBM. The LKF-like features in the BBM solution at 10 km show a tendency to align horizontally, vertically and diagonally with the

455 grid. As of now, we are unable to provide an explanation on the mechanism responsible for these alignments; apart from their apparent connection with the use of a relatively coarse spatial resolution, as the solution obtained with the 4 km setup, seems to be rid of them.

## 3.2 Coupled ocean/sea-ice Pan-Arctic simulations

The Pan-Arctic simulations use SI3 coupled to the 3D-ocean component of NEMO, named OCE. They are performed on the
460 so-called NANUK4 regional configuration, which is an Arctic extraction of the standard global 1/4°resolution NEMO gridded horizontal domain known as ORCA025 (Barnier et al., 2006). As such, and as shown in Figure 6, the actual grid resolution of NANUK4 typically spans 10 up to 14 kilometers in the central Arctic region. NANUK4 features two open lateral boundaries; the southernmost boundary is located at about 39°N in the Atlantic ocean, and the second boundary is located south of the Bering Strait, at about 62°N in the Pacific ocean. The vertical $z$-coordinate grid used for the ocean features 31 levels with a $\Delta z$
of 10 m at the surface up to about 500 m at the deepest level, at a depth of 5250 m.

The hindcast nature of the simulations is achieved through the use of interannual surface (atmospheric) and lateral (3D-ocean) forcings. For the atmospheric forcing, both the ocean and the sea-ice components receive, as surface boundary conditions, fluxes of momentum, heat and freshwater at the air-sea and air-ice interface, respectively. These fluxes are computed every hour by means of bulk formulae using the hourly near-surface atmospheric state from the ERA5 reanalysis of the ECMWF
(Hersbach et al., 2020) and the prognostic surface temperature of the relevant component (SST or ice surface temperature). For the lateral boundary conditions of OCE, the 3D ocean is relaxed towards the monthly-averaged 3D horizontal velocities, temperature, salinity and SSH (2D) of the GLORYS2v4 [1] ocean reanalysis version 4 (Ferry et al., 2012).

Both OCE and SI3 use a time-step of $\Delta T$ =720 s, the advective time-step. The coupling between these two components is also done at each advective time-step.

Our control simulation, named SI3-default, uses the default SI3 setup as provided in NEMO, and thereby, uses the aEVP rheology of Kimmritz et al. (2016). The second simulation, named SI3-BBM, only differs from SI3-default through the use of our implementation of the BBM rheology in place of aEVP and a higher value of the air/ice drag coefficient. Value of parameters relevant to both rheologies used in the two simulations are provided in tables 1 and 2, respectively.

The two simulations are initialized on December 1st 1996 using the *restart* data generated at the end of a two-month spinup
performed with the SI3-default setup, and run until April 20th 1997. This spinup is initialized on October 1st 1996 by using the daily-averaged ocean and sea-ice data of the GLORYS2v4 reanalysis as an initial condition. More specifically, OCE is initialized at rest (no current) with the 3D temperature and salinity state of the reanalysis. The two-month spinup we use is long enough to get the ocean velocities in the upper ocean into a good state with the given temperature and salinity fields. SI3 is initialized with the daily-averaged sea-ice concentration and thickness of the reanalysis. This implies that SI3-BBM is
initialized with a value of ice damage set to zero everywhere, which poses no issue as the time required to spinup the damage is very short (Bouillon and Rampal, 2015; Rampal et al., 2016), typically of the order of a few days. The analysis of the results

---

[1]https://data.marine.copernicus.eu/product/GLOBAL_REANALYSIS_PHY_001_031/description

is performed on the period December 15$^{th}$ – 20$^{th}$ 1997), leaving the upper ocean and the sea-ice cover in SI3-BBM two weeks to respond to the changed rheology, which should be ample.

For these simulations, the adjustable tuning parameters of SI3 are kept as close as possible to those of the reference configuration of NEMO (tables 1 & 2). As such, the thermodynamic component uses 5 ice-categories. Yet, the ice-atmosphere drag coefficient $C_D^{(a)}$ has been adjusted from $1.4 \cdot 10^{-3}$ to $2 \cdot 10^{-3}$ in SI3-BBM in order for the mean simulated deformation rate at the 10 km scale to be in agreement with that derived from the satellite observations against which we evaluate the model in section 2.5. In SI3-default, the default values of $1.4 \cdot 10^{-3}$ satisfies this requirement and is left unchanged. For the time-splitting approach (section 2.2), we use a dynamical time-step of 7.2 s in SI3-BBM, which relates to a time-splitting by a factor $N_s = 100$.

### 3.3   Comparison of simulated sea-ice deformation statistics against satellite data

#### 3.3.1   Probability density function of sea-ice deformation rates

As illustrated by the maps of the 3-day total deformation rates shown in figure 7, RGPS clearly exhibits narrow and long features, commonly called *Linear Kinematic Features* (LKFs, Kwok, 2001) along which the deformation is concentrated. Visually, LKFs simulated by SI3-BBM appear somewhat realistic, both in terms of length and orientation, and the magnitude of the deformation rates along these LKFs is similar to that of RGPS. We note that SI3-default exhibits very smooth fields of deformation with no or very little number of such localized features; this is consistent with the findings of recent studies that evaluate VP-driven sea-ice simulations run with a horizontal grid size larger than a few kilometers, *i.e.* typically more than 5km (*e.g.* Ólason et al., 2022; Bouchat et al., 2022).

The probability density functions (PDFs) of the total deformation rates depicted in figure 8.d show that SI3-BBM exhibits a heavy tail similar to that of RGPS and that it can be approximated by a power-law over the values corresponding to the last two percentiles of the RGPS distribution, although with slightly different exponents (-2.9 and -3, respectively). A look at the other invariants of the deformation (*i.e.* shear, divergence, and convergence rates) in figure 8.a,b,c shows that SI3-BBM simulates large deformation events as seen in the observations, which suggests the ability of BBM to capture the heterogeneous character of sea-ice deformation in this setup. In contrast, SI3-BBM is clearly unable to reproduce the observed convergence over the full range of values present in the RGPS data (Fig. 8.c). This deficiency of the BBM rheology is further discussed in section 4.2.

Our results suggest a propensity for SI3-default to underestimate the extreme values of deformation rates. This insufficiency of the model could very likely be mitigated by conducting a finer tuning of the parameters related to the viscous-plastic rheology, in particular through the better adjustment of the ratio between the ice compressive strength and the ice shear strength (Bouchat and Tremblay, 2017). Yet, conducting such a tuning is out of the scope of this paper.

### 3.3.2 Time-series of sea-ice deformation rates

Following Ólason et al. (2022), we examine the 90[th] percentile of total deformation (P90), chosen for its sensitivity to the high values that contribute to shaping the long tail of the PDFs of deformations. Technically, P90 is the value of deformation below which 90% of deformation values in the frequency distribution fall. P90 is computed from each snapshot of deformation from mid-December 1996 to late April 1997 to evaluate the temporal evolution of the deformation. Values of P90 from RGPS and SI3-BBM are plotted and inter-compared using the bias ($b$), root mean square error (RMSE, $e$), and the *Pearson* correlation coefficient ($\rho$). For reference, we also provide these statistics for SI3-default. In addition to the 90[th] percentile, we also consider the 95[th] and 98[th] percentiles.

As illustrated in figure 9, SI3-BBM is in fairly good agreement with the observations, in particular for the P90 values, (see table 3). We note, however, that despite the ability of SI3-BBM to reproduce a variability similar to that observed, the higher the percentile value, the lower the agreement between the magnitudes. This suggests an inability of the BBM rheology to capture the most extreme deformation events.

We note that the biases and RMSEs are very similar between the two simulations. For P90 and P95, the values suggest a fairly good agreement between the two simulations and the observations. The values for P98, however, highlight the incapacity of both models to reproduce extreme deformation events. This is in qualitative agreement with what Ólason et al. (2022) already reported. Yet, further investigation remains necessary to assess whether this is inherent to the BBM model, or could be improved through the better adjustment of the rheological parameters.

### 3.3.3 Multifractal scaling analysis

The presence of heavy tails in the distributions shown in figure 8 implies that one needs to consider higher moments than the mean to fully describe the statistics of the sea-ice deformation process (Sornette, 2006). Following Marsan et al. (2004), the calculation of moments should be limited to those of order $q > 0$, because zero values exist in the deformation field. And they should not exceed the order $q = 3$ since a transition is observed between typically $q_c = 2.5$ and $q_c = 3$ (Schertzer and Lovejoy, 1987). The reason for this is that the tails of the distributions for RGPS and SI3-BBM is close to a power-law with an exponent of about -3, hence their moments of order $q > q_c$ diverge (Savage, 1954).

We performed a multifractal spatial scaling analysis of the RGPS total deformation rates and their simulated counterparts, considering the moments $q = 1, 2$ and $3$ of the distributions. As shown in figure 10, both the observed and simulated statistics (mean, variance, and skewness) are following power-laws. In particular, the observed mean sea-ice deformation rate $\langle \dot\varepsilon \rangle$ is particularly well reproduced in SI3-BBM across the full range of spatial scales considered for this analysis, and can be approximated by a power-law scaling $\langle \dot\varepsilon \rangle \sim L^{-\beta}$, where $L$ is the spatial scale and $\beta$ an exponent of about 0.15. We note that the atmospheric drag coefficient was used as the adjustment parameter in SI3 (section 3.2), which led to the use $C_D^{(a)} = 2 \cdot 10^{-3}$ in SI3-BBM, while the default $C_D^{(a)} = 1.4 \cdot 10^{-3}$ in SI3-default did not require to be adjusted. Consistent with the results previously discussed, the higher moments, which characterize the largest and most extreme values of the distributions, remain underestimated in SI3-BBM compared to that derived from observations. Indeed, the exponents of the power-law that fits the

SI3-BBM data ($\beta$ =-0.6 and -1.34, for $q$ = 2 and 3, respectively) are lower than those derived from RGPS data ($\beta$ =-0.7 and -1.52). This indicates that SI3-BBM is not fully capturing the strength of the spatial scaling of sea-ice deformation revealed by the observations, or in other words that it fails to achieve the extremely high degree of spatial localization of the LKFs in the observations.

Figure 10 suggests that the total deformation rates simulated by SI3-default cease to follow the expected power-law for scales larger than typically 100 km. This is in line with published results (*e.g.* Hutter et al., 2018; Bouchat et al., 2022). Hutter et al. (2018) argue that the VP model needs approximately ten grid cells to be able to resolve features, which suggests that the "effective resolution" of the model is ten times coarser than that of the numerical grid on which it is run. This implies that one should instead consider fitting the deformation rates at a resolution ten times coarser than that used by the model, *i.e.* 130 km in our case. This would yield power-law slopes that are in better agreement with those derived from observations. We argue that since sea-ice deformation is a scale-invariant process at the geophysical scale, a sea-ice model should be able to represent this scaling down to the model grid cell. Figure 10 suggests that our BBM implementation allows SI3 to achieve this despite the use of the *Eulerian* framework.

The simulated and observed structure functions (*i.e.* the dependence of the scaling exponents of the power-law to the order of the moment) $\beta(q)$ are shown in Figure 11. The spatial scaling obtained from both the observations and SI3-BBM are multi-fractal, because their structure functions is well approximated (in the sense of the least square method) by a quadratic function of the type $\beta(q) = aq^2 + bq$. One should note that in the universal multi-fractal formalism, the structure functions are not required to be quadratic and can have a varying degree of non-linearity (Lovejoy and Schertzer, 2007). A quadratic structure function, as obtained here, simply means that the process of sea-ice deformation can be approximated by a log-normal multiplicative cascade model with a maximum degree of multi-fractality. The structure function of SI3-BBM shows a curvature $a$ that has a magnitude comparable to that of RGPS, *i.e.* 0.15 versus 0.17. These values of curvature are in fair agreement with those obtained from *Lagrangian* simulations performed with neXtSIM, and reported in previous studies: 0.14 in Rampal et al. (2016) and 0.11 in Rampal et al. (2019).

## 4 Discussion

### 4.1 On the numerical implementation

The cross-nudging bears a noteworthy analogy with the *Asselin* filter (Asselin, 1972) used when discretizing time derivatives of a prognostic variable by means of the *Leap Frog* scheme (three time-levels, centered, and second-order), in particular in the context of shallow-water equations. The goal of this *Asselin* filter is to subtly average the solutions of neighboring time levels to prevent the separation of trajectories between the even and odd time-step levels (Marsaleix et al., 2012). As such, the cross-nudging can be seen as a sort of spatial and two-dimensional analogue to the *Asselin* filter. Despite the crudeness of this approach, which tends to be problematic due to the unavoidable loss of conservation properties, the *Asselin* filter is still largely used in modern CMIP-class OGCMs like NEMO. Indeed, the ocean component of NEMO used in the simulations presented in this study still relies on it.

As of now, our cross-nudging approach clearly lacks physical and numerical consistency, but it somehow allows to demonstrate that the implementation of a brittle rheology, along with the advection of the internal stress tensor, is feasible onto an *E-augmented* C-grid, provided a method to prevent the separation of solutions is used. Nevertheless, we plan to further investigate the possibility to implement approaches that are more physically and numerically consistent. For instance, an option is to apply the cross-nudging on the two invariants of the stress tensor (*i.e.* $\sigma_I$ and $\sigma_{II}$) and the rate of internal work of the ice. This would introduce 3 equations for 3 invariant quantities, from which the 3 components of the stress tensor could be deduced afterward. Another option, is to explore the possibility of deriving a numerical formulation inspired from that of Mesinger (1973); Janjić (1974), in which auxiliary velocity (or stress) points are introduced midway between the neighboring tracer (or velocity) points.

Another critical requirement, this time stemming from the use of the *Eulerian* and finite-difference framework, has to do with the ability of the advection scheme to advect fields with as little numerical diffusion or dispersion as possible. This is particularly critical when using a brittle rheology like BBM, as most fields exhibit sharp gradients, often associated with linear kinematic features. We chose to use the scheme of Prather (1986), the dispersive scheme option of SI3, to favor the conservation of sharp gradients at the cost of potential noise and overshoots reminiscent of the *Gibbs* phenomenon. One could however consider the use of a different approach, that would optimize the advection of sharp gradients, for instance a spatial discretization based on the discontinuous *Galerkin* method. This method has proven to be efficient and accurate in treating the advection of sea-ice variables in the case of a brittle sea-ice rheology such as MEB (Dansereau et al., 2017), but has not yet been tested in the context of large scale, long-term sea-ice simulations. This is the scope of our present work and future papers.

As discussed in section 2.3.1, the use of the E-grid in the dynamics and advection modules of SI3 implies that equations specific to the momentum and the constitutive law are solved twice, on the T- and F-centric grids. Moreover, with the need to advect the stress tensor and the damage tracer, specific to brittle rheologies, 2×4 additional scalar fields need to be advected. This inevitably leads to an increase in the computational cost of SI3. We have estimated this extra cost by comparing the wall-time length required to complete a 90-day simulation with each rheology, using the same 29 cores in parallel, on the same computer. Our results, summarized in table 4, suggest that the increase in the computational cost associated with the use of BBM in place of aEVP is about 45% when SI3 is used in a standalone mode (SAS). In standard coupled mode, with SI3 coupled to OCE, the BBM-related cost increase is about 20%. This lower value is explained by the fact that by default, the coupling between OCE and SI3 is done sequentially. As such, the cost of SI3 simply adds up to that of OCE, and the cost of OCE is expected to be independent of the mode used (in our case: 113 and 114 cpu h for SI3-default and SI3-BBM, respectively). Based on our results, the relative cost of SI3 in coupled mode is about 40% when using the default aEVP setup and about 50% with our BBM implementation.

## 4.2 On the simulated sea-ice deformations

Based on comparisons against various types of observations, recent studies suggest that large-scale models using BBM can realistically simulate the dynamics and properties of sea-ice (Ólason et al., 2022; Rheinlænder et al., 2022; Boutin et al., 2023; Regan et al., 2023). Yet, the deformation in convergence, and the sub-grid-scale processes related to sea-ice ridging, are not represented by BBM with the same degree of accuracy. The model overestimates the number of converging events with

magnitudes of about 1 to 5% per day, and underestimates the most extreme events (Fig. 8.c, and Ólason et al., 2022). As of now, parameter tuning, in particular that of the BBM-specific ridging threshold parameter $P_{\text{max}}$, did not help to improve the agreement with the observed PDFs of convergence (not shown). Therefore, we conclude that some fundamental processes need to be reconsidered in BBM.

In section 3.3.3, we find that the degree of multi-fractality of the deformation fields simulated by SI3-BBM is slightly lower than that obtained from the RGPS data. The fact that the deformation fields simulated by neXtSIM in Ólason et al. (2022) are in better agreement with RGPS in this regard, suggests that this problem is linked to some numerical aspects of our BBM implementation rather than the BBM rheology itself. This is most likely the consequence of the introduction of some additional numerical dispersion and diffusion by the advection scheme and the cross-nudging treatment, respectively, as these two features are absent in neXtSIM. Moments of order two and three are expected to be more affected than the mean by an unwanted source of noise and diffusion, which might explain why SI3-BBM reproduces remarkably well the mean across all scales, and why the power-law exponents for the variance and the skewness are underestimated. In this regard, the use of the finite-element method together with the *Discontinuous Galerkin* method, might prove to be a promising combination to simulate even more accurately the multifractality of sea-ice deformation while remaining in the *Eulerian* and quadrilateral mesh framework.

## 5 Conclusions

The *Brittle Bingham Maxwell* rheology, known as BBM, has been successfully implemented into SI3, the CMIP-class, *Eulerian* finite-difference sea-ice model of the NEMO modeling system. We have shown that our implementation, which features a prognostic ice damage tracer and a prognostic stress tensor, is able to realistically simulate sea-ice deformation statistics on a pan-Arctic scale when compared to satellite observations.

Our implementation uses a new discretization approach that expands the C-grid of NEMO into the *Arakawa* E-grid in the parts of the code dedicated to sea-ice dynamics. We have chosen to do so in order to (i) avoid resorting to the spatial averaging of prognostic fields, in particular the damage tracer, as an interpolation technique between the center- and corner-points of the grid cells, and (ii) allow the straightforward advection of the shear component of the stress tensor. However, by solving the dynamics on the E-grid, the issue of the grid separation is introduced. We have introduced a simple technique to prevent this grid separation, in the form of a cross-nudging. This cross-nudging relies on the averaging of the components of the stress tensor, and as such, introduces a spatial smoothing of these components. Despite the fact that this aspect of our implementation is in contradiction with one of our initial motivation (*i.e.* avoid the use of spatial averaging), we think that our *E-augmented-C-grid* approach is promising. Because the damage tracer is never averaged, which we think is beneficial for the consistency of the brittle model, and the advection of the shear component of the stress tensor is straightforward and numerically consistent with that of the trace components.

For the advection of the stress tensor, we have chosen to use the upper-convected time derivative, rather than its lower-convected counterpart, a combination of the two, or simply the standard material derivative. This choice, based on arbitrary considerations, has no significant impact on the deformation statistics presented in this paper. Both formulations are available

in our implementation, which will allow SI3 users to further investigate on this matter, in particular by means of dedicated idealized test-cases.

     We carried out a statistical analysis of the sea-ice deformation rates obtained from a set of realistic pan-Arctic coupled ocean/sea-ice simulations of winter 1996-1997, performed with SI3 at a horizontal resolution of about 12-km. Based on a comparison with satellite observations, this analysis demonstrates that the use of the newly implemented BBM rheology results

in simulated sea-ice deformation statistics that are realistic. In particular, we show that the use of BBM allows to simulate highly-localized (nearly linear) kinematic features within the sea-ice cover, along which the most substantial deformation rates are concentrated.

The observed non-*Gaussian* statistics of the sea-ice deformation process are well present in the simulation that uses our newly-implemented BBM rheology, except the most extreme values and more particularly those corresponding to the convergent

mode of deformation. Since this drawback was already observed in the BBM-driven simulations of the *Lagrangian* sea-ice model neXtSIM presented in Ólason et al. (2022), we think that it probably shows an intrinsic limitation of the current BBM rheological model, an issue that certainly merits to be investigated and fixed in the future. Finally, we show that the observed spatial scaling invariance property of sea-ice deformation, and in particular its multi-fractal nature, is fairly well captured by the BBM-driven simulation but with a slightly lower degree of multifractality.

*Code and data availability.*

- The NEMO source code used to perform the experiments is based on the official release 4.2.2 of NEMO, it is available on Zenodo with DOI `10.5281/zenodo.10580759`: https://doi.org/10.5281/zenodo.10580759.

- New and modified Fortran-90 source files relative to our implementation of the BBM rheology in version 4.2.2 of NEMO/SI3 are available on Zenodo with DOI `10.5281/zenodo.10459449`: https://zenodo.org/records/10459449.

- The python software used to seed and build *Lagrangian* trajectories out of the SI3 hourly sea-ice velocities is named `sitrack`; the version used to perform the present study is available on Zenodo with DOI `10.5281/zenodo.10457918`: https://zenodo.org/records/10457918.

- The python software used to compute the RGPS and model-based sea-ice deformation rates based on quadrangles, and perform the scaling analysis is named `mojito`; the version used to perform the present study is available on Zenodo with DOI `10.5281/zenodo.10457924`:
https://zenodo.org/records/10457924.

- Model data produced and analyzed in this study, namely SI3 hourly output files for simulations SI3-BBM and SI3-aEVP, are available on Zenodo with DOI `10.5281/zenodo.10457955`: https://zenodo.org/records/10457955.

# Appendix A: Nomenclature

## A1 Table of symbols used in the text

| Symbol | Definition | Units |
|---|---|---|
| $m$ | mass of ice and snow per unit area | kg m$^{-2}$ |
| $\rho_i$ | density of sea-ice | kg m$^{-3}$ |
| $\rho_w$ | density of sea-water | kg m$^{-3}$ |
| $\vec{u} \equiv (u, v)$ | sea-ice velocity | m s$^{-1}$ |
| $A$ | sea-ice fraction | - |
| $h$ | sea-ice thickness | m |
| $g$ | acceleration of gravity | m s$^{-2}$ |
| $f$ | *Coriolis* frequency | s$^{-1}$ |
| $\vec{k}$ | vertical unit vector ($z$-axis) | s$^{-1}$ |
| $H$ | sea surface height | m |
| $\vec{\tau}$ | ice-ocean stress | Pa |
| $\vec{\tau}_a$ | wind (ice-atmosphere) stress | Pa |
| $\boldsymbol{\sigma}$ | internal stress tensor (2×2) | Pa |
| $\dot{\boldsymbol{\varepsilon}}$ | strain-rate tensor (2×2) | s$^{-1}$ |
| $d$ | damage of sea-ice | - |
| $\Delta x$ | local resolution (size) of the grid mesh | m |
| $C$ | compaction parameter | - |
| $\alpha$ | damage parameter (Dansereau, 2016) | - |
| $E_0, E$ | elastic modulus of undamaged & damaged ice | Pa |
| $\lambda_0, \lambda$ | apparent viscous relaxation time of undamaged & damaged ice | s |
| $\tilde{P}$ | BBM-specific ridging term | - |
| $P_{\text{max}}$ | ridging threshold | Pa |
| $P_0$ | scaling parameter for $P_{\text{max}}$ | Pa |
| $h_0$ | reference ice thickness for $P_{\text{max}}$ | m |
| $c$ | sea-ice cohesion | Pa |
| $\nu$ | Poisson's ratio | - |
| $\sigma_{\text{I}}$ | isotropic normal stress (first invariant of stress tensor) | Pa |
| $\sigma_{\text{II}}$ | maximum shear stress (second invariant of stress tensor) | Pa |
| $\mu$ | internal friction coefficient | - |
| $N$ | upper limit for compressive stress | Pa |
| $C_E$ | propagation speed of an elastic shear wave | m s$^{-1}$ |
| $t_d$ | characteristic time scale for the propagation of damage | s |
| $d_{crit}$ | damage increment (Dansereau, 2016) | - |
| $k_{th}$ | healing constant for damage | K s |
| $\Delta T_h$ | temperature difference between bottom & surface of ice | K |

## A2 Acronyms

| | |
|---|---|
| NEMO | Nucleus for European Modeling of the Ocean |
| SI3 | Sea-Ice modeling Integrated Initiative (sea-ice component of NEMO) |
| OCE | 3D ocean component of NEMO |
| SAS | Stand-Alone-Surface module of NEMO (*i.e.* SI3 standalone) |
| LIM | Louvain-La-Neuve sea-Ice Model |
| BBM | Brittle Bingham Maxwell (rheology) |
| MEB | Maxwell Elasto Brittle (rheology) |
| VP | Viscous-Plastic (rheology) |
| FD | Finite Difference (method) |
| CN | cross-nudging (treatment) |
| PDF | Probability Density Function |
| LKFs | Linear Kinematic Features |
| GCM | General Circulation Model |
| OGCM | Ocean General Circulation Model |
| SST | Sea Surface Temperature |
| SSH | Sea Surface Height |
| RGPS | RADARSAT Geophysical Processor System (dataset) |

## A3 Notations related to the discretization on the E-grid

The *bar + superscript* notation refers to a spatial interpolation; $\bar{\phi}^X$ is field $\phi$ interpolated onto $X$-points. Interpolation from T-
to F-point, or conversely, is the average of the four nearest surrounding points (Fig. 3.a):

$$\bar{\phi}^T_{i,j} = 1/4(\phi_{i,j} + \phi_{i-1,j} + \phi_{i-1,j-1} + \phi_{i,j-1}) \quad \text{(if } \phi \text{ defined @F)}$$
$$\bar{\phi}^F_{i,j} = 1/4(\phi_{i+1,j+1} + \phi_{i,j+1} + \phi_{i,j} + \phi_{i+1,j}) \quad \text{(if } \phi \text{ defined @T)}$$
(A1)

Note: surrounding points located on land or open-boundary cells are excluded from the averaging.

For the interpolation from tracer (T or F) to velocity (U or V) points , or conversely, only the two nearest surrounding points
are used:

$$\bar{\phi}^U_{i,j} = 1/2(\phi_{i+1,j} + \phi_{i,j}) \quad \text{(if } \phi \text{ defined @T)}$$
$$\bar{\phi}^V_{i,j} = 1/2(\phi_{i,j+1} + \phi_{i,j}) \quad \text{(if } \phi \text{ defined @T)}$$
$$\bar{\phi}^U_{i,j} = 1/2(\phi_{i,j} + \phi_{i,j-1}) \quad \text{(if } \phi \text{ defined @F)}$$
$$\bar{\phi}^V_{i,j} = 1/2(\phi_{i,j} + \phi_{i-1,j}) \quad \text{(if } \phi \text{ defined @F)}$$
(A2)

The *hat* notation $\hat{x}$ refers to the F-centric counterpart of $x$, $x$ being a prognostic scalar or tensor (rank 1 or 2) defined in
the T-centric grid (mind that if $x$ is the element of a tensor, $\hat{x}$ is not necessarily defined on F-points). Example: $\hat{d}$ and $\hat{\sigma}_{11}$ are

prognostic fields defined on F-points (natural location for $d$ and $\sigma_{11}$ on the C-grid is the T-point); similarly, $\hat{\sigma}_{12}$ is defined on T-points (natural location for $\sigma_{12}$ on the C-grid is the F-point).

## 695 A4   Miscellaneous notations

| | |
|---|---|
| $\underline{\boldsymbol{x}}$ | symmetric tensor $\boldsymbol{x}$ expressed in its *Voigt* form |
| $x^{(i)}$ | initial estimate of variable $x$ |
| @X | on the $X$-points of the grid |
| $\boldsymbol{\sigma_{kl}}$ | vertically-integrated components of tensor $\boldsymbol{\sigma}$ |
| | $\rightarrow \boldsymbol{\sigma_{kl}} \equiv h\,\sigma_{kl}$ if $\sigma_{kl}$ defined @T |
| | $\rightarrow \boldsymbol{\sigma_{kl}} \equiv \bar{h}^{F}\sigma_{kl}$ if $\sigma_{kl}$ defined @F |
| $\overset{\triangledown}{\boldsymbol{x}}$ | upper-convected time derivative of symmetric (rank 2) tensor $\boldsymbol{x}$ |

## A5 Table of symbols related to the numerical implementation

| Symbol | Definition | Units |
|---|---|---|
| $\Delta T$ | *advective* time-step for the advection and the thermodynamics | s |
| $\Delta t$ | *dynamical* time-step specific to BBM (time-splitting) | s |
| $N_s$ | $\equiv \Delta T/\Delta t$, time-splitting parameter | - |
| $k$ | time-level index of time splitting ($1 \leq k \leq N_s$) | - |
| $A, h, d$ | ice concentration, thickness and damage of ice @T | -, m, - |
| $\bar{A}^F, \bar{h}^F$ | ice concentration and thickness interpolated @F | -, m |
| $\hat{d}$ | damage of ice @F | - |
| $\dot{\boldsymbol{\varepsilon}} \equiv (\dot{\varepsilon}_{11}, \dot{\varepsilon}_{22}, \dot{\varepsilon}_{12})$ | strain-rate tensor (2×2) of the T-centric cell | s$^{-1}$ |
| $\hat{\dot{\boldsymbol{\varepsilon}}} \equiv (\hat{\dot{\varepsilon}}_{11}, \hat{\dot{\varepsilon}}_{22}, \hat{\dot{\varepsilon}}_{12})$ | strain-rate tensor (2×2) of the F-centric cell | s$^{-1}$ |
| $\boldsymbol{\sigma} \equiv (\sigma_{11}, \sigma_{22}, \sigma_{12})$ | internal stress tensor (2×2) of the T-centric cell | Pa |
| $\hat{\boldsymbol{\sigma}} \equiv (\hat{\sigma}_{11}, \hat{\sigma}_{22}, \hat{\sigma}_{12})$ | internal stress tensor (2×2) of the F-centric cell | Pa |
| $\bar{A}^U, \bar{A}^V$ | ice concentration interpolated @U and @V | m |
| $\bar{h}^U, \bar{h}^V$ | ice thickness interpolated @U and @V | m |
| $u, v$ | ice velocity at the $\Delta t$ level (@U and @V) | m s$^{-1}$ |
| $\hat{u}, \hat{v}$ | ice velocity at the $\Delta t$ level (@V and @U) | m s$^{-1}$ |
| $U, V$ | ice velocity at the $\Delta T$ level (@U and @V) | m s$^{-1}$ |
| $\hat{U}, \hat{V}$ | ice velocity at the $\Delta T$ level (@V and @U) | m s$^{-1}$ |
| $C_D^{(o)}$ | ice-ocean drag coefficient | - |
| $\tau_x, \tau_y$ | ice-ocean stress @U and @V | Pa |
| $\hat{\tau}_x, \hat{\tau}_y$ | ice-ocean stress @V and @U | Pa |
| $u_o, v_o$ | surface ocean current @U and @V | m s$^{-1}$ |
| $\bar{u}_o, \bar{v}_o$ | surface ocean current interpolated @V and @U | m s$^{-1}$ |
| $\gamma_{cn}$ | cross-nudging coefficient | - |
| $C_D^{(a)}$ | ice-atmosphere drag coefficient | - |
| $\Delta^{\scriptscriptstyle T}\!x$ | T-centered $\Delta x$ that connects 2 neighboring U-points | m |
| $\Delta^{\scriptscriptstyle T}\!y$ | T-centered $\Delta y$ that connects 2 neighboring V-points | m |
| $\Delta^{\scriptscriptstyle F}\!x$ | F-centered $\Delta x$ that connects 2 neighboring V-points | m |
| $\Delta^{\scriptscriptstyle F}\!y$ | F-centered $\Delta y$ that connects 2 neighboring U-points | m |
| $\Delta^{\scriptscriptstyle U}\!x$ | U-centered $\Delta x$ that connects 2 neighboring T-points | m |
| $\Delta^{\scriptscriptstyle U}\!y$ | U-centered $\Delta y$ that connects 2 neighboring F-points | m |
| $\Delta^{\scriptscriptstyle V}\!x$ | V-centered $\Delta x$ that connects 2 neighboring F-points | m |
| $\Delta^{\scriptscriptstyle V}\!y$ | V-centered $\Delta y$ that connects 2 neighboring T-points | m |

## Appendix B: Algorithm and discretization

 ## B1   Algorithm

**Time-splitting loop** $(\Delta t)$ / for $k = 1$ to $N_s$:

- compute elasticity $E, \hat{E}$ and viscous relaxation time $\lambda, \hat{\lambda}$ as a function of damage $d^k, \hat{d}^k$ and current sea-ice concentration $A, \bar{A}^F$ (Eq. B5, B6)

- compute the normal stress invariant of $\boldsymbol{\sigma}^k$ and $\hat{\boldsymbol{\sigma}}^k \rightarrow \sigma_{\mathsf{I}}^k, \hat{\sigma}_{\mathsf{I}}^k$ (Eq. B11)

 - compute $P_{\max}, \hat{P}_{\max}$ as a function of current sea-ice thickness $h, \bar{h}^F$ and concentration $A, \bar{A}^F$ (Eq. B7)

- compute $\tilde{P}, \hat{\tilde{P}}$ as a function of $P_{\max}, \hat{P}_{\max}$ and $\sigma_{\mathsf{I}}, \hat{\sigma}_{\mathsf{I}}$ (Eq. B8)

- compute the 3 components of each strain-rate tensor $\dot{\boldsymbol{\varepsilon}}, \hat{\dot{\boldsymbol{\varepsilon}}}$ based on sea-ice velocities at time-level $k$ (Eq. B1, B2, B3 & B4)

- initial prognostic estimate of the stress tensors at time-level $k+1 \rightarrow \boldsymbol{\sigma}^{(\mathsf{i})k+1}$ and $\hat{\boldsymbol{\sigma}}^{(\mathsf{i})k+1}$ (Eq. B10)

- apply cross-nudging between $\boldsymbol{\sigma}^{(\mathsf{i})k+1}$ and $\hat{\boldsymbol{\sigma}}^{(\mathsf{i})k+1}$ (Eq. 16):

 - *Mohr-Coulomb* test on $\boldsymbol{\sigma}^{(\mathsf{i})k+1}$ and $\hat{\boldsymbol{\sigma}}^{(\mathsf{i})k+1}$
  - ⋆ compute the 2 invariants of $\boldsymbol{\sigma}^{(\mathsf{i})k+1}$ and $\hat{\boldsymbol{\sigma}}^{(\mathsf{i})k+1} \rightarrow \sigma_{\mathsf{I}}^{(\mathsf{i})k+1}, \sigma_{\mathsf{II}}^{(\mathsf{i})k+1}$ and $\hat{\sigma}_{\mathsf{I}}^{(\mathsf{i})k+1}, \hat{\sigma}_{\mathsf{II}}^{(\mathsf{i})k+1}$ (Eq. B11)
  - ⋆ compute $d_{crit}$ and $\hat{d}_{crit}$ based on $\sigma_{\mathsf{I}}^{(\mathsf{i})k+1}, \sigma_{\mathsf{II}}^{(\mathsf{i})k+1}$ and $\hat{\sigma}_{\mathsf{I}}^{(\mathsf{i})k+1}, \hat{\sigma}_{\mathsf{II}}^{(\mathsf{i})k+1}$ (Eq. B12)

- prognostic estimate of the stress tensors and damage at time-level $k+1 \rightarrow \boldsymbol{\sigma}^{k+1}, d^{k+1}$ and $\hat{\boldsymbol{\sigma}}^{k+1}, \hat{d}^{k+1}$
  - ⋆ where $0 < d_{crit} < 1$ and/or $0 < \hat{d}_{crit} < 1$ (overcritical stress state):

     → damage growth and stress adjustment (Eq. B13)

  - ⋆ elsewhere:

    → no damage growth and no stress adjustment (Eq. B14)

- compute the divergence of the vertically-integrated $\boldsymbol{\sigma}^{k+1}$ and $\hat{\boldsymbol{\sigma}}^{k+1}$ (Eq. B16 & B17)

- prognostic estimate of sea-ice velocity at time-level $k+1 \rightarrow u^{k+1}, v^{k+1}$ and $\hat{u}^{k+1}, \hat{v}^{k+1}$ (Eq. B19 & B18)

 **NEMO (advective) time-step** $(\Delta T)$:

- BBM rheology (*time-splitting loop* above)

- advection of generic SI3 prognostic tracers ($A, h$, etc) at T-points using $U, V$

- advection of $d, \sigma_{11}, \sigma_{22}$ and $\hat{\sigma}_{12}$ at T-points using $U, V$

- advection of $\hat{d}, \hat{\sigma}_{11}, \hat{\sigma}_{22}$ and $\sigma_{12}$ at F-points using $\hat{U}, \hat{V}$

 - healing of damage ($d$ and $\hat{d}$) (Eq.14)

- thermodynamics module of SI3 (update of $A, h$, etc)

## B2 Update of internal stress tensor in the T- and F-centric worlds

### B2.1 Divergence, shear and strain-rate tensor of ice velocity

Following Hunke and Dukowicz (2002), here is how the components of the strain rate of the sea-ice velocity vector are computed on the T- and F-centric grids, based on the finite-difference method.

⋄ Divergence rate $(\partial_x u + \partial_y v)$:

$$D_{i,j} = \frac{[\Delta^{\mathrm{U}}y\, u]_{i,j} - [\Delta^{\mathrm{U}}y\, u]_{i-1,j} + [\Delta^{\mathrm{V}}x\, v]_{i,j} - [\Delta^{\mathrm{V}}x\, v]_{i,j-1}}{[\Delta^{\mathrm{T}}x\, \Delta y]_{i,j}},$$

$$\hat{D}_{i,j} = \frac{[\Delta^{\mathrm{V}}y\, \hat{u}]_{i+1,j} - [\Delta^{\mathrm{V}}y\, \hat{u}]_{i,j} + [\Delta^{\mathrm{U}}x\, \hat{v}]_{i,j+1} - [\Delta^{\mathrm{U}}x\, \hat{v}]_{i,j}}{[\Delta^{\mathrm{F}}x\, \Delta y]_{i,j}}.$$

(B1)

⋄ Tension rate $(\partial_x u - \partial_y v)$:

$$T_{i,j} = \frac{\left([u/\Delta^{\mathrm{U}}y]_{i,j} - [u/\Delta^{\mathrm{U}}y]_{i-1,j}\right)[\Delta y^2]_{i,j} - \left([v/\Delta^{\mathrm{V}}x]_{i,j} - [v/\Delta^{\mathrm{V}}x]_{i,j-1}\right)[\Delta^{\mathrm{T}}x^2]_{i,j}}{[\Delta^{\mathrm{T}}x\, \Delta y]_{i,j}},$$

$$\hat{T}_{i,j} = \frac{\left([\hat{u}/\Delta^{\mathrm{V}}y]_{i+1,j} - [\hat{u}/\Delta^{\mathrm{V}}y]_{i,j}\right)[\Delta y^2]_{i,j} - \left([\hat{v}/\Delta^{\mathrm{U}}x]_{i,j+1} - [\hat{v}/\Delta^{\mathrm{U}}x]_{i,j}\right)[\Delta^{\mathrm{F}}x^2]_{i,j}}{[\Delta^{\mathrm{F}}x\, \Delta y]_{i,j}}.$$

(B2)

⋄ Shear rate $(\partial_y u + \partial_x v)$:

$$S_{i,j} = \frac{\left([u/\Delta^{\mathrm{U}}x]_{i,j+1} - [u/\Delta^{\mathrm{U}}x]_{i,j}\right)[\Delta^{\mathrm{F}}x^2]_{i,j} + \left([v/\Delta^{\mathrm{V}}y]_{i+1,j} - [v/\Delta^{\mathrm{V}}y]_{i,j}\right)[\Delta^{\mathrm{F}}y^2]_{i,j}}{[\Delta^{\mathrm{F}}x\, \Delta y]_{i,j}},$$

$$\hat{S}_{i,j} = \frac{\left([\hat{u}/\Delta^{\mathrm{V}}x]_{i,j} - [\hat{u}/\Delta^{\mathrm{V}}x]_{i,j-1}\right)[\Delta^{\mathrm{T}}x^2]_{i,j} + \left([\hat{v}/\Delta^{\mathrm{U}}y]_{i,j} - [\hat{v}/\Delta^{\mathrm{U}}y]_{i-1,j}\right)[\Delta^{\mathrm{T}}y^2]_{i,j}}{[\Delta^{\mathrm{T}}x\, \Delta y]_{i,j}}.$$

(B3)

From which the 3 components of the 2D strain-rate tensors are obtained:

$$\begin{pmatrix} \dot{\varepsilon}_{11} \\ \dot{\varepsilon}_{22} \\ \hat{\dot{\varepsilon}}_{12} \end{pmatrix}_{i,j} = \frac{1}{2}\begin{pmatrix} D_{i,j} + T_{i,j} \\ D_{i,j} - T_{i,j} \\ \hat{S}_{i,j} \end{pmatrix}$$

$$\begin{pmatrix} \hat{\dot{\varepsilon}}_{11} \\ \hat{\dot{\varepsilon}}_{22} \\ \dot{\varepsilon}_{12} \end{pmatrix}_{i,j} = \frac{1}{2}\begin{pmatrix} \hat{D}_{i,j} + \hat{T}_{i,j} \\ \hat{D}_{i,j} - \hat{T}_{i,j} \\ S_{i,j} \end{pmatrix}$$

(B4)

### B2.2 Update of the stress tensors

⋄ Elasticity and viscous relaxation time of damaged ice:

$$E = E_0(1-d)\, e^{C(1-A)}$$
$$\hat{E} = E_0(1-\hat{d})\, e^{C(1-\bar{A}^F)}$$

(B5)

$$\lambda = \lambda 0 \left[ (1-d)\, e^{C(1-A)} \right]^{\alpha-1}$$
$$\hat{\lambda} = \lambda 0 \left[ (1-\hat{d})\, e^{C(1-\bar{A}^F)} \right]^{\alpha-1}$$

(B6)

Note that it is the averaged value of $A$ at F-points, $\bar{A}^F$, that is used in the equations for the F-centric grid.

⋄ Ridging threshold:

$$P_{\text{max}} = P0 \left[ h/h_0 \right]^{3/2} e^{C(1-A)}$$
$$\hat{P}_{\text{max}} = P0 \left[ \bar{h}^F /h_0 \right]^{3/2} e^{C(1-\bar{A}^F)}$$

(B7)

Note that it is the averaged value of $h$ at F-points, $\bar{h}^F$, that is used in the second equation.

⋄ $\tilde{P}$ term:

$$\tilde{P} = \begin{cases} \dfrac{\sigma_{\text{I}}}{-P_{\text{max}}} & \text{for } \sigma_{\text{I}} < -P_{\text{max}} \\ -1 & \text{for } -P_{\text{max}} \leq \sigma_{\text{I}} < 0 \\ 0 & \text{for } \sigma_{\text{I}} > 0 \end{cases}$$

$$\hat{\tilde{P}} = \begin{cases} \dfrac{\hat{\sigma}_{\text{I}}}{-\hat{P}_{\text{max}}} & \text{for } \hat{\sigma}_{\text{I}} < -\hat{P}_{\text{max}} \\ -1 & \text{for } -\hat{P}_{\text{max}} \leq \hat{\sigma}_{\text{I}} < 0 \\ 0 & \text{for } \hat{\sigma}_{\text{I}} > 0 \end{cases}$$

(B8)

⋄ Multiplicator for stress update:

$$\Omega = \frac{\lambda}{\lambda + (1+\tilde{P})\Delta t}$$
$$\hat{\Omega} = \frac{\hat{\lambda}}{\hat{\lambda} + (1+\hat{\tilde{P}})\Delta t}$$

(B9)

◇ Initial update of stress tensor:

$$\sigma_{11}^{(i)k+1} = \Omega \left[ E \, \Delta t \, \frac{1}{1-\nu^2} \, (\dot{\varepsilon}_{11}^k + \nu \, \dot{\varepsilon}_{22}^k) + \sigma_{11}^k \right]$$

$$\sigma_{22}^{(i)k+1} = \Omega \left[ E \, \Delta t \, \frac{1}{1-\nu^2} \, (\nu \, \dot{\varepsilon}_{11}^k + \dot{\varepsilon}_{22}^k) + \sigma_{22}^k \right]$$

$$\hat{\sigma}_{12}^{(i)k+1} = \hat{\Omega} \left[ \hat{E} \, \Delta t \, \frac{1-\nu}{1-\nu^2} \, \hat{\dot{\varepsilon}}_{12}^k + \hat{\sigma}_{12}^k \right]$$

$$\hat{\sigma}_{11}^{(i)k+1} = \hat{\Omega} \left[ \hat{E} \, \Delta t \, \frac{1}{1-\nu^2} \, (\hat{\dot{\varepsilon}}_{11}^k + \nu \, \hat{\dot{\varepsilon}}_{22}^k) + \hat{\sigma}_{11}^k \right]$$

$$\hat{\sigma}_{22}^{(i)k+1} = \hat{\Omega} \left[ \hat{E} \, \Delta t \, \frac{1}{1-\nu^2} \, (\nu \, \hat{\dot{\varepsilon}}_{11}^k + \hat{\dot{\varepsilon}}_{22}^k) + \hat{\sigma}_{22}^k \right]$$

$$\sigma_{12}^{(i)k+1} = \Omega \left[ E \, \Delta t \, \frac{1-\nu}{1-\nu^2} \, \dot{\varepsilon}_{12}^k + \sigma_{12}^k \right]$$

(B10)

◇ Invariants of stress tensor:

$$\sigma_{\mathrm{I}} = \frac{1}{2}(\sigma_{11} + \sigma_{22}), \quad \sigma_{\mathrm{II}} = \sqrt{\left(\frac{\sigma_{11} - \sigma_{22}}{2}\right)^2 + \hat{\sigma}_{12}^2}$$

$$\hat{\sigma}_{\mathrm{I}} = \frac{1}{2}(\hat{\sigma}_{11} + \hat{\sigma}_{22}), \quad \hat{\sigma}_{\mathrm{II}} = \sqrt{\left(\frac{\hat{\sigma}_{11} - \hat{\sigma}_{22}}{2}\right)^2 + \sigma_{12}^2}$$

(B11)

◇ Damage increment:

$$d_{crit} = \begin{cases} \dfrac{c}{\sigma_{\mathrm{II}}^{(i)} + \mu\,\sigma_{\mathrm{I}}^{(i)}} & \text{if } \sigma_{\mathrm{I}}^{(i)} > -N \\[2em] \dfrac{-N}{\sigma_{\mathrm{I}}^{(i)}} & \text{otherwise} \end{cases}$$

$$\hat{d}_{crit} = \begin{cases} \dfrac{c}{\hat{\sigma}_{\mathrm{II}}^{(i)} + \mu\,\hat{\sigma}_{\mathrm{I}}^{(i)}} & \text{if } \hat{\sigma}_{\mathrm{I}}^{(i)} > -N \\[2em] \dfrac{-N}{\hat{\sigma}_{\mathrm{I}}^{(i)}} & \text{otherwise} \end{cases}$$

(B12)

◇ Update of damage and stress tensors:
    ⋆ in regions where $0 < d_{crit} < 1$:

$$d^{k+1} = d^k + (1-d_{crit})(1-d^k) \, \Delta t/t_d \qquad \text{with } t_d = \Delta^{\mathrm{T}} x \sqrt{\frac{2(1+\nu)\rho_i}{E}}$$

$$\underline{\underline{\sigma}}^{k+1} = \underline{\underline{\sigma}}^{(i)k+1} - (1-d_{crit}) \, \underline{\underline{\sigma}}^{(i)k+1} \, \Delta t/t_d$$

$$\hat{d}^{k+1} = \hat{d}^k + (1-\hat{d}_{crit})(1-\hat{d}^k) \, \Delta t/\hat{t}_d \qquad \text{with } \hat{t}_d = \Delta^{\mathrm{F}} x \sqrt{\frac{2(1+\nu)\rho_i}{\hat{E}}}$$

$$\underline{\underline{\hat{\sigma}}}^{k+1} = \underline{\underline{\hat{\sigma}}}^{(i)k+1} - (1-\hat{d}_{crit}) \, \underline{\underline{\hat{\sigma}}}^{(i)k+1} \, \Delta t/\hat{t}_d$$

(B13)

⋆ elsewhere:

$$d^{k+1} = d^k$$
$$\underline{\sigma}^{k+1} = \underline{\sigma}^{(i)k+1}$$

$$\hat{d}^{k+1} = \hat{d}^k$$
$$\underline{\hat{\sigma}}^{k+1} = \underline{\hat{\sigma}}^{(i)k+1}$$

(B14)

## B3 Momentum equation

As opposed to aEVP for which SI3 uses the scheme of Kimmritz et al. (2016, 2017), here we chose to solve the equation for momentum (in both the T- and F-centric worlds) using the implicit scheme approach of Bouillon et al. (2009).

### B3.1 Divergence of the vertically-integrated stress tensor

◇ Definition:

$$\begin{pmatrix} \Delta_x \\ \Delta_y \end{pmatrix} \equiv \begin{pmatrix} \dfrac{\partial(h\,\sigma_{11})}{\partial x} + \dfrac{\partial(h\,\sigma_{12})}{\partial y} \\ \dfrac{\partial(h\,\sigma_{22})}{\partial y} + \dfrac{\partial(h\,\sigma_{12})}{\partial x} \end{pmatrix}$$

(B15)

◇ Discretized in the T-centric cell:

$$\Delta_x^{k+1}\big|_{i,j} = \frac{\left[\sigma_{11}^{k+1} h\,\Delta^{\scriptscriptstyle\mathrm{T}}\!y^2\right]_{i+1,j} - \left[\sigma_{11}^{k+1}\,h\,\Delta^{\scriptscriptstyle\mathrm{T}}\!y^2\right]_{i,j}}{\left[\Delta^{\scriptscriptstyle\mathrm{U}}\!x\,\Delta^{\scriptscriptstyle\mathrm{U}}\!y^2\right]_{i,j}} + \frac{\left[\sigma_{12}^{k+1}\,\bar{h}^{\scriptscriptstyle F}\,\Delta^{\scriptscriptstyle F}\!x^2\right]_{i,j} - \left[\sigma_{12}^{k+1}\,\bar{h}^{\scriptscriptstyle F}\,\Delta^{\scriptscriptstyle F}\!x^2\right]_{i,j-1}}{\left[\Delta^{\scriptscriptstyle\mathrm{U}}\!y\,\Delta^{\scriptscriptstyle\mathrm{U}}\!x^2\right]_{i,j}} \quad (@U)$$

(B16)

$$\Delta_y^{k+1}\big|_{i,j} = \frac{\left[\sigma_{22}^{k+1} h\,\Delta^{\scriptscriptstyle\mathrm{T}}\!x^2\right]_{i,j+1} - \left[\sigma_{22}^{k+1}\,h\,\Delta^{\scriptscriptstyle\mathrm{T}}\!x^2\right]_{i,j}}{\left[\Delta^{\scriptscriptstyle\mathrm{V}}\!y\,\Delta^{\scriptscriptstyle\mathrm{V}}\!x^2\right]_{i,j}} + \frac{\left[\sigma_{12}^{k+1}\hat{h}\,\Delta^{\scriptscriptstyle F}\!y^2\right]_{i,j} - \left[\sigma_{12}^{k+1}\,\hat{h}\,\Delta^{\scriptscriptstyle F}\!y^2\right]_{i-1,j}}{\left[\Delta^{\scriptscriptstyle\mathrm{V}}\!x\,\Delta^{\scriptscriptstyle\mathrm{V}}\!y^2\right]_{i,j}} \quad (@V)$$

◇ Discretized in the F-centric cell:

$$\hat{\Delta}_x^{k+1}\big|_{i,j} = \frac{\left[\hat{\sigma}_{11}^{k+1}\,\bar{h}^{\scriptscriptstyle F}\,\Delta^{\scriptscriptstyle F}\!y^2\right]_{i,j} - \left[\hat{\sigma}_{11}^{k+1}\,\bar{h}^{\scriptscriptstyle F}\,\Delta^{\scriptscriptstyle F}\!y^2\right]_{i-1,j}}{\left[\Delta^{\scriptscriptstyle\mathrm{V}}\!x\,\Delta^{\scriptscriptstyle\mathrm{V}}\!y^2\right]_{i,j}} + \frac{\left[\hat{\sigma}_{12}^{k+1} h\,\Delta^{\scriptscriptstyle\mathrm{T}}\!x^2\right]_{i,j+1} - \left[\hat{\sigma}_{12}^{k+1}\,h\,\Delta^{\scriptscriptstyle\mathrm{T}}\!x^2\right]_{i,j}}{\left[\Delta^{\scriptscriptstyle\mathrm{V}}\!y\,\Delta^{\scriptscriptstyle\mathrm{V}}\!x^2\right]_{i,j}} \quad (@V)$$

(B17)

$$\hat{\Delta}_y^{k+1}\big|_{i,j} = \frac{\left[\hat{\sigma}_{22}^{k+1}\,\bar{h}^{\scriptscriptstyle F}\,\Delta^{\scriptscriptstyle F}\!x^2\right]_{i,j} - \left[\hat{\sigma}_{22}^{k+1}\,\bar{h}^{\scriptscriptstyle F}\,\Delta^{\scriptscriptstyle F}\!x^2\right]_{i,j-1}}{\left[\Delta^{\scriptscriptstyle\mathrm{U}}\!y\,\Delta^{\scriptscriptstyle\mathrm{U}}\!x^2\right]_{i,j}} + \frac{\left[\hat{\sigma}_{12}^{k+1} h\,\Delta^{\scriptscriptstyle\mathrm{T}}\!y^2\right]_{i+1,j} - \left[\hat{\sigma}_{12}^{k+1}\,h\,\Delta^{\scriptscriptstyle\mathrm{T}}\!y^2\right]_{i,j}}{\left[\Delta^{\scriptscriptstyle\mathrm{U}}\!x\,\Delta^{\scriptscriptstyle\mathrm{U}}\!y^2\right]_{i,j}} \quad (@U)$$

### B3.2 Update of sea-ice velocity

For clarity, we gather the contributions of the wind stress, the *Coriolis* term, and the SSH tilt vectors in a single vector term named $(R_x, R_y)$. Because these 3 terms do not present any particular challenge to express with respect to the existing imple-

mentation of aEVP. Note, however, that with the E-grid no spatial interpolation is needed to express the discretized *Coriolis* term.

Implicitness of the scheme is introduced through the use of sea-ice velocity at level $k+1$ in the estimate of the basal ice-water stress vector $(\tau_x, \tau_y)$:

$$
\begin{aligned}
\tau_x &= Z_x^k \left( u_o^k - u^{k+1} \right) \quad &\text{with:} \quad Z_x^k &= \bar{A}^U \, \rho_w \, C_D^{(o)} \sqrt{\left( u_o^k - u^k \right)^2 + \left( \bar{v}_o^{Uk} - \hat{v}^k \right)^2} \quad &\text{(@U)} \\
\tau_y &= Z_y^k \left( v_o^k - v^{k+1} \right) \quad &\text{with:} \quad Z_y^k &= \bar{A}^V \, \rho_w \, C_D^{(o)} \sqrt{\left( \bar{u}_o^{Vk} - \hat{u}^k \right)^2 + \left( v_o^k - v^k \right)^2} \quad &\text{(@V)} \\
\hat{\tau}_x &= \hat{Z}_x^k \left( \bar{u}_o^{Vk} - \hat{u}^{k+1} \right) \quad &\text{with:} \quad \hat{Z}_x^k &= \bar{A}^V \, \rho_w \, C_D^{(o)} \sqrt{\left( \bar{u}_o^{Vk} - \hat{u}^k \right)^2 + \left( v_o^k - v^k \right)^2} \quad &\text{(@V)} \\
\hat{\tau}_y &= \hat{Z}_y^k \left( \bar{v}_o^{Uk} - \hat{v}^{k+1} \right) \quad &\text{with:} \quad \hat{Z}_y^k &= \bar{A}^U \, \rho_w \, C_D^{(o)} \sqrt{\left( u_o^k - u^k \right)^2 + \left( \bar{v}_o^{Uk} - \hat{v}^k \right)^2} \quad &\text{(@U)}
\end{aligned}
\tag{B18}
$$

Then, the discretized equation of momentum yields the expression of the 2 velocity components at time-level $k+1$:

$$
\begin{aligned}
u^{k+1} &= \frac{\frac{\rho_i \, \bar{h}^U}{\Delta t} \, u^k + Z_x \, u_o^k + \Delta_x^{k+1} + R_x^k}{\frac{\rho_i \, \bar{h}^U}{\Delta t} + Z_x} \quad &\text{(@U)} \\[2em]
v^{k+1} &= \frac{\frac{\rho_i \, \bar{h}^V}{\Delta t} \, v^k + Z_y \, v_o^k + \Delta_y^{k+1} + R_y^k}{\frac{\rho_i \, \bar{h}^V}{\Delta t} + Z_y} \quad &\text{(@V)} \\[2em]
\hat{u}^{k+1} &= \frac{\frac{\rho_i \, \bar{h}^V}{\Delta t} \, \hat{u}^k + \hat{Z}_x \, \bar{u}_o^{Vk} + \hat{\Delta}_x^{k+1} + \hat{R}_x^k}{\frac{\rho_i \, \bar{h}^V}{\Delta t} + \hat{Z}_x} \quad &\text{(@V)} \\[2em]
\hat{v}^{k+1} &= \frac{\frac{\rho_i \, \bar{h}^U}{\Delta t} \, \hat{v}^k + \hat{Z}_y \, \bar{v}_o^{Uk} + \hat{\Delta}_y^{k+1} + \hat{R}_y^k}{\frac{\rho_i \, \bar{h}^U}{\Delta t} + \hat{Z}_y} \quad &\text{(@U)}
\end{aligned}
\tag{B19}
$$

*Author contributions.* This study is based on the original idea of PR and EO, who suggested using the E-grid. The implementation of the BBM rheology into SI3 has been carried out by LB, with the help of PR. The statistical analyses have been performed by LB. LB also prepared all the figures. Interpretations of the results were provided by LB, PR and EO. LB and PR led the writing of the manuscript, incorporating input from EO. Suggestions and improvements on the text have also been provided by VD.

*Competing interests.* The authors declare that the research was conducted in the absence of any commercial or financial relationships that could be construed as a potential conflict of interest

*Acknowledgements.* This study has been financially supported by the Schmidt Futures foundation (grant number 353) through the SASIP international project. Schmidt Futures is a philanthropic initiative founded by Eric and Wendy Schmidt that bets early on exceptional people making the world better, particularly through innovative breakthroughs in science and technology.

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

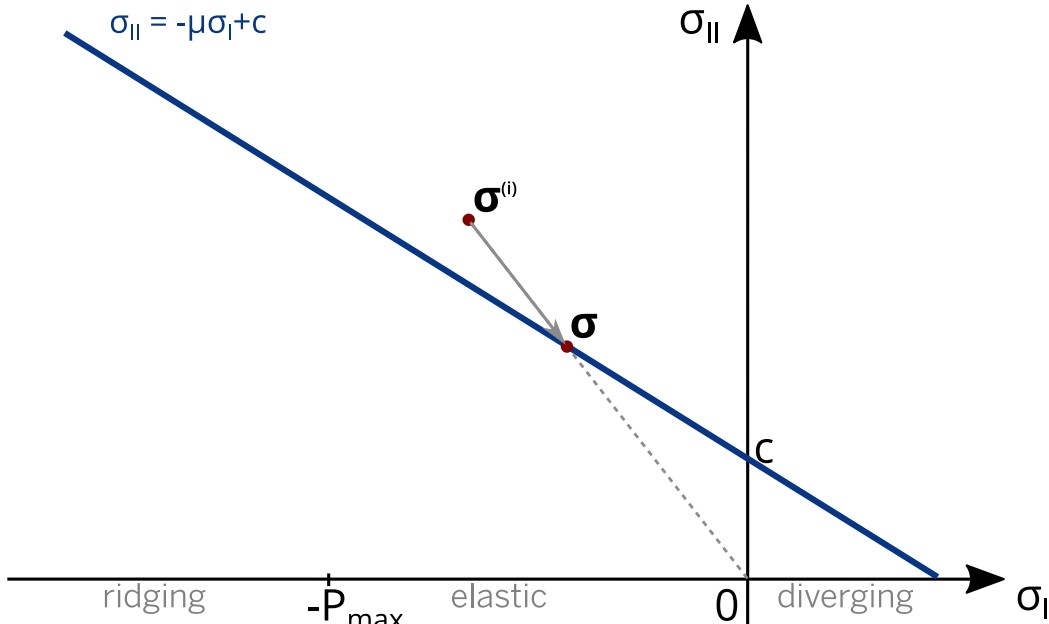

**Figure 1.** *Mohr-Coulomb* yield envelope in the internal stress invariant coordinates (blue line). Illustration of how an over-critical stress state $\sigma^{(i)}$ (initial estimate) is evolving (gray arrow) towards the corrected state $\sigma$ when using the BBM rheology.

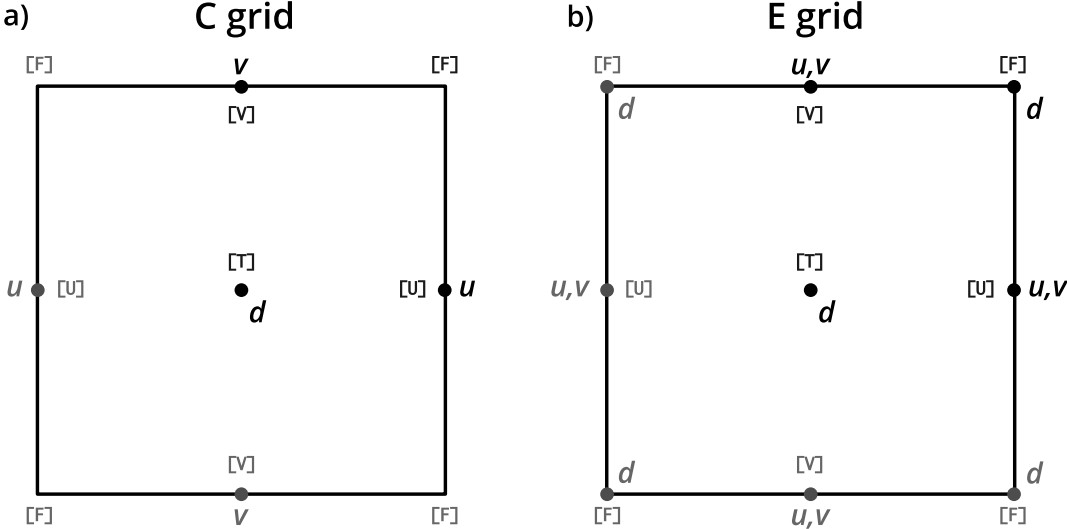

**Figure 2.** Point arrangement and staggering in a grid cell: (a) the C-grid as used in NEMO, and (b) the E-grid. The letter $d$ indicates the location of tracers, while letters $u$ and $v$ that of the $i$- and $j$-wise components of the velocity vector. Letters in brackets indicate the name of the grid points as referred to throughout the paper.

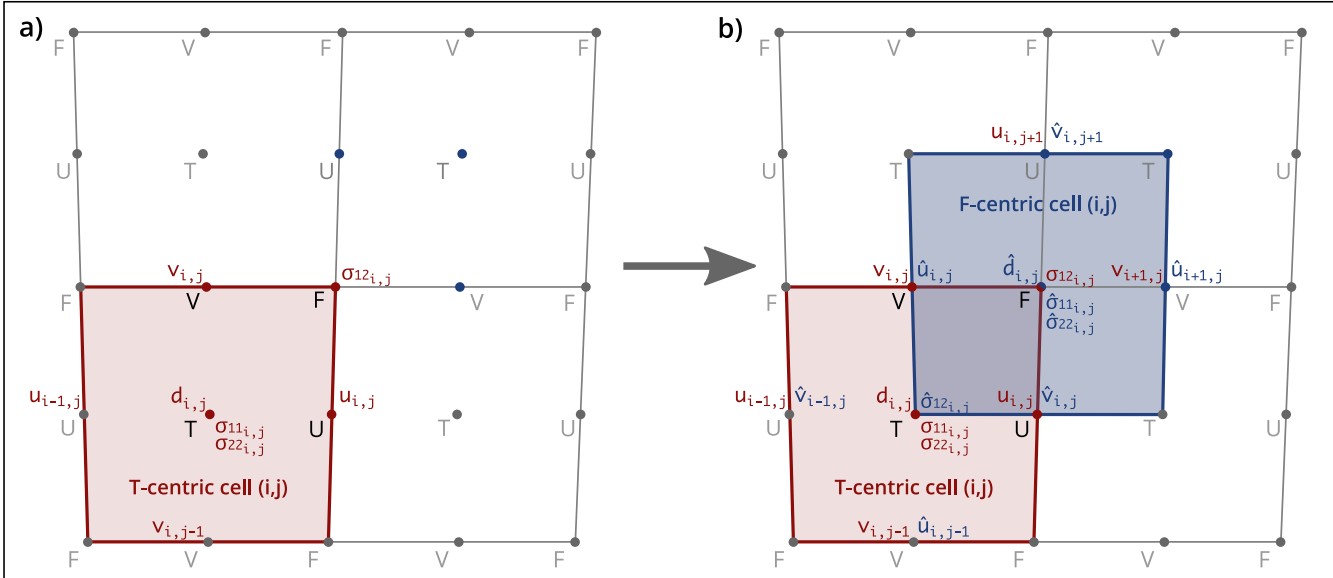

**Figure 3.** Transition from (a) the conventional C-grid staggering as used in NEMO to (b) the E-grid staggering proposed in this study. T-centric (red) and F-centric (blue) cells. $d$ is the damage tracer, $u$ and $v$ are the $i$- and $j$-wise components of the sea-ice velocity vector, and $\sigma_{kl}$ are the components of the internal stress tensor. The $\hat{x}$ notation indicates that variable $x$ is specific to the F-centric grid. Note: the F-centric counterparts of $u_{i,j}, v_{i,j}$ of the T-centric cell are $\hat{u}_{i+1,j}, \hat{v}_{i,j+1}$.

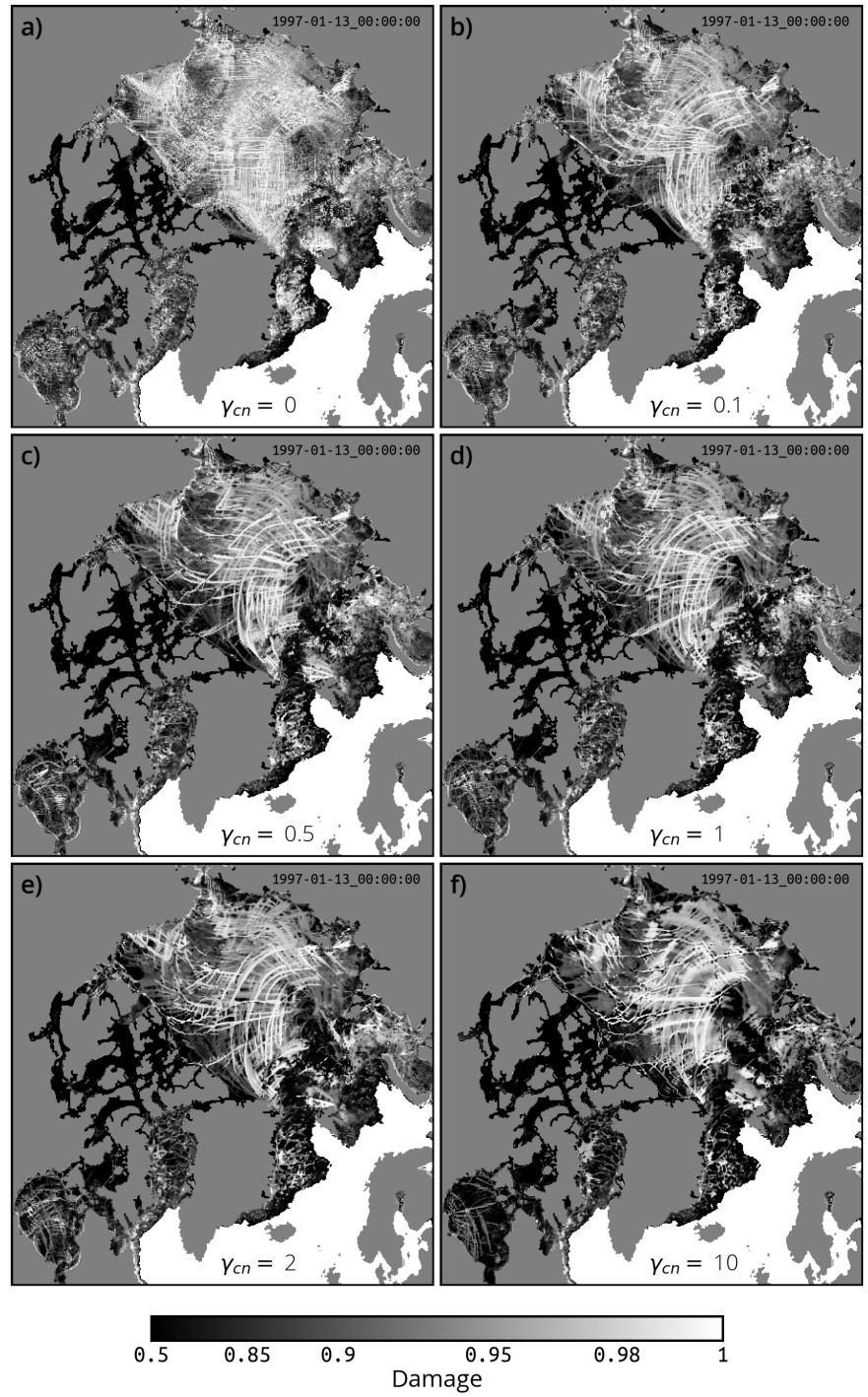

**Figure 4.** Effect of using different values for the cross-nudging coefficient $\gamma_{cn}$ on the simulated sea-ice damage. Random snapshot of damage (at T-points, January 13$^{th}$ 1997) after 30 days of simulation using the specified value of $\gamma_{cn}$, in a set of sensitivity experiments identical to SI3-BBM: (a) no cross-nudging , (b) $\gamma_{cn}$ = 0.1, (c) $\gamma_{cn}$ = 0.5, (d) $\gamma_{cn}$ = 1 as in SI3-BBM, (e) $\gamma_{cn}$ = 2, and (f) $\gamma_{cn}$ = 10.

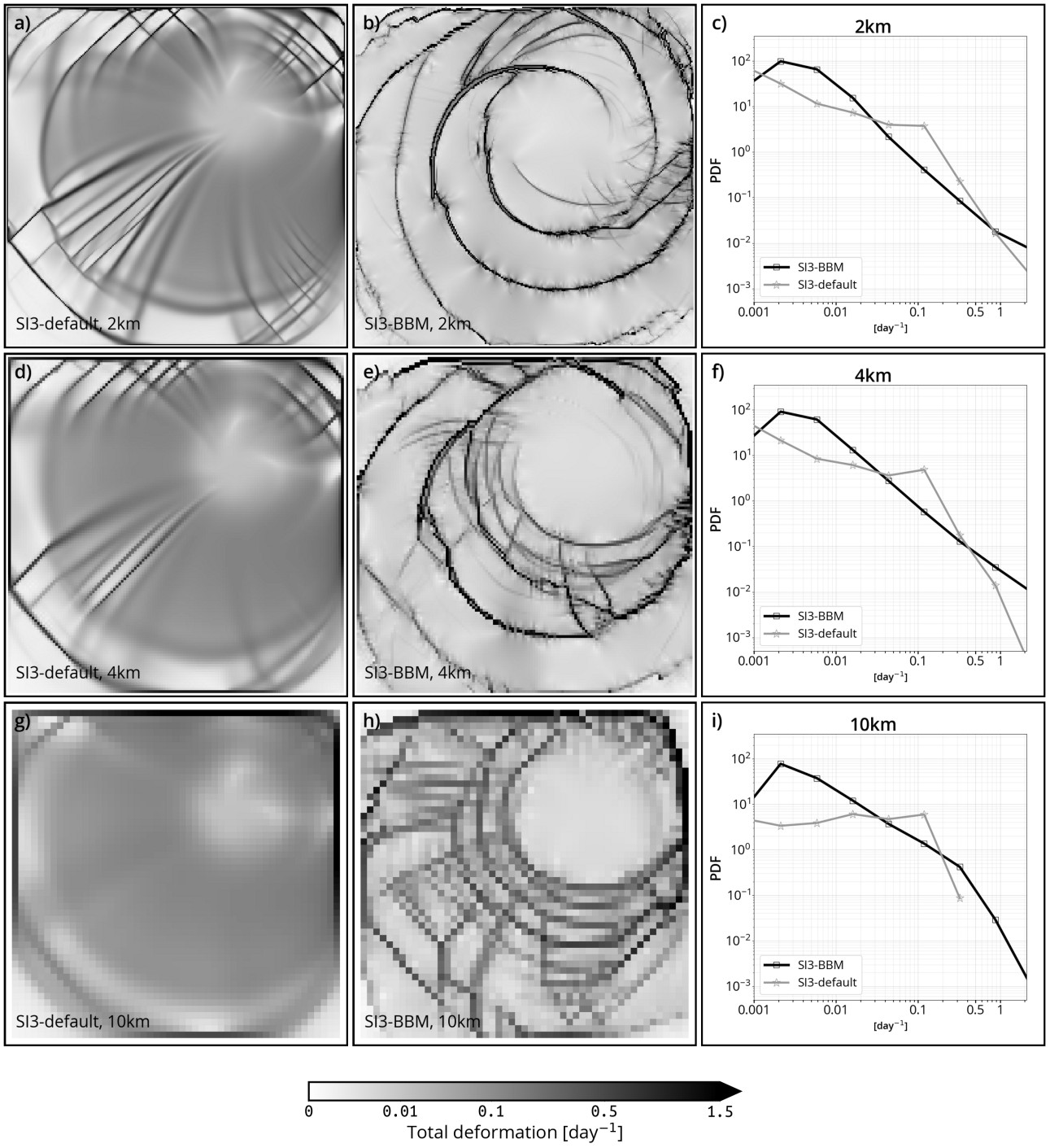

**Figure 5.** Sea-ice total deformation (instantaneous) in the test-case described by Mehlmann et al. (2021), after 48 hours of simulation with SI3, using the default SI3 aEVP setup and the newly-implemented BBM rheology (l.h.s. and middle column, respectively) and the corresponding PDFs (r.h.s. column), for simulations run at a spatial resolution of 2, 4, and 10 km (1st, 2nd and 3rd row, respectively).

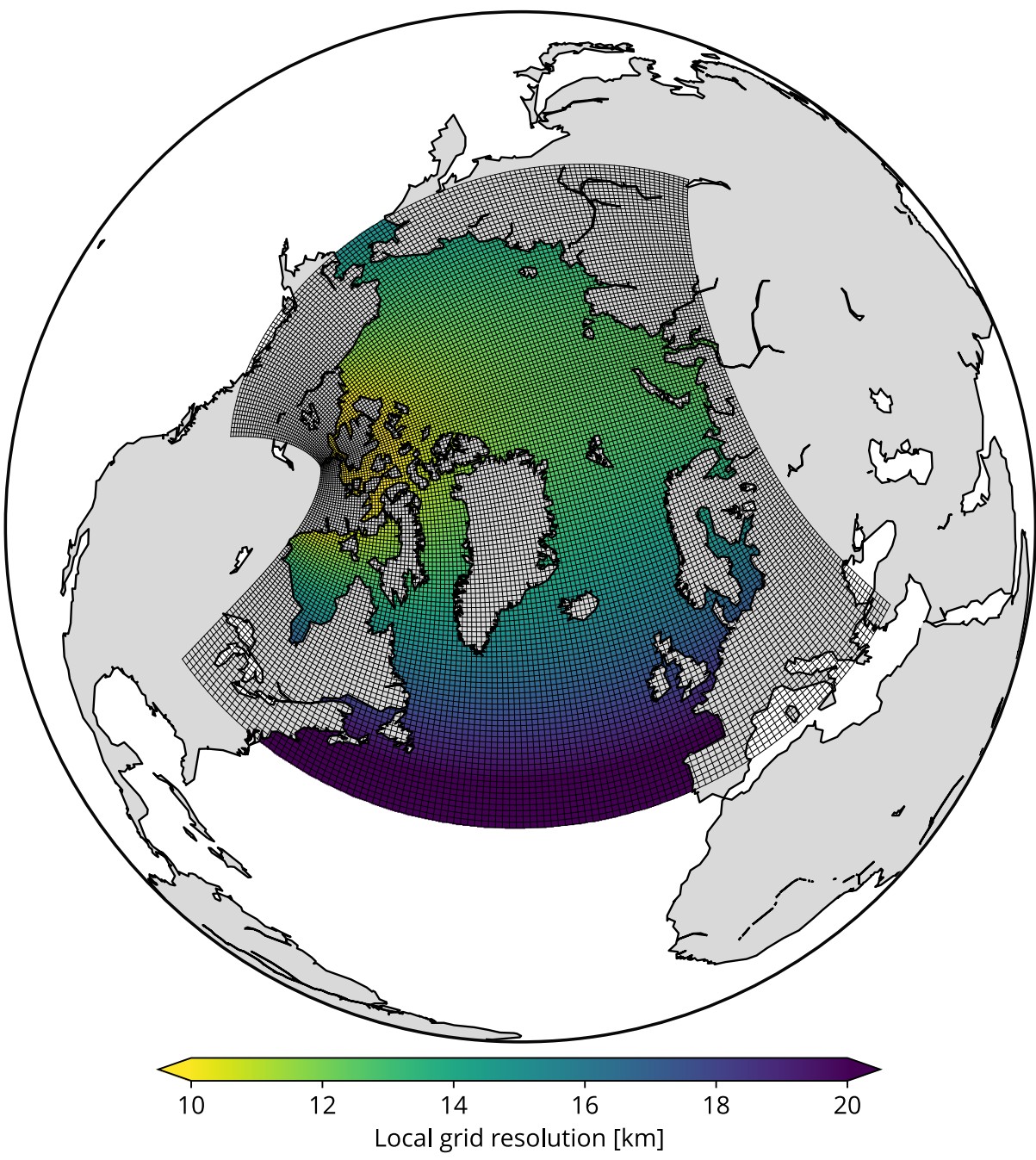

**Figure 6.** Geographical extent, numerical grid, and actual local spatial resolution of the NANUK4 computational domain that is used in the experiments. For ease of visual representation of the grid cells, grid points have been subsampled by a factor of 4.

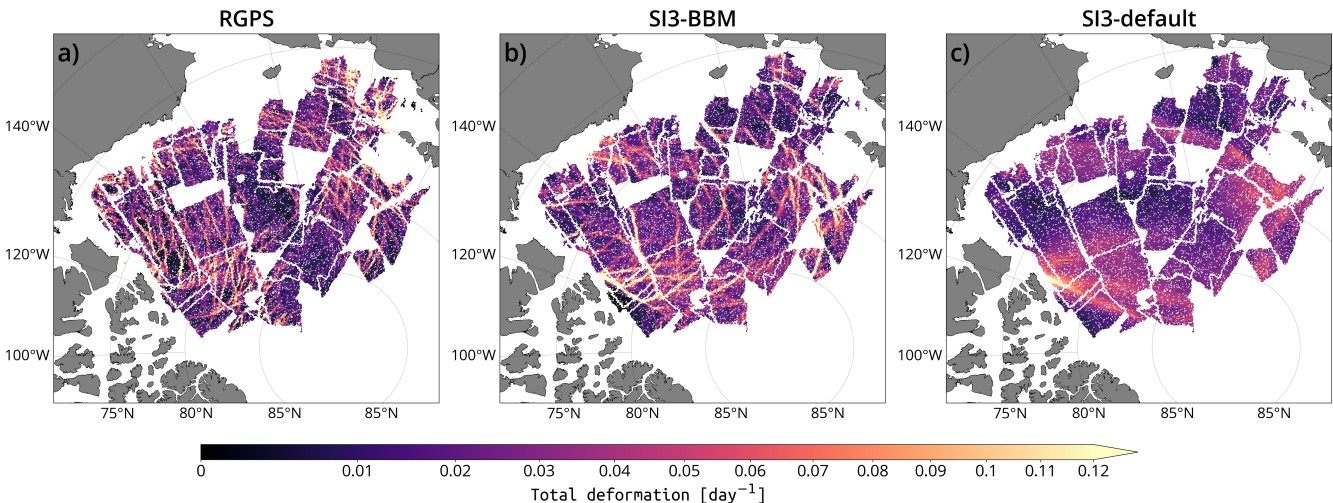

**Figure 7.** Maps of the sea-ice total deformation rate, at the 10 km spatial and 3-day temporal scale, for the period centered about December 24th 1996, computed based on (a) RGPS *Lagrangian* data and (b,c) their synthetic counterparts constructed using the simulated sea-ice velocities of SI3-BBM and SI3-default, respectively. Empty regions correspond to the absence of satellite data during the period concerned.

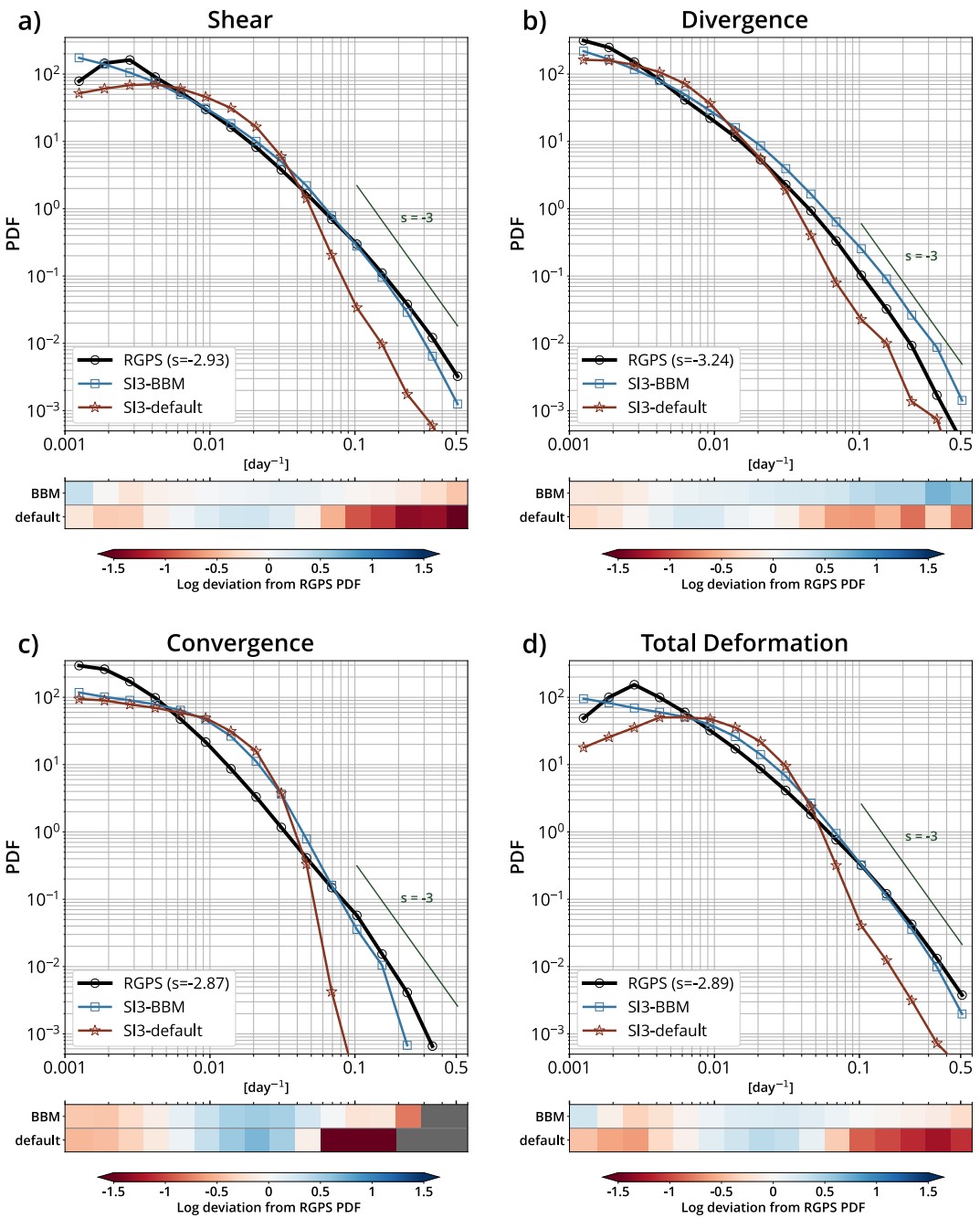

**Figure 8.** PDFs of the (a) shear, (b) divergence, (c) convergence, and (d) total deformation rates at the 10 km spatial and 3-day temporal scale, for RGPS data and their synthetic counterparts constructed using the simulated sea-ice velocities of SI3-BBM and SI3-default. The light gray lines are for reference and correspond to a power-law with an exponent of -3. Below each panel, the departure between the logarithm of the simulated and observed distributions is shown for each bin.

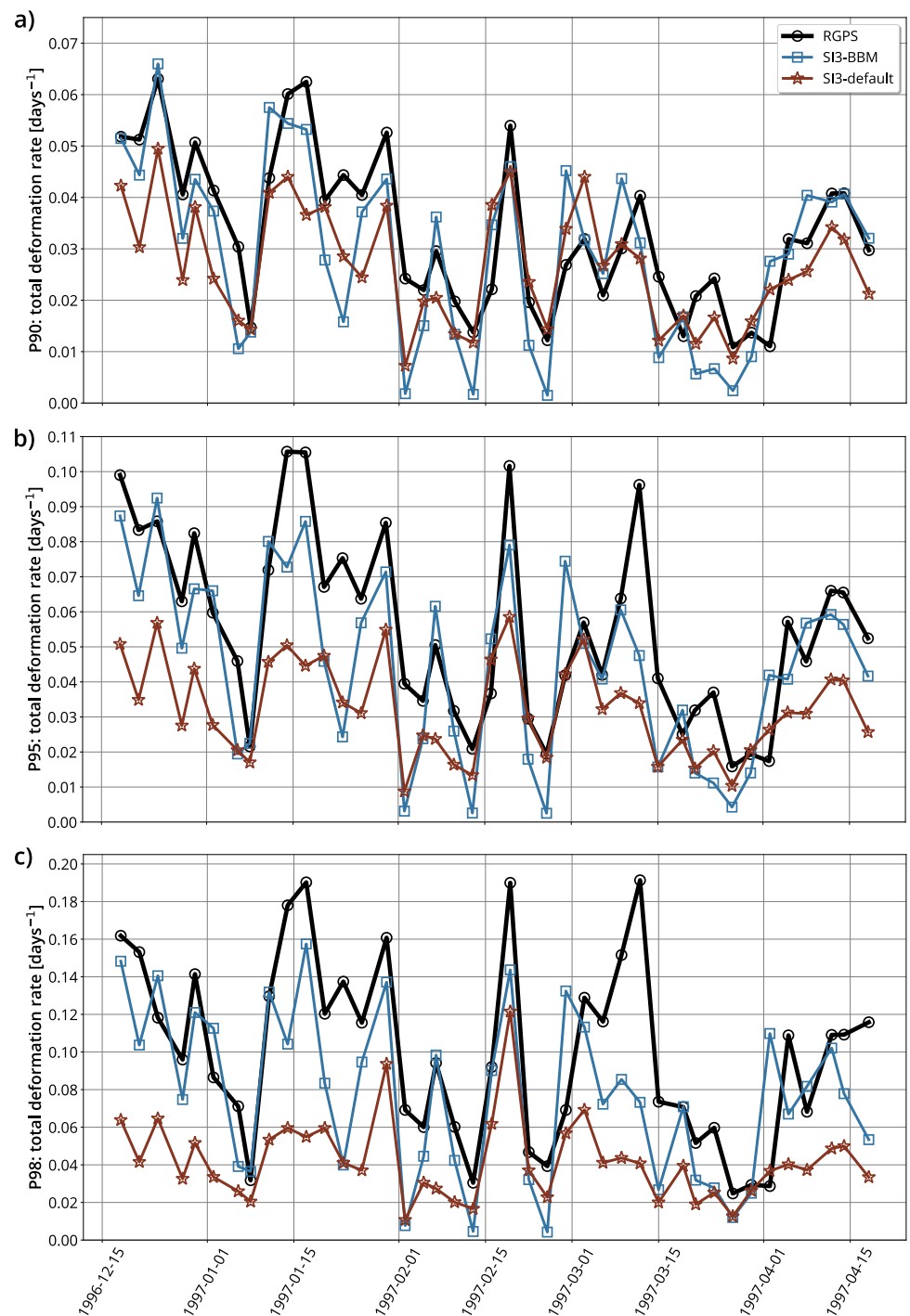

**Figure 9.** Time-series of the a) 90, b) 95, and c) 98 percentiles of the sea-ice total deformation rate for winter 1996-1997, at the 10 km spatial and 3-day temporal scale, for RGPS data and and their synthetic counterparts constructed using the simulated sea-ice velocities of SI3-BBM and SI3-default.

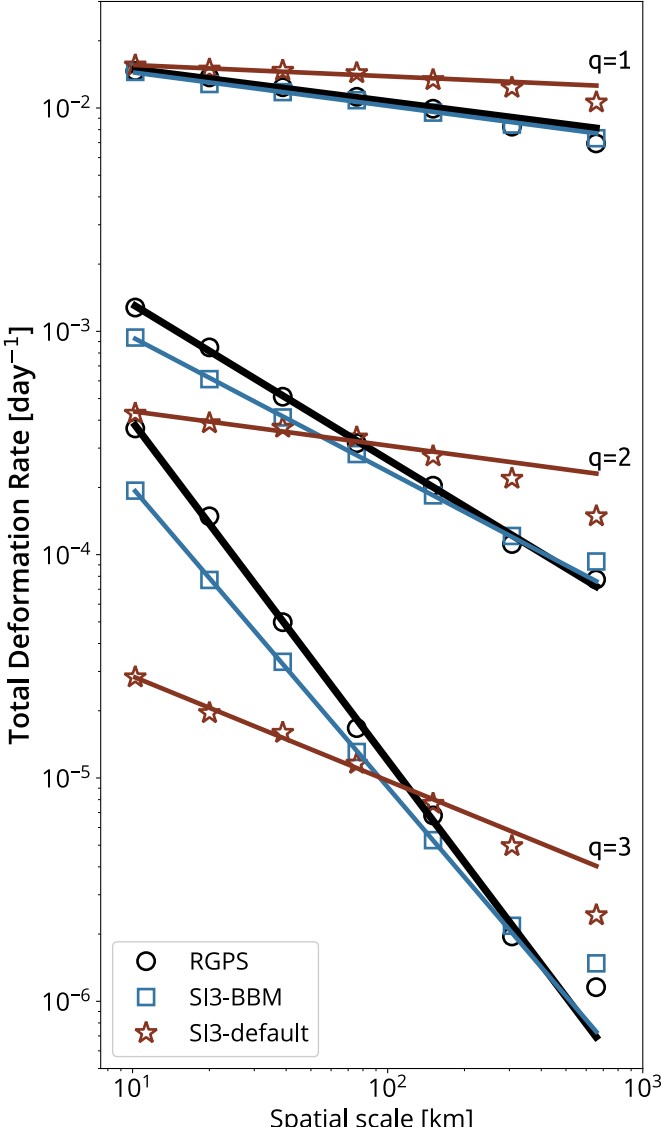

**Figure 10.** Spatial scaling analysis of the observed and simulated total deformation rate calculated over a 3-day time scale (all based on the motion of the same RGPS quadrangles) , based on RGPS data and their synthetic counterparts constructed using the simulated sea-ice velocities of SI3-BBM and SI3-default. Moments of order $q = 1, 2, 3$ of the distributions of the total deformation rate calculated at scales spanning 10 up to 640 km. The solid straight lines indicate the associated power-law scaling based on the least-square fit using values from 10 km to 160 km. Values for 320 km and 640 km are excluded due to excessive uncertainty resulting from the small sample size. Note: we used logarithmically spaced bins and applied an ordinary least square method to the binned data in log-log space to get reasonably accurate estimate of these power-law fits (Stern et al., 2018).

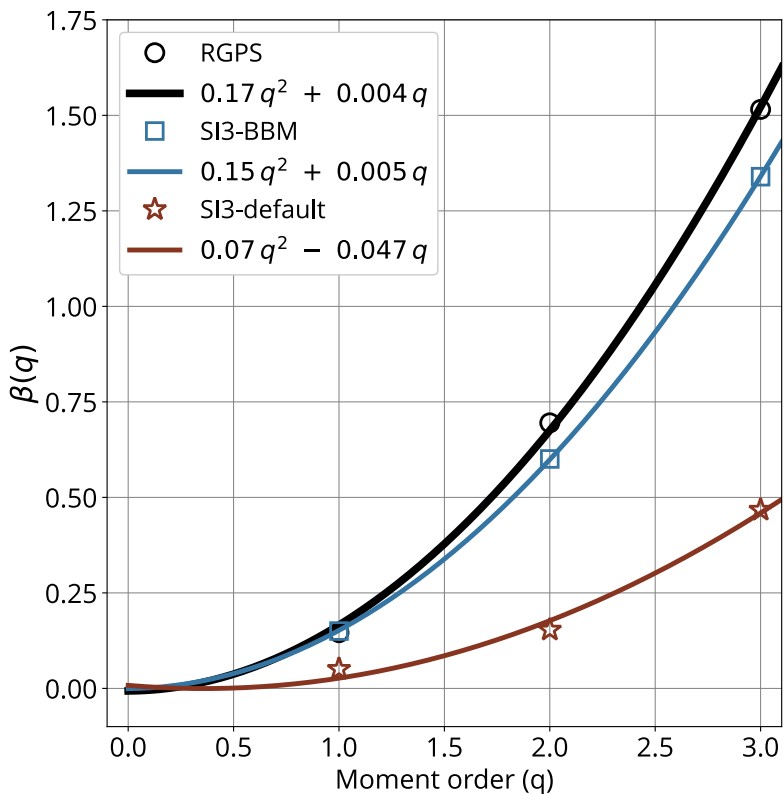

**Figure 11.** Structure functions $\beta(q)$ for the RGPS data (black), SI3-BBM (blue), and SI3-default (red), where $\beta$ indicates the exponent of the power-law fits indicated in figure 10 and $q$ is the moment order.

**Table 1.** Values of default aEVP-related parameters in SI3, as used in experiment SI3-default.

| Parameter | Definition | Value used |
|:---:|---|---|
| $C$ | compaction parameter | 20 |
| $P^*$ | ice strength thickness parameter | $20 \cdot 10^3$ N m$^2$ |
| $e$ | eccentricity of the elliptical yield curve | 2 |
| $C_D^{(a)}$ | ice-atmosphere drag coefficient | $1.4 \cdot 10^{-3}$ |
| $C_D^{(o)}$ | basal ice-water drag coefficient (Eq. B18) | $5 \cdot 10^{-3}$ |
| $N_{\text{EVP}}$ | number of iterations | 100 |

**Table 2.** Values of BBM-related parameters implemented in SI3, as used in experiment SI3-BBM.

| Parameter | Definition | Value used |
|:---:|---|---|
| $\nu$ | *Poisson*'s ratio (Eq. 4) | 1/3 |
| $E_0$ | elasticity of undamaged sea-ice (Eq. 5) | $5.96 \cdot 10^8$ Pa |
| $\lambda_0$ | viscous relaxation time of undamaged sea-ice (Eq. 6) | $10^7$ s |
| $C$ | compaction parameter (Eq. 6, 5, 8) | 20 |
| $\alpha$ | damage parameter (Eq. 6) | 5 |
| $P_0$ | scaling parameter for ridging threshold (Eq. 8) | $10^4$ Pa |
| $h_0$ | reference ice thickness for ridging threshold (Eq. 8) | 1 m |
| $c$ | sea-ice cohesion (Eq. 13) | $5.8 \cdot 10^3$ Pa |
| $\mu$ | internal friction coefficient (Eq. 13) | 0.7 |
| $N$ | upper limit for compressive stress (Eq. 13) | $2.9 \cdot 10^7$ Pa |
| $k_{th}$ | healing constant for damage (Eq. 14) | 26 K s |
| $N_s$ | time-splitting parameter (Eq. 15) | 100 |
| $\Delta t$ | dynamical time-step (Eq. 15) | 7.2 s |
| $\gamma_{cn}$ | cross-nudging coefficient (Eq. 16) | 1 |
| $C_D^{(a)}$ | ice-atmosphere drag coefficient | $2 \cdot 10^{-3}$ |
| $C_D^{(o)}$ | basal ice-water drag coefficient (Eq. B18) | $5 \cdot 10^{-3}$ |

**Table 3.** Bias, RMSE and *Pearson* correlation of the deformation rates time-series of figure 9 obtained between each simulation and RGPS.

|  | Experiment | Bias | Error | $\rho$ (p-value) |
|---|---|---|---|---|
| P90 | SI3-BBM | -0.004 | 0.011 | 0.81 ($2 \cdot 10^{-10}$) |
|  | SI3-default | -0.006 | 0.011 | 0.78 ($2 \cdot 10^{-9}$) |
| P95 | SI3-BBM | -0.01 | 0.02 | 0.79 ($1.2 \cdot 10^{-9}$) |
|  | SI3-default | -0.02 | 0.03 | 0.77 ($4.2 \cdot 10^{-9}$) |
| P98 | SI3-BBM | -0.02 | 0.04 | 0.69 ($5.4 \cdot 10^{-7}$) |
|  | SI3-default | -0.06 | 0.07 | 0.7 ($3 \cdot 10^{-7}$) |

**Table 4.** Computational cost of 3 months (90 days) of Pan-Arctic sea-ice simulation at 1/4°resolution with SI3 on the NANUK4 regional domain (31 vertical levels), with an advective time step of 720 s, run on 29 cores in parallel (with output data writing disabled to limit the influence of I/O). Default SI3 aEVP setup ($N_{EVP}$ = 100) versus newly-implemented BBM rheology ($N_s$ = 100), for both standalone (SI3-SAS) and coupled (SI3-OCE).

|  | SI3-default | SI3-BBM | BBM-related increase |
|---|---|---|---|
| SI3 – SAS (standalone) | 80 cpu h | 116 cpu h | +45% |
| SI3 – OCE (coupled) | 193 cpu h | 232 cpu h | +20% |
| → added cost of OCE | 113 cpu h | 114 cpu h | - |
| Relative cost of SI3 in coupled mode | 41% | 50% | - |

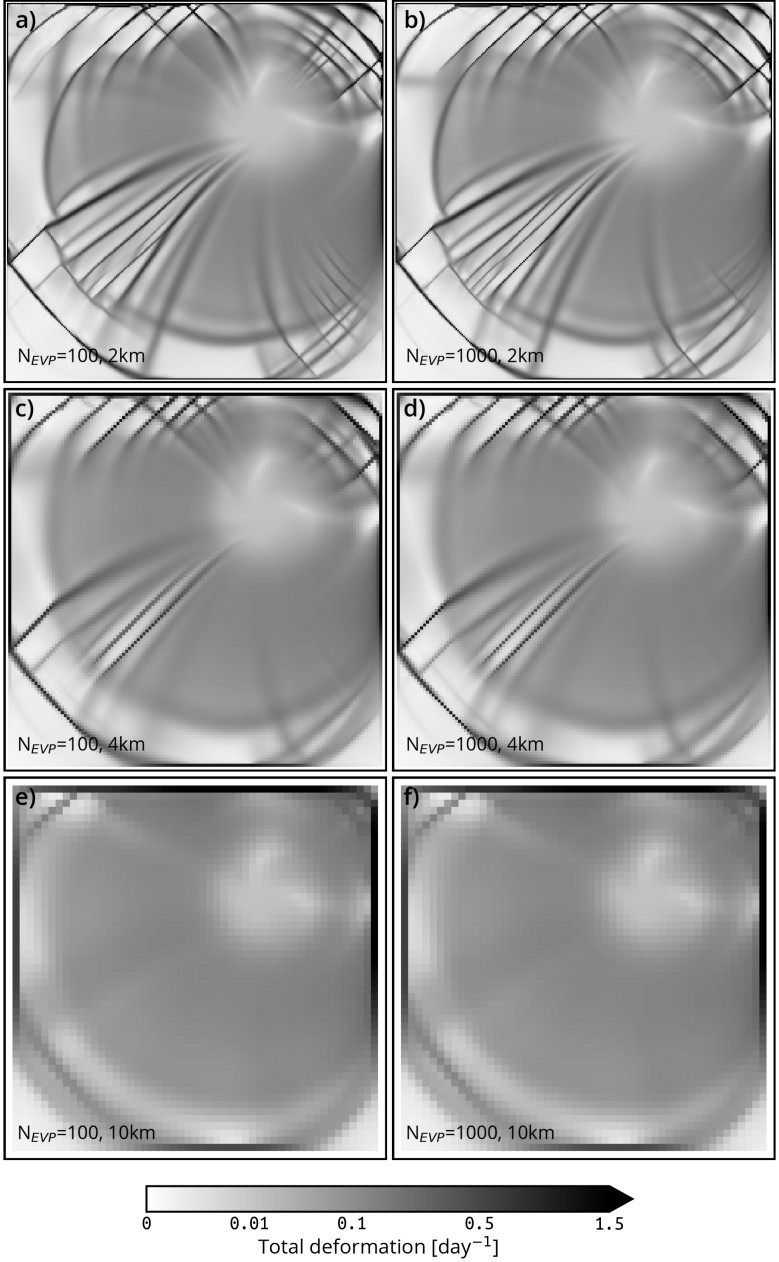

**Figure C.1.** Sea-ice total deformation (instantaneous) in the test-case described by Mehlmann et al. (2021), after 48 hours of simulation with SI3, using the default SI3 aEVP rheology with $N_{\mathrm{EVP}} = 100$ and $N_{\mathrm{EVP}} = 1000$ (l.h.s. and r.h.s. column, respectively), at 2, 4, and 10 km resolution ($1^{\mathrm{st}}$, $2^{\mathrm{nd}}$ and $3^{\mathrm{rd}}$ row, respectively).

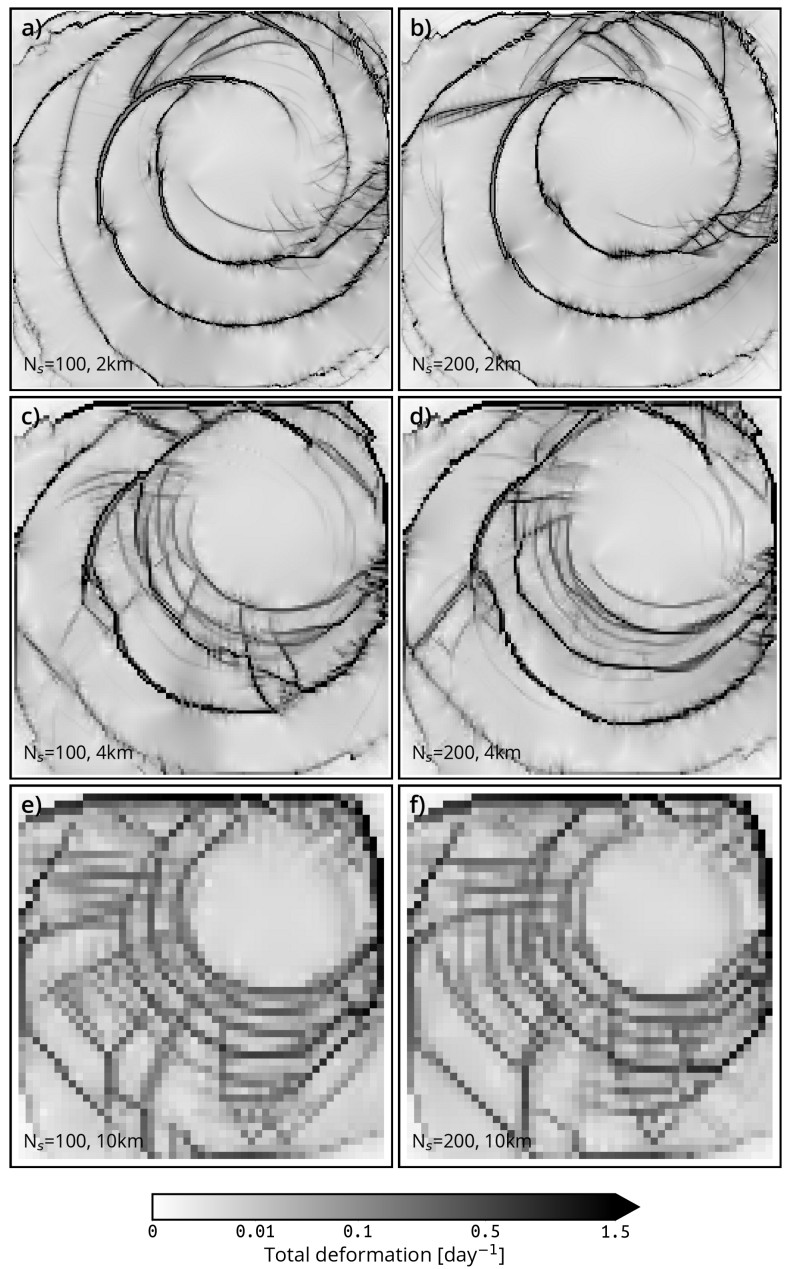

**Figure C.2.** Sea-ice total deformation (instantaneous) in the test-case described by Mehlmann et al. (2021), after 48 hours of simulation with SI3, using the newly-implemented BBM rheology with $N_s = 100$ and $N_s = 200$ (l.h.s. and r.h.s. column, respectively), at 2, 4, and 10 km resolution (1st, 2nd and 3rd row, respectively).

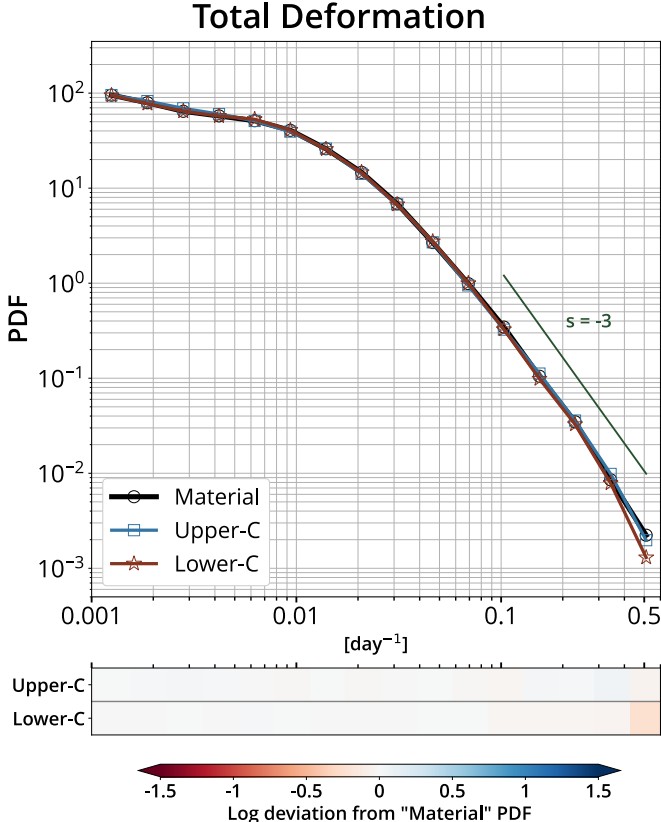

**Figure C.3.** PDFs of the total deformation rates at the 10 km spatial and 3-day temporal scale, for the synthetic counterparts of RGPS data constructed using the simulated sea-ice velocities of three BBM-driven SI3 experiments that use different time-derivative formulations for the stress tensor: only material derivative ("Material", black line with circle markers), upper-convected time derivative as in SI3-BBM ("Upper-C", blue line with square markers), and lower-convected time derivative ("Lower-C", red line with star markers).