# Peer review of "Implementation of a brittle sea-ice rheology in an Eulerian, finite-difference, C-grid modeling framework: Impact on the simulated deformation of sea-ice in the Arctic"

_Geoscientific Model Development, 2023_

## Referee Comment (RC3)

Review of:

"*Implementation of a brittle sea-ice rheology in an Eulerian, finite-difference, C-grid modeling framework: Impact on the simulated deformation of sea-ice in the Arctic*" by Laurent Brodeau, Pierre Rampal, Einar Ólason, and Véronique Dansereau.

This manuscript presents the implementation of the Brittle Bringham Maxwell (BBM) rheology in the SI3 community sea ice model, which uses a C-Grid Finite Difference framework. In particular, the authors present their use of the E-Grid discretization scheme to avoid difficulties with the staggered velocity and stress components in the C-Grid. They also present their use of an upper-convected time derivative approach to advect the stress components. The performance of their BBM implementation is then assessed by comparing the sea ice deformation statistics in SI3 simulations using the implemented BBM and the aEVP rheology.

I find that the manuscript is interesting, important and has potential for publication in GMD. In particular, it provides detailed discussions on difficulties implementing the BBM rheology in the SI3 sea ice model, and presents the numerical tools used by the authors to overcome them. This is an important step for the rheology to be more thoroughly investigated and used by the community. However, the methods used to validate the implementation is not adequate, and leaves doubts on the realism of the implemented BBM. In particular (among others listed below): why is SI3 initialised from neXtSIM fields? This seems to imply that the implemented BBM is not performing well unless the damage is restarted from the other model. This method also makes the comparison between the implemented BBM and the aEVP difficult to interpret, as the BBM benefits from the initial neXtSIM damage, and thus likely has an inherited heterogeneity.

As such, while the implementation itself makes for an interesting publication, significant work needs to be done to address these validation method weaknesses. This may require changing the methods, and thus significant time to be adequately addressed.

For these reasons, I recommend this manuscript to be send back to the authors for major revisions.

Best regards,

Mathieu Plante

**Major points:**

- In the methods, it is indicated that the sea ice model is initialized with fields from neXtSIM simulations. This choice rise doubts on whether this BBM implementation is usable on its own, or if it is it only realistic if initialized from neXtSIM fields. Also, and most importantly, this method brings a significant heterogeneity advantage for the BBM in the results, as it is additionally inheriting the damage from neXtSIM. It could very well be that the differences discussed in the result section is only a reflection of this difference. This is an important validation weakness and needs to be addressed.

- It is difficult to assess the performance of the rheology implementation without any confirmation that the sea ice drift, thickness and concentration are realistic. Also, it is mentioned in the methods (L340-342) that the simulations are tuned to yield similar mean sea ice deformation values. While this method may make sense for the presented metrics, I expect that this tuning comes with significant (and possibly unrealistic) differences in sea ice drift, transport and thickness. Yet, this is not investigated. Furthermore, the difference in drag coefficient means that there are larger internal forces in the BBM simulation than in the aEVP simulation, which likely largely impacts the PDF of sea ice deformations. This tuning method is thus also likely advantaging the BBM simulation, and may not be well suited for this validation. It should thus be revised / discussed more in-depth.

- Results from the aEVP simulation seem particularly bad, even for an EVP model at 10km resolution. Is this using the standard aEVP model parameters in SI3? Please provide more information in that regard. While I understand that tuning the aEVP parameters might be out of the scope of this paper, this will nonetheless likely be picked by the community as being an unfair comparison, especially that we know some model parameters that are better at representing LKFs (i.e., decreasing the ellipse aspect ratio). This is especially noteworthy given that the BBM parameters are themselves tuned for a better performance (e.g. in section 2.3.3). A similar effort should be done for the aEVP parameters, or at least the authors should indicate and justify the choice of not doing so in the analysis.

- It is often implied, as motivation for this implementation, that it is not possible to implement the brittle models with the staggered components in the C-grid, citing Plante at al., (2020) as an example. This is inaccurate and misleading: Plante et al., (2020) in truth demonstrated how to implement the MEB with staggered components without yielding numerical problems. The numerical inaccuracies assessed in Plante et al., (2020) were not related to the staggered stress components, but to the mathematical expression of the prognostic damage equation under large convergence (i.e., their paper demonstrated the same instabilities mentioned at L131 in the current manuscript, with a follow-up, Plante and Tremblay, 2021, offering a solution). The proposed method here should thus not be presented as a solution to a numerical instability problem, but as a novel approach to avoid the treatment of staggered stress components.

- While the text is mostly clear, some sentences are a bit wordy and tedious. This is throughout the text but most particularly in the introduction and section 3.

**Specific comments** (some repeating the comments above):

L8-12: This sentence is tedious; I am not sure what you are trying to say. Do you mean results show that the BBM in SI3 is suitable for climate simulations? If so, I think that this is far-fetch, only based on the localisation of sea ice deformations. I don't think that this is appropriate for this manuscript.

Introduction: This section needs some work: it is a bit "wordy" and it feels more drafty then the rest of the manuscript.

L15: remove "all"

L42: "as an upgrade of MEB": this is a non-physical way to put it. The two rheology represent different physics, and we cannot call the BBM as an "upgrade" of the MEB. Please rephrase more objectively.

L45: "preserving […] the thickness pattern of the sea-ice cover consistent with observations", preserving here does not work. Rephrase.

L46-47: If it is proven, add the associated references. Otherwise, rephrase.

L64-65: This should be re-worded, as the McGill (and MITgcm) model is capable of producing pan-Arctic simulations (i.e., the models are not limited to the experiments presented in specific papers). The idealised experiments in Plante et al., (2020) were simply not designed to compare against sea ice drift and deformation observations. What you mean here is rather that the other implementations have not yet been assessed in a pan-Arctic context against sea ice drift and deformation observations? This might not be the right focus for this manuscript.

L69-75: Here it is implied that the problem has no solution yet, although it was overcome in Plante et al., (2020). You should rephrase to reflect that here you propose a different approach, one that avoids interpolation.

L70: "is not well-suited for brittle rheologies": This is too vague and feels subjective. Do you mean that "it makes for a difficult implementation of brittle rheologies, in which any error in the prognostic stress variables are also integrated in the damage parameter"?

L74: Perhaps: → we propose a novel approach to address this difficulty ?

Eq. 7: I see that the exponential SIC term is inside the parenthesis, and thus also has the dependency on alpha? This is different from other brittle implementations, and the rationale behind this change should be provided. Note also that this parameter is indicated as beta in Eq. B6, please rectify.

Eq. 13: This formulation (Plante and Tremblay, 2021) miss-matches the comment at L121-122, as the stress state is only going back directly to the yield curve if $\Delta t = T_d$ .

L139-141 : "that of a viscous-plastic one such as VP" → that of a viscous-plastic (VP) rheology?

L143: This is not exact: the ice is always viscoelastic in the MEB and BBM, not only when damaged. Only, the viscous dissipation acts on a (much) longer time-scale in the undamaged case.

L144: Watch for VP vs. EVP (here and through the text). Here I believe that you mean the EVP.

L150-151: I am not sure that this is a real distinction between MEB and VP models. I.e., regardless of the mechanism to represent "discontinuities" (damage in the brittle models, rate-invariance in the VP), it remains that a continuum model does not resolve discontinuities, which are thus only represented by Finite Difference gradients.

L158: Rephrase: a rheology is not bound to a numerical scheme. For instance, the MEB in the McGill model also uses an implicit iterative approach. The EVP uses explicit formulation.

L161-163: It could be noted that this small time step is also necessary to resolve the elastic waves, otherwise the elastic component (and the damage) is no-longer physically meaningful.

L166: In the case of the stress component, from a scale analysis, the advection is likely very negligible: the time-derivative term is the elastic term, acting at very short time scales, at which the advection is indeed very insignificant, especially at a 10km grid scale.

L170: "tracers are defined at the point located at the center of each cell" → tracers are defined at the cell centers.

L183-188 (suggestion): it could also be noted that in the VP formulation, the stress components are only diagnostics, and this staggering of the strain rate components only affects the computing of the Delta parameter, which itself is not prognostic (not integrated). However, in the brittle models, the staggering affects a larger number of prognostic variables (d, sigma), each of which are integrated and inter-dependent. This make the implementation unforgiving for any incoherence in the treatment of the staggered components.

L194-195: "in addition to being conceptually debatable in the context of a brittle model, which is expected to simulate very sharp spatial gradients", again, I find this a bit misleading, as this points to a caveat of using a continuum model, not the choice of rheology. You never truly resolve discontinuity in FD continuum approach. This sentence thus suggests that the brittle models should not be used in a continuum context to begin with.

L197-198: This is false and inacceptable: Plante et al., (2020) actually demonstrated how to implement the MEB with the staggered components without having these errors, while it is written that Plante et al. (2020) implemented the MEB with has all these errors and inaccurate solutions.

L199-202: This is quite vague, subjective yet strongly worded. What do you mean exactly by "consistent"? Do you mean that one needs interpolation but not the other? How much is it an issue?

L221-222: For consistency, wouldn't it be better to also have A and h defined also at the f-points? So that all tracers defining the E and lambda parameters are treated the same way?

Eq. 16: I am not sure if this is an error (?): I would expect the stresses to be corrected (and not defined) by the right hand side, which represents a (weighted) difference between the two offset

grids. Am I missing something? More precisely, I would expect an equation on the form of sigma_new = sigma_old + weight * Delta-Sigma, but here we have sigma_new = weight* Delta-Sigma. Please clarify.

L271: Each time step, to me, seem to refer to the smaller time step, but the thermodynamics is the longer time step. Please clarify. It would also be useful to define the small and large time steps as the "dynamical" and "advective" time steps.

L307: remove "so-called" and "which is".

L333-334: Can you clarify here that this initialization from OCE-neXtSIM is made for both the BBM and the aEVP simulations? Also, please give more insight on why the sea ice model needs to be initialized from neXtSIM. Are results similar if the model is used without this neXtSIM dependency? I would argue that the analysis would be most meaningful and convincing if the simulations were ran without this neXtSIM initialization.

L333: "and damage (only for SI3-BBM)": this is a very important difference in the method between the simulations, yet this effect is not at all discussed or investigated. It means you initialize the BBM with heterogeneity imported from neXtSIM, but less so for the aEVP case.

L342: This tuning of the sea ice deformation could be interesting, as opposed to tuning the sea ice drift, but we need to understand how else it impact the simulations. How is it impacting the sea ice drift and thicknesses? Are they very different between the simulations? Are they all realistic, or is this tuning made at the cost of the large-scale dynamics? This should be shown and discussed.

L354-355: How does your method compares with the SIREx methods?

L368: This to me looks like a particularly bad result even for the aEVP at 10km: we indeed do not expect the EVP to produce much LKF at this resolution, but this result is particularly smooth... Is this using the default aEVP parameters in SI3, or can it be a result of the drag coefficient tuning? This should be discussed, and add some notes here to show awareness that the aEVP results could be improved by simple modifications to the model parameters, and justify the choice of not doing so for the comparison.

Section 3.4.2: This feels a bit strange as you demonstrate just above that the aEVP simulation does not have a tail. So, it is unclear what is the purpose of this analysis, as it only repeats what is already demonstrated in section 3.4.1.

L398: "a transition" : Please clarify what transition this refers to.

L403: "Interestingly", do you mean "In particular"? It sounds as if this result is not expected.

L408: These $c_{da}$ values should be provided in the method section, at L342. Also, this is a surprisingly wide difference between the simulations, and thus suggest that the internal stresses would be much smaller in the aEVP simulations. This surely impacts the production of large

deformation rates and LKFs. This needs to be investigated as to show how much this impacts the difference in representing the tail of the sea ice deformation PDFs.

L417-426: What is the purpose of this analysis? It is unclear yet what the structure functions tell us, in terms of model performance (Bouchat et al., 2022).

L429-431 Again, this is a strong comment that is not demonstrated. It also does not reflect that there have been successful implementations of the MEB on the staggered C-grid.

L472-474: A note could be added here that the fully-converged EVP likely influences the production of LKFs.

L483-485: This investigation is interesting and should be shown.

L487-489: This has not been assessed here, there are no comparisons with neXtSIM.

Conclusion: This section includes several strongly worded but subjective statements that do not serve well the actual contribution of the manuscript. In the context of a scientific manuscript, such subjective statements have a history of distracting readers from the actual analysis and to raise doubt on the transparency. This should be revised to be more nuanced and to keep the focus on the presented analysis.

L499-501: This is not accurate and too strongly worded. As the authors acknowledge in the introduction, the MEB has been implemented in the McGill sea ice model, and the MITgcm, which are both capable of producing pan-Arctic simulations.

L501-502: "has proven to be poorly fitted for brittle rheologies." Again, it is not demonstrated. This is a rather subjective statements not reflecting that the MEB was successfully implemented on staggered C-grids in other models.

L520-523: This last paragraph needs to be removed. It has not even been assessed here: there are no comparison with neXtSIM, only against aEVP. It is particularly blind to the listed numerical sensitivities associated with this implementation, which likely strongly affect the results (e.g., just looking at Figure 5).

L571: If I understand well, you get the stress state as a function of the (d, u, v, h, A) from the previous iteration (i.e. from the constitutive equation), but then apply the cross-nudging before computing the new damage. One issue I see with this is that the propagation of damage is supposed stem from the stress concentration around the previous damage. By applying the cross-nudging between these steps, I fear that the cross-nudging would affects the development of damage and the orientation of its propagation. As this method is a main contribution of this paper, I believe it would be worth providing some analysis and discussion on this regard. Also, to me, the cross-nudging is not very different, in terms of numerical effect, than using interpolation: you are smoothing between two solutions instead of between staggered components. Not that it is an issue to me, but as the text highlights that interpolating is not

appropriate, it then raises the question: how is this cross-nudging more appropriate than interpolating?

References:

*Bouchat A, Hutter NC, Dupont JCF, Dukhovskoy DS, Garric G, Lee YJ, Lemieux JF, Lique C, Losch M, Maslowski W, Myers PG, Ólason E, Rampal P, Rasmussen T, Talandier C, Tremblay B and Wang Q (2022) Sea Ice Rheology Experiment (SIREx), Part I: Scaling and statistical properties of sea-ice deformation fields. J. Geophys. Res., 127(4), e2021JC01766 (doi: 10.1029/2021JC017667)*

*Hutter NC, Bouchat A, Dupont F, Dukhovskoy DS, Koldunov NV, Lee YJ, Lemieux JF, Lique C, Losch M, Maslowski W, Myers PG, Ólason E, Rampal P, Rasmussen T, Talandier C, Tremblay B and Wang Q (2022) Sea Ice Rheology Experiment (SIREx), Part II: Evaluating simulated linear kinematic features in high-resolution sea-ice simulations. J. Geophys. Res., 127(4), e2021JC017666 (doi: 10.1029/2021JC017666)*

*Plante M, Tremblay LB, Losch M and Lemieux JF (2020) Landfast sea ice material properties derived from ice bridge simulations using the Maxwell elasto-brittle rheology. The Cryosphere, 14(6), 2137–2157*

*Plante M and Tremblay LB (2021): A generalized stress correction scheme for the Maxwell elasto-brittle rheology: impact on the fracture angles and deformations. The Cryosphere, 15 (12), 5623–5638*

---

## Author Comment (AC4)

$$\begin{pmatrix} \sigma_{11} \\ \sigma_{22} \\ \sigma_{12} \end{pmatrix} = \begin{pmatrix} \sigma_{11} \\ \sigma_{22} \\ \sigma_{12} \end{pmatrix} + \gamma_C \, \frac{\Delta t}{\Delta T} \begin{pmatrix} \sigma_{11} - \texttt{interpF@T}(\hat{\sigma}_{11}) \\ \sigma_{22} - \texttt{interpF@T}(\hat{\sigma}_{22}) \\ \sigma_{12} - \texttt{interpT@F}(\hat{\sigma}_{12}) \end{pmatrix} \qquad \text{(even time-step )}$$

$$\begin{pmatrix} \hat{\sigma}_{11} \\ \hat{\sigma}_{22} \\ \hat{\sigma}_{12} \end{pmatrix} = \begin{pmatrix} \hat{\sigma}_{11} \\ \hat{\sigma}_{22} \\ \hat{\sigma}_{12} \end{pmatrix} + \gamma_C \, \frac{\Delta t}{\Delta T} \begin{pmatrix} \hat{\sigma}_{11} - \texttt{interpT@F}(\sigma_{11}) \\ \hat{\sigma}_{22} - \texttt{interpT@F}(\sigma_{22}) \\ \hat{\sigma}_{12} - \texttt{interpF@T}(\sigma_{12}) \end{pmatrix} \qquad \text{(odd time-step )}$$

(16)

---

## Author Comment (AC5)

**Review of "Implementation of a brittle sea-ice rheology in an Eulerian, finite-difference, C-grid modeling framework: Impact on the simulated deformation of sea-ice in the Arctic" by Brodeau et al, (MS gmd-2023-231)**

The manuscript describes the Brittle Bingham-Maxwell (BBM) rheology of Olason et al (2022) and its implementation into SI3, the sea ice component of NEMO. Particular emphasis is placed on the implementation of a staggered C-grid. A few simulations serve to evaluate the implementation relative to statistics of observations and the default VP-rheology with an aEVP solver. The structure of the text is mostly clear.

Clearly this is important work that will make it possible for a large community sea-ice/ocean/climate models to use the BBM model in their simulations. The manuscript, however, requires major revisions to turn this into a scientific paper.

As a preamble, we want to start by sincerely thanking the anonymous reviewer for his extensive review of our manuscript, both in terms of depth and length. It contains numerous good and constructive points, and we think it significantly contributed to making our manuscript way more accurate and solid than it was at the first submission stage. We tried to address every comment and suggestion carefully, which is also why our response to the reviews took quite a significant time.
Yet, we regret the tone used sometimes by the reviewer, which in our opinion was not always appropriate, not always respectful of our work, and suggesting an intentional lack of intellectual honesty from our end.

Before moving to a point-by-point reply format, here is some important information regarding the new simulations performed for the new version of the manuscript.

Based on a concern expressed by both the anonymous reviewer and Dr Plante on our initialization/spinup strategy, we have decided to use a simpler approach. As such, our simulations have been performed again: initialized with the new strategy (described in section 3.2 of the new manuscript), and consequently, all diagnostics, figures, and table values have been processed and generated again, based on these new simulations.

Also based on concerns raised by the reviewers, on what has been wrongly interpreted as unfairness towards the aEVP rheology, we now make it very clear, all over the paper, that what we use as the reference SI3 simulation is the default workhorse setup of SI3 as provided in the current version of NEMO. As such, for instance, experiment "SI3-aEVP" has been renamed to "SI3-default", and it is clearly stated in the paper that a better-tuned "aEVP" would likely perform better (L513-516).

Furthermore, following the suggestion of reviewer #2, we have included a new section (3.1) that presents results of idealized SI3 simulations run on the "cyclone" test-case of Mehlmann

et al. 2021, using both our BBM implementation and the default aEVP setup of SI3 (new figure 5).

Another important information regarding the new simulations:
In the first version of the manuscript, our developments and simulations were based on the version 4.2.1 of NEMO. At the time, it was the current stable release of NEMO. But during the course of the review process of this paper, a bug related to the ocean-ice drag parameterization has been identified and fixed by the NEMO team. This bug has been judged sufficiently severe by the NEMO team to justify the release of a new stable version: the 4.2.2. Link to release note: https://forge.nemo-ocean.eu/nemo/nemo/-/releases/4.2.2
Consequently we have switched to version 4.2.2 of NEMO for all the simulations presented in the new version of the manuscript.
The bug fix has significantly impacted the values of the simulated deformations and has therefore required a new tuning of the air-ice drag coefficients.

**Major issues**

The main problem with the manuscript is that the statements in the introduction, discussion and conclusion are often not backed by the presented work. Instead, the reader gets the impression that this is a text that tries to "sell" this implementation, where a scientific paper should describe the work and provide objective statements about the value of the work in the context of the scientific discussion. Here, most of the text implies that one should only use the current sea ice model with BBM, because everyone else is getting it work. I am exaggerating a little. Because is a general "impression" after reading the text, it is  very hard to pinpoint individual issues, because the phrasing etc is scattered throughout the manuscript. Here are some more prominent example, and I have listed many places where I found the wording or phrasing to fit for scientific paper. For example:

- The statement about heterogeneity and coupled model in lines 17-20 is too strong. It should be clear that the heat fluxes are very strong in leads and that this is important, but it not as clear how important the transition from parameterising this (i.e. in terms of fraction ice cover) to resolving the leads is. As long as the atmosphere does not resolve the heterogeneity of the high resolution surface fields (typically in coupled system, atmosphere models have a much lower resolution than the ocean/sea ice system, because the characteristic eddy scale is so much larger), it is not clear if this heterogeneity has any impact on the atmosphere, because one needs to average over it anyway.

We toned down the sentence the reviewer is referring to. It now reads:

"This stresses the relevance of accurately representing sea-ice dynamics in simulations of the coupled/multi-component earth system, such as regional and global climate simulations, and even short-term sea-ice predictions.". (L17)

We bring to the reviewer's attention the following sentence taken from the introduction of Hutter et al. (2022) that seems to point towards the same idea, and with whom we tend to agree:

"To directly simulate these processes and to provide a more detailed picture of the complex Arctic climate system, we need sea ice models that explicitly resolve LKFs."

- Already in the model description section (2.3, page 7), the authors come to the conclusion that (l180) "using the C-grid is not the most appropriate choice". Then (l183) "The spatial staggering between the point definition of the normal (diagonal) and shear (off-diagonal) elements of these tensors becomes an issue whenever the parameterization of the constitutive law requires e12 or sig12 to be known at a T-point."

These are unbacked statement. It may be the result of the work presented here, but cannot be the starting point. It's clear that the spatial staggering is not good for all terms. One may argue that the stress tensor divergence is the leading term in the momentum balance. But it is not clear, why this is worse for sea ice models than for ocean models, where the dominant geostrophic balance also suffers from the velocity staggering on C-grids. That's why early ocean models were implemented on B-grids with co-located velocities. The C-grid model has been shown to violate wave dispersion relationships and the Coriolis terms leads to numerical noise. But this noise has been found to be less problematic than the issues of the B-grid, so that now ocean models (e.g. NEMO) mostly use the C-grid in spite of problematic discretisation of the main balance in the ocean (geostrophy).

Since this manuscript stresses the staggering issue so much, it should become clear to the reader, why it is worse for the stress tensor than for velocity vector to not be collocated at one grid point.

We have completely rewritten the parts the reviewer mentions, taking into consideration his remarks, and avoiding any judgmental or "negative" phrasing about any type of grid arrangement everywhere in the new version of the manuscript.

- Interpretation of figures, e.g.

— l372 "power-law tail (in Fig7)"

Power laws lead to a straight line in a loglog plot (as the reference with -3 shows). None of the curves, not even the RGPS curve show that. Clearly the aEVP solution differs more

from the RGPS solution than the BBM solution. But I think that this metric does not allow any strong conclusions (such as "suggests the advantage of BBM over aEVP").

This is true that none of the pdfs are showing perfect power-law tails, and that they are indeed showing a slight curvature/departure from a straight line. This is, by the way, why we include the -3 power law exponent as a "reference" in the figure of the PDFs. The point of our statement about the tails of the PDF being close to a straight line in a loglog scale is another way to say that the variable follows a distribution of values whose moments are strongly impacted by the most extreme values when being computed. This would for example not be the case for a process characterized by Gaussian statistics.

In other words, what is important here is not the perfect alignment with a power law, but rather to note that the tail is clearly closer to a power law than what a Gaussian distribution would exhibit. That being said, we corrected the text to report that the tails of the PDFs can be "approximated" by a power-law. (L606)

Further the aEVP model is not tuned to give high deformation events. It is possible to do so (see Bouchat et al 2022 and Bouchat and Tremblay 2017), so this comparison does not "demonstrate higher skills" of the BBM model, but only the difference between two simulations with default parameters. I think that the authors need to tone down their conclusions.

As suggested by the reviewer, we toned down our conclusions, and modified the text accordingly, which now reads: "We note that the extreme values of deformation rates are, if not absent, largely underestimated for SI3-default in our setup, as highlighted by the departure between the observed and simulated PDFs shown as color bars below each panel of figure 8. This may be improved by better tuning the parameters of the VP model, in particular the ratio between the ice compressive strength and the ice shear strength (Bouchat et al. 2017). However, the scope of this paper is not to perform such a tuning." (L513-516)

— l389 "As illustrated in figure 8, …"

Same as before, clearly the agreement of the BBM simulation with RGPS is better, but the specific aEVP solution has not been tuned to give high deformation, so that designing a metric that emphasises high deformation (for which the brittle model family was designed) is a self-fullfilling prophecy and of little value.

It is not clear to us what the reviewer wishes to achieve with this comment. The figure demonstrates that we manage to do what we set out to do; namely to have a good representation of high deformation events in the model compared to observations. The standard aEVP solution is included as a reference.

What's intersting is that the aEVP solution is not always worse than the BBM-solution (e.g. 03-15 to 04-01). It would be interesting to understand, why BBM, from which we always expect higher deformation, underestimates the high deformation events here.

We agree that there is still much to learn from the analysis presented in (now) figure 9. Unfortunately, a deep analysis of a single metric, including the investigation of single events, is outside the scope of the current paper.

— l416/Fig9: " … but seems to break for scales larger than about 300 km"

Hutter et al 2018, doi:10.1002/2017JC013119 found that the scaling for VP models breaks down at around 10 times the grid spacing, marking the effective resolution of grid-point model where the VP model dynamics are resolved (i.e. you need 5-10 grid points to represent a sharp transition without inflicting numerical issues). With dx=10km in this simulation, it means that anything **below** 100km should be considered as unusable, i.e. the fit should be applied to the 3 points above 100km, which would lead to larger slopes. Brittle models scale down the grid scale, as they are designed to produce heterogeneity at the grid scale. The interpretation of the fig9 hence needs to be revised.

The reviewer is correct when suggesting that our interpretation of (now) figure 10 could have been more extensive. The important point here is not the slopes of the fit, but the fact that BBM in SI3 scales to the resolution of the grid. This has only been shown in more "exotic" modeling frameworks, so it's a new result. The reviewer suggests that we fit only the coarser spatial scales, but whether or not we do so is immaterial to the point above, so we prefer to keep the same, statistically rigorous approach for both models. In the revised paragraph we try to emphasize these points. The new text now reads:

"Figure 10 suggests that the total deformation rates simulated by SI3-default ceases to follow the expected power-law for scales larger than typically 100 km. This is in line with published results (e.g. Hutter et al., 2018; Bouchat et al., 2022). Hutter et al. (2018) argue that the VP model needs approximately ten grid cells to be able to resolve features, which suggests that the "effective resolution" of the model is ten times coarser than that of the numerical grid on which it is run. This implies that one should instead consider fitting the deformation rates at a resolution ten times coarser than that used by the model, i.e. 130 km in our case. This would yield power-law slopes that are in better agreement with those derived from observations. We argue that since sea-ice deformation is a scale-invariant process at the geophysical scale, a sea-ice model should be able to represent this scaling down to the model grid cell. Figure 10 suggests that our BBM implementation allows SI3 to achieve this despite the use of the Eulerian framework.."

- In the conclusions we read (l520) "Based on our results, we conclude that the ability of a continuous sea-ice model to simulate the complex sea-ice dynamics across scales, as observed from satellites, depends primarily on the type of rheology used rather than on the type of modeling formalism chosen (i.e. Eulerian versus Lagrangian)."

It is very likely that this is the case, but the statement is way too strong. I think, one can conclude that the difference between neXtSIM and SI3-BBM is smaller than between the

SI3-BBM and aEVP implementation (although no direct comparison with neXtSIM was made here, only indirectly), in particular in the observed statistics that are attributed to "complex dynamics" of deformation. There are no large scale comparisons about sea-ice distribution, thickness etc. so that the concluding statement can only be about the deformation properties.

Due to the major re-working of the conclusion, this sentence does not exist anymore.

Cited literature is quoted only where it fits the story of superior brittle models, but the same literature is not used to discuss the pros and cons, e.g.

- l32: "poses a fundamental and major challenge (e.g. Bouchat et al., 2022; Hutter et al., 2022)".

Interestingly, "challenge" appears only once in Hutter et al, and never in Bouchat et al. Instead, both papers point to the ambiguity of scaling metrics and multi-fractality as a metric for evaluation different sea ice model. These references do not seem support the statement in this paragraph.

Hutter et al. (2022) conclude that "In this comparison between different sea ice models and model configurations, only very few models reproduce some statistics of LKF properties, namely density, number, length, and growth rate, within an acceptable range as defined by the interannual variability of satellite-based RGPS deformation data. Most models, however, simulate unrealistic LKF distributions". This can surely be called a "challenge", even if Hutter et al. do not use this exact phrase.

Bouchat et al. (2022) conclude that "the VP/EVP rheologies implemented in an Eulerian framework need to be run at higher resolution than that of the observations to yield spatial scaling exponents as high as those observed". Again, it is clearly challenging to reproduce the observed spatial scaling, since very high resolution is a prerequisite.

In our opinion, it is not necessary to go to such lengths in explaining exactly how we interpret that the results of Hutter et al. (2022) and Bouchat et al. (2022) constitute a challenge in the introduction. Specifically, because we do not intend to address these challenges directly, but rather to present a new implementation of a rheology that does attempt to address these challenges.

Instead, I think that Bouchat and Hutter especially highlight the need for more than one MEB model to compare to and introduce more and different comparison and evaluation tools. This manuscript introduces an additional brittle model implementation, but uses the scaling analysis that was shown by Bouchat and Hutter to be not sensitive enough to discriminate between models.

In this paper, we oblige Bouchat and Hutter by presenting a new implementation of the successor of the MEB model. We hope that the community will take advantage of it and use it in future model comparison studies!

The point of our analysis is not to discriminate between models, but to show that our BBM implementation can reproduce the observed spatial scaling. The fact that the default SI3 setup does not show that despite Bouchat et al's conclusions, then doing so is not a foregone conclusion. Unfortunately, Bouchat et al. never show the higher order moments in their analysis, so we don't know how well the models in their study reproduce this important part of the observed statistics or whether they scale down to the grid resolution. By showing the scaling of all three moments, we demonstrate that BBM in SI3 does scale down to the resolution, which is an important goal with the development of the brittle rheologies.

- Plante et al. 2020 is probably the first implementation of MEB on a C-grid; it's not BBM, so the author's can claim that their implementation is the first BBM model the C-grid, but as far as I understand the differences between MEB and BBM (limit the maximum allowed compressive strength P), the C-grid plays no role in this so we can assume that Plante et al have experienced the same issues with the C-grid report here. It is true that (l63) "The idealized nature of these simulations prevented their results from being assessed against observations of sea-ice drift and deformation.", but equally one could say that avoiding idealised configurations makes it impossible to detect (small) implementation errors, that would, e.g., break symmetry etc. In l70, it is written that "[spatial interpolation the C-grid] is not well-suited for brittle rheologies", but it is not mentioned that Plante et al (2020) did find a solution to obtain stable solutions by a mix of double defined variable akin to the E-grid solution presented here and averaging. In line 196, Plante et al is cited to have observed checkerboard (not chessboard) instabilities, but the authors fail to write, that these patterns never appear in the presented solution, because Plante et al designed (and described) a the numerical scheme to avoid them. Instead, the manuscript gives the impression that their E-grid solution is the only way to solve this, thereby neglecting previous successful methods.

It is also not discussed that in Plante's model, the heterogeneity does not appear to be as chaotic and noisy as in the neXtSIM publications. Maybe it makes sense to actually try to reproduce Plante's results?

As you will see, in the new version of the manuscript, we have completely rewritten the parts (and we have added new parts) related to the work achieved by Plante et al. 2020.

To reply to your question, in the early phase of our development, before choosing to use the E-grid, we have tried to implement the method of Plante et al. 2020.

Unfortunately we have not been able to get rid of the checkerboard instabilities, we see two possible reasons for this:

- A/ we have not done it properly

- B/ the numerical scheme of SI3 used to solve the equation of momentum is explicit (i.e. $u^{t+1} = f(u^t)$, apart from the use of $u^{t+1}$ in the ocean-ice stress term), whereas with the McGill model, Plante et al. 2020 used an implicit solver (i.e. $u^{t+1} = f(u^{t+1})$), an approach that is more sophisticated and more numerically sound.

Even if we would have been able to correctly implement the method of Plante et al. 2020, we would have chosen to go for the E-grid approach for the following reasons:
- i) the damage variable at the corners of the grid cells is never averaged, so $\sigma_{12}$ does not use a smoothed damage while $\sigma_{11}$ and $\sigma_{22}$ use a non-smoothed damage
- ii) we can advect all the components of the stress tensors using the exact same numerical scheme as that used for all the tracers
- iii) we obviously liked the idea of testing a new approach

Reasons (i) and (ii) are now clearly stated in the new version of the manuscript (end of section 3.2). We also added some discussion on why we think it is important to include the advection terms of the stress tensor components in a brittle rheology that relies on a damage tracer (section 2.2, L159-168).

"Cross-nudging" is a new filter introduced in the MS and this seems to be a crucial part of the E-grid it requires more attention. The authors (l263)) "conclude that the right compromise is achieved when gamma_C typically lies between 1 and 3, with 2 being the value used in our experiments".

This is a central issue. In order to avoid the staggering issues of the C-grid, a new issue is introduced: the E-grid solutions diverge and need to be coupled explicitly. The coupling parameter is found by trial and error (fine with me), but there is no reference as to which metric is used other than "the right compromise [between smoothing and coupling]", which is somehow related to Fig5. What is this "right compromise"? I need to know if I want to, as suggested, reject the C-grid (without attempting to fix the issues on it) in favour of an E-grid with different issues that need to be fixed. "spatial consistency", "smoothness" (or rather the absence of it) are more soft metrics, which I cannot evaluate. I do not see a qualitative difference between 5b and 5c, but 5d appears even smoother hence it is rejected. The structure in 5a (no cross nudging) shows the checkerboard like patterns that Plante describe, but I don't see a figure with gamma_C below 1 (but >0), where the solution is supposed to become increasingly noisy/

We think that we have now significantly improved the parts dealing with the cross-nudging (CN), providing more discussion behind its logic (section 2.3.2).
With respect to the first version of the manuscript, the method has also been improved after realizing that the CN should be applied on the vertically-integrated stress components (i.e. h$\sigma$) rather than on $\sigma$ alone (little reasoning with a pen and a sheet of paper easily demonstrates that not using h$\sigma$ in equation 16 may introduce errors in regions with strong gradient of ice thickness). Some discussion has also been added about this point.

Also, as you suggest we have added more maps showing the effect of using different values of the CN parameter. Since we now use a value of 1 (rather than 2 as in our old simulations) we show figures for values: 0, 0.1, 0.5, 1., 2., and 10.

We make clear, in the discussion about this CN parameter, that it is up to the user to choose their own value, as a tuning parameter, but we also indicate that based on the various experiments we have performed, and as suggested by the new figure 4, the value of the CN should be of the order of 1. (L302-314)

For choosing gamma_C, I only see the possibility of generating some reference (observational data? NextSIM output?) and somehow define a "goodness of fit" to this reference data when you tune the cross-nudging parameter. It would be very interesting to see, if one couldn't achieve something similar with a C-grid and some averaging as done in Plante et al (2020).

Yes, it would be interesting to do what you suggest, yet we still have hope to find a method more elegant than the cross nudging, and more physical, or at least more numerically sound. We think that the message we convey in the new version of the paper clearly underlines the fact that the CN is an approach that is quite *ad-hoc*, but an approach that has the benefit of demonstrating that the E-grid approach works.

There are many judgemental statements that support the general "salesman" tone of the manuscript, for example,
- l204: "To avoid the problems related to the staggering of the C-grid"

- l234: "Thanks to"

that suggest that something is problematic, or better or worse, without any support in the text (or proof).

The language is often sloppy, there are many unnecessary repetitions. Often it sounds like listening to an informal talk about the subject (where the language would be OK to my mind). There are a few grammar problems, and many places where the formulations could be made much more concise (by removing unnecessary words and phrases).

We apologize for the sloppy language. As for the repetitions and grammar problems, we tried to remove them in the revised version of the manuscript. We also definitely tried to follow the reviewer's recommendation of making our text more concise. And again, we have done our best to strip all the judgmental/negative wording off the text.

Smaller problems, typos, technical issues:

Abstract:

Taking into consideration both the major and abstract-specific remarks of the reviewer, our abstract has undergone a substantial re-writing.

page 1

l2 new spatial discretisation framework
The E-grid is only new to NEMO, rephrase
This sentence no longer exists.

l3 well adapted to solve the equations of sea-ice dynamics
What is "well adapted" in this context. Can be removed
This sentence no longer exists.

l3: the numerical issues posed by the use of the staggered C-grid.
What are these issues?
This sentence no longer exists.

l6: "when using the newly-implemented BBM rheology and when"
grammar? Main clause is missing
This part has been completely rewritten.

l8: "the relevance of the use of this newly-implemented rheology for future modeling":
Awkward, rephrase, no need to emphasise the usefulness of the present work.
We agree, we have completely removed this sentence.

l10: "This includes, in particular, coupled climate simulations performed with CMIP-class Earth System Models at coarse to moderate spatial resolution.": There is no information in this sentence.
Same here, sentence removed.

Introduction:

l14 Sea-ice is one of the most important physical interfaces

Not sure if sea-ice can be reduced to an "interface" (especially since this "interfaces" uses about 50% of the computer time, Table 3)
"interface" has been replaced with "component". (L13)

page 2

l24: "the abundance ...": Please add some references

We now have added the references to both the IABP (buoys trajectories) and RGPS (satellite-derived ice trajectories) datasets (L24-25).

l34: "Following the work of Girard et al.": I do not think that we need the history of brittle models again.

We removed this part and now get directly to the point (L34).

l45: These two constraints have proved to be impossible to respect with MEB because of an incomplete treatment of the convergence of highly damaged sea-ice, which results in unrealistic sea-ice thicknesses after a couple of years of model integration.

Does that mean, that MEB, while always being superior to other models (according to cited references), cannot even get the fundamentals right? Also, how much of this can be attributed to MEB, and how much to the specific implementation in neXtSIM?

Our answer, starting with a short historical recap:

This limitation of the MEB rheology, as introduced in Dansereau et al (2016) and implemented into neXtSIM in Rampal et al. (2016), was not documented in these two papers simply because it was at that time tested in the context of winter simulations only, typically for the Jan-Apr time period. Over these relatively short winter time periods, the bias in the thickness of the ice pack cannot be clearly seen. This is only after running the model for several years that the bias was discovered, and its origin identified/interpreted in terms of missing physics. This motivated the development of the BBM rheology to fix this problem, among others (Olason et al. 2022).

We can confirm to the reviewer that this issue should not be attributed to the implementation in neXtSIM, but to the lack of any ice resistance to converging motion when the ice concentration is close to 1 and the ice is highly damaged.

l49: pure -> purely

The sentence is no longer present.

l56: "excellent": What does make the scalability "excellent"? Reference? I think the use of superlative adjectives needs to be re-considered.

While one could argue that the scalability of NEMO is "excellent" with respect to that of neXtSIM, we agree that the adjective "excellent" is too strong in a broader GCM-related context. Therefore, we have replaced "excellent" by "good" and added the reference to Tintó Pims et al. 2019 (L55).

In the same sentence, we have also replaced "SI3" by "NEMO", because mentioning SI3 rather than NEMO was an inaccuracy from our side. SI3, just as its predecessor LIM3,

entirely relies on NEMO's MPI horizontal partitioning, and cannot be run in a standalone mode outside of the NEMO environment.

page 3

l60: double ")"
Corrected.

l74: In this paper, we propose a solution to this problem and provide a detailed description of the implementation of BBM into 75 an Eulerian, finite-difference, staggered-grid modeling framework; namely that of SI3, the sea-ice component of the NEMO modeling system.

There is a lot of repetition of previous paragraphs in this paragraph. It may be worth it to try to streamline the introduction to avoid unnecessary repetitions.
We have done our best to do so.

l84: some important aspects

It's always good to "discuss some important aspects", but what are they? Rewrite to be more specific.
The "important aspects" are now clearly stated in our new sentence:
"In section 4, we discuss some numerical aspects of our implementation and some limitations of the BBM rheology." (L81)

page 4

eq(1) sign error in Coriolis term? Unless \vec{k} points downward.
It was an error, well spotted. Corrected.

l91: Appendix A1

It is tedious, but I think the notation needs to be introduced where it appears (on top of Appendix A1, to which I do not always want to refer, when reading the manuscript)
The notation is now introduced after Eq. 1, and other equations. (L86)

l92: writes -> is written as
Corrected. (L91)

l98: where the underbar notation indicates that the tensors are expressed in their pseudovector form, and K is the elastic stiffness tensor

I guess "K" is also in its pseudovector form? Why no underbar? The pseudo vector form is also called Voigt notation, maybe add to make it clearer to more readers.

Yes, K is in its pseudovector form, we have added the underbar. And we now use the term "Voigt form" rather than "pseudo-vector form" throughout the text. (e.g. L99)

l109: I don't think it "happens to differentiate", but it

differentiates, also since it is specific to BBM, this is a bit of repetition. Please rewrite.

The sentence has been rewritten as follows:
"The BBM constitutive equation (4) only differs from that of MEB through the inclusion of the term $\tilde{\mathbf{P}}$." (L106)

page 5

l110: the excessive convergence of ice when damaged

I am sure that the physical motivation for the form of this term (eq8) is described at length in Olason et al. Still it would be a courtesy to the reader to repeat the reasoning here, because it appears to be so fundamental to BBM. Otherwise it just appears to be a quick fix to solve a severe MEB problem.

We've added new information summarizing the physical motivation for this term (L106-110).

l114: Ólason et al. (2022) follow Dansereau et al. (2016)

That's nice of Ólason et al., but what do you do in this paper?

We changed the sentence to "We follow Dansereau et al. (2016) and Ólason et al. (2022) in using a two-step approach to solve equation 4." (L116)

Also, what is a "two-step approach" in this context? I guess this follows in after l115, but it's not clear from this sentence.

The next sentence now starts with "As the first step, …", which helps the reader understand that we are explaining what these two steps are. (L116)

Don't get me wrong, I am fine with omitting details and referring to previous papers for them, but here the mix is strange: Many (all?) details of the equations of the model are repeated, but some steps in the solution method are omitted. As a reader, I would be

fine with saying: The BBM model is described in Olason et al, we do everything in the same way, please refer to Olason et al. Or put all or most of this into the appendix, as it is not new. The only information that I need as a reader is the stress tensor and its discretisation (according to the introduction and abstract).

Thanks to your feedback and some reshaping, we think this part is now easier to follow in the new version of the manuscript and we wish to keep it.

l127: "In the case of the BBM framework, Ólason et al. (2022) and the damage criterion shown in Fig. 1 and dcrit expresses as follows"

Something is wrong with this sentence, please fix

The sentence has been simplified to "In the BBM framework, $d_{crit}$ is expressed as follows:" (L128)

l132: "for the healing the ice": for the healing of ice OR for healing the ice

For the healing of the ice. Corrected (L133).

l132: which is associated with refreezing within open leads and which is therefore based on a rate of decrease of the damage that depends on the temperature of the ice.

Please rewrite to disentangle the various relative clauses and the not always correct usage of "which" and "that".

We have rewritten the two following sentences instead (L133):
"As suggested by Rampal et al. (2016), a slow restoring process is applied to the damage to account for the healing of ice under refreezing conditions. The rate of decrease of the damage associated with this refreezing is taken proportional to $\Delta Th$ , the temperature difference between basal and surface ice:"

page 6

l145: "which" Start a new sentence, too many relative clauses

We now start a new sentence (L144).

l146: "In non-regularized frameworks" Not clear what this means in this context. What is a regularised framework, in contrast?

A regularized framework would be one in which we would introduce, via the constitutive equation for instance, a characteristic length scale for the process of interest: here, deformation (or damage). In such a framework, deformation (or damage) would localize at this characteristic length scale. We do not introduce such regularization in our model and in sea ice models in general. One good reason is the fact that no characteristic length scale for the deformations seems to stand out in the observations, over a wide range of space scales. Even if we would consider a mean, characteristic width for leads, this length scale would be much smaller than the model resolution.

The mention of the term "non-regularized" was not really necessary in the context, and confusing, therefore we have rephrased this sentence as follows:

"The combination of elasticity and damage, even if treated in an isotropic manner, naturally simulates a strong anisotropy and localization of the deformation, down to the nominal spatial and temporal scale…" (L146)

l150: tend to exhibit very sharp gradients, or "near-discontinuities"

It needs to be shown that this is specific to brittle models. The neXtSIM MEB papers always imply this, but e.g. the fields in Plante et al (2020) are generally much smoother. Also these "near-discontinues" also appear in high-resolution VP simulations (e.g. Ringeisen et al 2019, 2021).

We removed the term "near-discontinuities" from the sentence (L149-150)

L152 remove the ","

This sentence is no longer present.

l159: "In BBM …"

This seems to be a specific issue with the specific BBM implementation of Olason. In theory (Dansereau et al 2016 describe an iterative procedure with a tolerance, their page 1350, rhs column, Plante et al update damage, E, \lambda during an iteration), it should be possible to iterate the "two-step approach" until convergence.

As noted by the reviewer, Dansereau et al. use an iterative approach to solve the MEB problem. This is not the choice made by Olason et al. . In order to fix this inaccurate statement in which we were opposing the brittle rheologies in general to the viscous-plastic ones. This part have been heavily re-worked and instead at the end of the section 2.2, we have a more developed about the time splitting approach (starting L169):

"Finally, note that in their numerical implementation of BBM, Ólason et al. (2022) chose to solve the dynamics explicitly using a time-step sufficiently small to account for the propagation of damage in the ice in a physically realistic manner."

This discussion about details of the time stepping and convergence is a little akin to the EVP evolution, where the somewhat naive iterative process lead to a (noisy) solution

that was not intended, until papers like Lemieux et al 2012, Boullion et al 2013, Kimmritz et al 2016 came up with a solution for this (revised, modified, adaptive EVP).

We agree.

page 8

l204: To avoid the problems related to the staggering of the C-grid, namely the interpolation

I strongly suggest avoiding this type of judgemental phrasing here and elsewhere, write instead: To avoid the interpolation of … due to the staggering of the C-grid …

I am pretty sure that the E-grid is not without issues, and I am waiting for similar statements about the "problems related to the E-grid".

We have modified the sentence accordingly, it now reads:

"To avoid the interpolation of the damage and the stress components between the center and the corner points of the grid cell, and allow the consistent advection of all the components of the stress tensor, an additional sea-ice velocity vector, noted $(\hat{u}, \hat{v})$, is introduced." (L225)

l209: Arakawa E-grid

BTW, there was a successful ocean model that used the E-grid: The Hamburg Large-Scale Geostrophic (LSG) model (by Ernst Maier-Reimer). Here, the E-grid was chosen, because the dominant balance (geostrophy) can be expressed more accurately and without noise while retaining some of the "nice" properties of the C-grid (representation of divergence). Maybe that's an analogy worth mentioning.

Absolutely, we were not aware of this and we thus thank the reviewer for sharing the reference. We have added a reference to Maier-Reimer et al., 1993 when first discussing the E-grid. (L230)

In section 2.3.2, that deals with the problem of the decoupling of the 2 solutions with the E-grid, we also added a couple of sentences mentioning the reason behind the choice of this grid in LSG and how this decoupling, which primarily affects "short gravity waves" is not a problem based on the large time step of the model, and how their use of "moderate" viscosity and diffusion prevents the "split-modes" for the slow modes that are resolved. (L264)

page 9

L234 "Thanks to" -> "With"

Again this is judgemental, this time in the "positive" sense. Statements like these set a certain tone that appears biased and scientifically non-objective.
Done (L255).

l237 "and no interpolation is needed to solve the equations" But this advantage comes with the disadvantage of the cross-nudging
Yes. The following sentence starting with "It does, however, result in an …" we think the reader is made rapidly aware that the E-grid approach is not perfect (L266).
We also mention this in the conclusion (L641-L644).

l248: upper-convected time derivative

Later this term is introduced properly (l289), maybe do it here already. Or just use "advection" for simplicity here and introduce the upper-convected time derivative later.
We have completely re-written this part. (L270)

L259 and eq16: denoted by interpF@T and interpT@F,

As this is a math expression, why not use more "math-like" symbols to denote the interpolation, e.g. $\overline{\hat{\sigma}_{11}}^T$, where $\overline{...}^T$ means interpolation to T points, etc.
Equation completely re-written following your advice (L283).

page 10

l264: Don't refer to figure 5 before figure 4?
Figure ordering should be fine now.

l266: horizontally and vertically aligned with the grid cells. -> aligned with the grid
There is no "vertical" in a 2D horizontal grid.
The corrected sentence is now: "... that are aligned along the x- or y-axis of the grid." (L308)

L289 upper-convected time derivative

It would be good to add a reference here, as this terminology of complex fluids is probably not common knowledge of the GMD reader.

After reading up on this (in a book about polymer flow!) it is not even clear, why we have to use the upper-convected time derivative, and not the lower-convected time derivative or a linear combination of the two. In Danserau et al (2016) something similar is called the Gordon-Schowalter derivative (it's not the same but some linear combination of the

upper and lower convected derivative), and obviously it is not entirely clear what is the correct form to use, as any frame-invariant time derivative is formally allowed. So some discussion with appropriate reference seems in place here.

Section 2.4.1 has been largely re-written taking into account your suggestions, it now includes:
- appropriate references
- discussion about the existence of the 2 formulations (upper- and lower-convected), and about the resulting dilemma faced by the modeler
- inclusion of the equations for the lower-convected formulation in the manuscript, that we have also coded in our implementation to perform sensitivity experiments

We also added a paragraph to justify our choice to use the upper-convected form in our simulations at the end of section 2.4.1. Also a new figure (C3) in Appendix C, at the very end of the manuscript, shows the impact, on the PDFs of simulated deformations, of using or not these two different formulations.

Eq17 and 18 do not describe the upper-convected time derivative. WIth L in eq18 and a plus-sign in Eq17, this would be LOWER convected time derivative.

The component form (eq19) is correct (and consistent with the form of L = (\nabla\vec{u})^T \cdot \sigma + \sigma\cdot\nabla\vec{u}) for the upper convected time derivative)

Eq18 would give components with \partial_y U and \partial_x V exchanged (w.r.t eq19).
Yes, we have corrected our mistake, and now equations for both lower- and upper-convected time derivatives are provided (Eq. 18,19,20,21).

page 11

l299: Then, the tensor-specific contribution −L is added.

Is this done successively, I.e. use the sigmas after advection with the material derivative D/DT to compute L (i.e. some sort of split operator method), or do you use the sigmas before applying D/DT? Please be more precise.

Thank you for raising this point, it helped us find a bug: we were wrongly using the sigmas that had already undergone the material derivative update!
This has been corrected in the new version of our implementation and all the simulations discussed in the manuscript have used the correct version of the code.
So thanks to your input, we can definitively write that:
L is computed using the stresses that have not been updated by the D/DT yet. It is indeed important to mention this, and the text has been modified accordingly. (L358-359)

l305: It largely inherits from LIM3 Rousset et al. (2015), to which it succeeds

Please simplify and fix in-line citations.
We have removed this sentence. Just as you point out below, this information is not important for the paper.

What are "significant inclusions"? Are they important for this manuscript? If not I wouldn't mention that (again, we do need a full history of the model components), if important, then we need more information.
Sentence removed, see previous reply.

page 12

l324: "We carried out a twin coupled ocean/sea-ice hindcast,"

That is a lot of words for just saying "we compared two simulations"
This sentence has been removed (see beginning of section 3.2)

l329: "while", wrong connector, -> and
This sentence is gone (re-working due to new spinup strategy).

l331: For the second spin-up segment,

Why does SI3 need this "re-initialisation"? The model should be more or less in balance with the ocean state.

Also the initialisation is short. In my experience, a sea ice model needs some 2-3 years to spin-up (the ocean model much longer, but that's not really necessary for this paper).
Based on your remarks and the concern expressed by reviewer #3 about our original spinup strategy, we have chosen to use a simpler, yet shorter, spinup strategy and the two simulations have been run again (see the introductory part about the major changes undergone by the paper at the beginning of this document).
We agree that it would take years of simulations to spinup the Arctic ocean in our model, but, as you underline, based on the scope of the paper, we think that it is not a problem to use a short spinup as we do. Note that we initialize the model using the 3D ocean and sea-ice data of a reanalysis (GLORYS2) that used an earlier version of NEMO on almost the same numerical grid and bathymetry (global 1/4° ORCA025 configuration).
See beginning of section 3.2 for details about the new spinup strategy.
We have added a sentence to stress that while our spinup is much too short to obtain a spun up ocean circulation, its duration is acceptable for the scope of the paper (L482).

l333: were extracted from a coupled OCE-neXtSIM simulation

Why do you need the solution from a different simulation to restart the model?
As mentioned in the previous reply we now use a different spinup strategy, and no longer use fields from a different model/simulation. Also see the introductory part about the major changes undergone by the paper at the beginning of this document.

l335: a duration sufficiently long for the coupled system to recover from the ad-hoc reinitialization.
I doubt, that this is long enough. The fast waves will have left the domain, but everything else ...?
We agree, see our two previous replies.

l339: the tuning of SI3 is kept as close as possible to the default namelist

rewrite: tuning is a process and you cannot kept a process close to a namelist.
Our new sentence is: "For these experiments, the adjustable tuning parameters of SI3 are kept as close as possible to those of the reference configuration of NEMO." (L489).

l340: "thermodynamics features"; Grammar?
The sentence is now: "As such, the thermodynamic component uses 5 ice-categories." (L490)

L343 :Table A1 in appendix C.

There are many problems: Why is called A1 if it belongs to section C? Section C1 is one sentence, and is not needed. Table A1 does not contain any parameter values, only descriptions; according to SI standards, units should not be in brackets "[]"; especially in this context the brackets make no sense (same for Table A5).
Section C1 has been removed. The table in question (old table "A1") has also been removed because it is not needed anymore: experiment SI3-default is now 100% in agreement with the default setup of SI3. The fix of the "ocean-ice drag" bug when switching from NEMO 4.2.1 to 4.2.2 (see the introductory part of this reply) allows to use the default ice-atmosphere drag coefficient of 1.4 in SI3-default, in place of the previously used value of 1.15, with the same results when it comes to the PDFs of deformation. And we are now using the default number of aEVP iterations in SI3-default: 100 (180 in previous version).
All brackets surrounding units have been removed from the text.

Paragraph l339 to l347 could much clearer, including the extra information, supposedly in tables in the appendix. The tables could be in the main text, so that the reader does not have to flip back and forth in the paper (from here to appendix C, then back to Table

A1, etc.). Since the time-splitting was already introduced, no need for phrases like "As mentioned in … " (better: For the time-splitting approach (Section 2.2), we use a small timestep of …)

Reference to appendix C and table A1 no longer exist (see reply to former remark). There is now Table 1 that lists SI3 default parameters relevant to both experiments SI3_default ( in particular those related to the aEVP rheology) and SI3-BBM. And there is Table 2, that lists the BBM-related parameters used in SI3-BBM and their value. These tables are no longer to be found in an Appendix C but in the text.

We reckon that the changes and simplifications made to this paragraph, based on this point and the previous one, have made it clearer.

The Copernicus Latex template for GMD encourages authors to gather Tables and Figures at the end of the manuscript so we keep it this way.

Section 3.3 should be part of a data and methods section. Now it is strangely split between the model evaluation section and the appendix. It would be much nicer for the reader to have everything in one place.

Everything is now gathered at the end of section 2, in the new subsection 2.5, and Appendix C is now used to present some additional figures.

L351 "(RGPS hereafter)" Unnecessary; it's enough to introduce the abbreviation RGPS earlier in the line.

Done. (L375)

• Highlight, page 13

?

l358: "(see the Code and data availability section." Closing ")" is missing

We have removed this sentence.

L366 in the literature

I guess it's fair to cite Ron Kwok's paper about this.

We added the reference to Kwok, 2001. (LX)

L367 quite realistic

What's the meaning of "quite" in this context?

We have replaced "quite realistic" with "appear somewhat realistic". (L500)

L368 very smooth fields of deformation with no such localized features

Definitely true, but the figures, where the quadrilateral data is plotted on triangles with gaps in between makes it very hard to read the figures. For example, the aEVP solution seems to be noisy, but I cannot tell if this is an effect of the plotting. Convergence is an issue with aEVP (it's very slow) and one can only expect "smooth" fields at all times if the aEVP parameters are tuned properly, see Kimmritz et al 2016. The smoother the solution, the slower the convergence.

We can assure the reviewer that what he sees as noise is just/and only the result of the plotting using the mask defined by the presence of RGPS observations. There is thus not much we can do to prevent these gaps in these figures, because our deformations are calculated using quadrangles, which, based on our construction process (see new section 2.5.2), tend to exclude some "left-over" triangles that were not merged into quadrangles. Some "empty" space is thus left in between.

For your information, here is a snapshot of the (instantaneous) total deformation (computed online with the *Eulerian* velocities), at roughly the same period, for the two simulations.

L369 consistent

I am not sure, if you can say that, because there are also "coarse" (ie. 10km) runs in Bouchat et al 2022 the have quite some LKFs (their Fig10, the McGill model with smaller e). EVP models tend to have fewer LKFs in that paper (see previous comment about convergence), there's even a comment about EVP models, convergence and deformation rates in Bouchat et al (their section 4.1.1)

It is true that the number and other metrics for LKFs is affected by different model parameters, including the *e* parameter. Our point here is that our aEVP solution is quite comparable with what other people get when using standard (or commonly used) parameter values. We have, therefore, rephrased this sentence as: "... this is consistent with the findings of recent studies that evaluate VP-driven sea-ice simulations run with a horizontal grid size larger than a few kilometers and standard parameter values."

page 15

L429 "A critical requirement for the consistent implementation of the brittle rheology"

This is phrased as a well known fact, whereas this is just what the authors find. An ill-meaning reader could conclude: The authors did not manage to succeed in stabilising the model on a C-grid. Please tone down these statements.

This sentence is no longer present.

L434 The "Leap Frog scheme" may be a good analogy, but by the same analogy, the leap frog scheme is very much outdated and more stable schemes are commonly used in

general circulation models nowadays (e.g. 2nd or 3rd order Adams-Bashforth in FESOM, MITgcm, ROMS, 2nd order Runge-Kutta in MOM6). I am surprised to learn that NEMO still uses a Leap Frog scheme.

We agree. The NEMO development team has been working on the implementation of the RK3 scheme for the last few years. RK3 should normally become the default in version 5 of NEMO, which should be released before the summer.

In this light, introducing yet another filter like the Asselin filter does not "serve[] a useful purpose" (l441)

The sentence has been changed to:

"As of now, our cross-nudging approach clearly lacks physical and numerical consistency, but it somehow allows to demonstrate that the implementation of a brittle rheology, along with the advection of the internal stress tensor, is feasible onto an E-augmented C-grid." (L583)

page 16

l462: that -> which

Corrected. (L596)

l464: When SI3 is coupled to OCE, however, the cost increase is somehow dissolved by the overwhelming cost of OCE and falls below 30%.

It is interesting to note that other groups find that the sea ice dynamics, especially at high resolution, can become the most expensive part of a sea ice-ocean model (e.g. Koldunov et al 2019, doi:10.1029/2018MS001485). I do not share the relief, that the sea ice model does not get "that much" more expensive when coupled to an ocean model. The cost increase definitely does not "dissolve".

In NEMO, the relative cost of the ice component is a bit influenced by the number of vertical ocean levels used by OCE, but with the NEMO reference number of 75 levels (we use 31) the cost of SI3 is reported to be of typically 40% (Clément Rousset, lead developer of SI3, personal communication). This number has increased with the years as more advanced parameterizations, categories, etc, have been added in LIM and then SI3.

Now about our use of the term "dissolved" (that we have removed) and the fact that the numbers are smaller in coupled mode: in regular ocean/sea-ice coupled mode, NEMO couples the ocean and sea-ice modules in a sequential, and not in a parallel, way. This means that the cost of SI3 simply adds up to that of OCE.

The way we formulated this was confusing and we have rephrased the sentence:

"This lower value is explained by the fact that by default, the coupling between OCE and SI3 is done sequentially. As such, the cost of SI3 simply adds up to that of OCE, and the cost of

OCE is expected to be independent of the mode used (in our case: 113 and 114 cpu h for SI3-default and SI3-BBM, respectively)." (L607)

Also, based on a remark of RW2, these "performance" diagnostics have been performed again with the default value of $N_{EVP}$=100. Also note that in all our new BBM simulations (including those on which the deformation analysis is performed) we now use a number of sub-time-steps of $N_S$=100 for the time-splitting, and not $N_S$=180 as before; because based on sensitivity experiments that we have performed during the course of the review we have come to the conclusion that $N_S$=100 in our BBM implementation is largely sufficient. So now, both aEVP and BBM simulations perform 100 subcycles under one advective time-step, in all the results presented. In that regard, we have also added two new figures in the new Appendix C (figures C1 & C2) that compare the cyclone test-case results for aEVP and BBM using more sub-cycles, $N_{EVP}$=1000 and $N_S$=200, respectively.

The numbers in Table 3 tell me that the ocean model uses 157/159 cpu h in this configuration (not sure where the difference of 2h between the setups comes from), the sea ice model 139 or 223 cpu h, so that with BBM, the sea ice model already uses more than 50% of the total time. Even the 139 cpu h of aEVE appear long in this context (47% of the total run time). That's where Koldunov start to worry about overall performance. I think that this needs to be discussed in more general terms, i.e. how much time to allocate to a small part of our coupled model.

Please see our reply to the previous point, and note that these numbers have been updated due to modifications mentioned in our previous reply.

The cost of the ocean component is now included in the new version of our Table 4.

We will not discuss what you suggest as it is quite out of the scope of this paper, and it is something that should be addressed by the NEMO development team. We plan to work together with them on this matter when merging our implementation into a dedicated branch of NEMO, which should be done before the end of the year we hope.

L472 "an insufficient number of iterations"

aEVP was designed to lead to smooth solution even when the solver is not fully converged. If there are checkerboard patterns in the solutions, then the choice of aEVP parameters is poor and should be improved ("In practice, the value of $c$ depends on forcing, geometry of boundaries and on resolution and has to be selected experimentally", Kimmritz et al 2016).

Since we now use the default $N_{EVP}$=100, and based on the fact that a new sensitivity experiment we have performed with $N_{EVP}$=500 would show the same results (for the diagnostics presented in the paper, see for instance Figure 2 at the end of this document) as the one with $N_{EVP}$=100, we have removed these 2 last sentences of section 4.1.

We also show the impact of having $N_{EVP}$=1000 instead of $N_{EVP}$=100 on the solution of the cyclone test-case (figure C1 in Appendix C).

l477: like for instance about the Arctic sea-ice thickness distribution.

Please rephrase. Also, what are "promising results"? I think it would serve the manuscript, if this were formulated more concisely.

We have completely rewritten the sentence in a more concise way:

"Based on comparisons against various types of observations, recent studies suggest that large-scale models using BBM can realistically simulate the dynamics and properties of sea-ice (Ólason et al., 2022; Rheinlænder et al., 2022; Boutin et al., 2023; Regan et al., 2023)." (L613)

If this were my manuscript, I'd rewrite the entire parargraph along these lines:

Large-scale realistic sea-ice simulations with a model using the BBM rheology showed encouraging agreement of, for instance, the Artic sea-ice thickness distribution with observations (see also, Ólason et al., 2022; Boutin et al., 2023). Yet, deformation in convergence and sub grid-scale processes related to sea-ice ridging are not represented by BBM with the same degree of accuracy. The model overestimates the number of converging events with magnitudes of about 1 to 5% per day, and underestimates, although not as much as the aEVP solution, the most extreme events (Fig 7c, and Ólason et al. (2022). So far, parameter tuning, in particular the BBM-specific ridging threshold parameter $P_{max}$, did not help to improve agreement with observed convergence PDFs (not shown), so that we conclude that some fundamental processes need to be reconsidered in BBM (and aEVP) [ or now some other educated guess/speculation about the BBM (and maybe aEVP) equations that makes it impossible in principle, to get the convergence right ].

Thank you! This was helpful. Apart from the first sentence, the paragraph has been re-written accordingly with only minor modifications. (L613-620)

page 17

L488 "are doing better in this particular matter" , Rephrase: "agree better with observations" or similar.

The sentence has been shortened and re-written:

"The fact that the deformation fields simulated by neXtSIM in Ólason et al. (2022) are in better agreement with RGPS in this regard, suggests that this problem is linked to some numerical aspects of our BBM implementation rather than the BBM rheology itself." (L622)

l490: best -> most likely (and remove the following "likely")

Done. (L624)

L491 "as these steps are absent"

Rephrase, "numerical dispersion and diffusion" are not "steps".
The sentence has been improved:
"This is most likely the consequence of the introduction of some additional numerical dispersion and diffusion by the advection scheme and the cross-nudging treatment, respectively, as these two features are absent in neXtSIM." (L624)

Also you could test that by using other, more diffusive/less dispersive advection schemes.
Yes we should test this at some point. The Eulerian version of neXtSIM that we are currently developing (neXtSIM-DG) uses the Discrete-Galerkin method for the advection, in this regard, it will be interesting to compare BBM simulations run with neXtSIM-DG and SI3.

But what about the cross-nudging process? That was not part of neXtSIM either and could very well be a likely candidate for differences.
Definitely. The cross-nudging was indeed already mentioned in the 1$^{st}$ version of the manuscript (see old L91). It is still mentioned in the updated version, see re-worked sentence above.

L495 "relevant" not the right word here. -> promising?
Yes, promising it is, corrected. (L629)

Conclusions:

In general, I don't think that the tone of the conclusions is appropriate (see main points). As if there is only this solution and everything else is wrong from the beginning. Many conclusions are drawn from trial and errors (as described here and in the text). I think it is good to show and discuss failures, so that others do not stumble into the same problems again, but there needs to be a more systematic list of things that did not work and why. The way this manuscript is written, I get the impression, we are presented with the result of something that finally works in spite of all the hardships encountered along the way, told by the fireside.
Based on your remarks, the conclusion has been significatively reworked. We also put more emphasis on the "ad-hoc" aspects of our implementation, such as the cross-nudging, and those that remain problematic.

L501 The use of the Arakawa C-grid, as used in SI3, has proven to be poorly fitted for brittle rheologies.

The paper does not show this. It states that there are fundamental issues related to the staggering of grid points (but that is a problem not only for brittle rheologies) and describes a method to overcome noise issues that were not even demonstrated. By no means there is "proof" that it cannot work on a C-grid.

You are right. This part has been largely re-written taking your remark into consideration.

"poorly fitted" -> I don't think that the C-grid has been "fitted" to anything.

Gone with the re-writting.

L505 This approach prevents the numerical schemes at play in the rheology

What "numerical schemes"? A numerical scheme is designed by a person (or soon by AI), but is not "at play in the rheology". The paper does not present any stability analysis and or show development of noise due to numerical instability, grid staggering or whatever, so all you can say that with your scheme you are able to suppress any noise that may appear.

Yes, you are right. This sentence is no longer present.

L512 deformations -> deformation statistics
Corrected. (L655)

L513 "with respect to the viscous-plastic rheology", "the aEVP simulation with admittedly not properly tuned parameters (reported checkboard noise)".

The part about the checkerboard noise in the aEVP of SI3 has been removed. We use the exact default and recommended setup of SI3, which is the result of years of collaboration between the NEMO development team and the Sea-ice Working Group of NEMO. As such, we cannot do better than moderating our conclusions, and making it clear in the text that our reference simulation is the workhorse setup as provided by NEMO and not the most "appropriately-tuned aEVP". This is also why we have changed the name from "SI3-aEVP" to "SI3-default".

This particular sentence has been re-written, and aEVP is not mentioned anymore:

"Based on a comparison with satellite observations, this analysis demonstrates that the use of the newly implemented BBM rheology results in simulated sea-ice deformation statistics that are realistic." (L654)

The sentence should include, that this happens on the same grid, with the same grid spacing etc.

Not relevant anymore in the re-writtten sentence.

The aEVP solution also has LKFs, just very few.
Yes, and one can also see them well in the figure showing the Mehlmann test-case that has been added to the paper (new Figure 5).

page 20

L550 "The average of the four surrounding points is used"  What is done near the boundaries?
We have added the following sentence right after eq. A1:
"Note: surrounding points located on land or open-boundary cells are excluded from the averaging." (L688)

A1 I would avoid "if \phi @F (@T)", but instead use superscripts as in A2
We have done so, but we still need to keep a "(if \phi defined @X)" after each equation.

page 22

A5 Table of symbols related to the numerical implementation

The text promise values for the parameters but there are none. Remove [] around units
Parameter values are now provided in Table 1 & 2. "[]" around units have been removed everywhere in the text.

A5: e1t, etc. why not use \delta x ^{U} for this (or similar). This looks like it's a Fortran variable from SI3. I think this can appear in the paper, but not in the list of symbols, and I would refrain from using the variable names in equations and text (as done in B2), because for a non-NEMO user/developer it makes the expressions impossible to read and check.
We agree, all the et1, etc. are now renamed as you suggest.

page 23

l565: I think that equations, here "(Eq. B5, B6)", should be introduced before they are referenced. Now I have to read appendix B2, etc before Appendix B1.
We don't agree. B1, the algorithm block, serves as a summary of what comes next, in a more detailed way, in B2. So we think it should stay this way.

page 24

B1 (@T), etc not really necessary, as the notation has defined this already
We agree, they have all been removed.

page 25

B7: it should be pointed out in the text that \bar{h} is used and not \hat{h}
Yes, we have added a sentence about this, same for the two previous equations (B5,B6) that use \bar{A}. (L748 & L752).

page 26

B12: what is "N"?
N is the upper limit for compressive stress, it is declared in the parameter table of Appendix A. It is also mentioned right after Eq. 13.

Eq B13, t_d is not a free parameter?
The damage time scale, t_d, is not a free parameter, neither in Ólason et al. (2022) nor in Dansereau et al. (2016), where it is first introduced. In Dansereau et al. (2016), it depends on the grid spacing and the speed of propagation of elastic waves - which is a free parameter. Ólason et al. (2022) calculate this propagation speed from the damaged elasticity, which is locally influenced by the damage, as well as using the local grid size. We follow the approach of Ólason et al. (2022) here.

elsewhere: (B14) is implied by B13 when d_crit=1
Yes. But d_crit can be > 1 (when the current stress state is within the Mohr-Coulomb yield envelope), and <0 under weak shearing and compression (see Eq.13).

page 28

L645 "The value of the BBM-specific parameters ..."

Unfortunately they are not listed in the table
They are now listed in Table 2.

**L660 What are the criteria for "reasonably well shaped"?**

We are now more specific, this part has been rewritten as follows:

"Triangles with an area smaller than 25%, or larger than 75% of the nominal area of the quadrangles to be constructed (i.e. the square of the spatial scale under consideration), or with an angle below 5° or above 160°, are excluded. Neighboring pairs of remaining triangles are then merged into quadrangles in order to transform the triangular mesh into a quadrangular mesh." (now in section 2, L397)

page 29

**l666: "consistent with the spatial scale of interest," what does that mean here?**

This sentence was indeed vague, we have rephrased it adding our criterion, it now reads:

"- the square root of the area of the quadrangle falls within a ±12.5% range of agreement with the horizontal scale under consideration"
(now in section 2, L405)

**L667: "almost identical"?  Aren't they within a 3day bin anyway? What does this mean in addition?**

Yes they are within a 3-day bin. But each RGPS buoy position has its own time-position, with an accuracy at the level of the second. And these individual time positions can slightly differ between neighboring buoys, in particular when considering large quadrangles, typically when dealing with scales from 40 km and above. We tolerate a time gap of 60 seconds between the vertices of an aspiring quadrangle, hence the initial use of "almost". We have rewritten this sentence that was indeed confusing in a more accurate way:

"the time position of each of the four points defining the vertices of the quadrangle should not differ from that of any of the other three points by more than 60s"
(now in section 2, L408)

**L678 To prevent computational waste**

**I don't think that's an appropriate term here. "To save computer time/resources" ... or similar**

We now use your suggestion with "resources". (now in section 2, L419)

**L681 "feeds on"**

**I don't think that an algorithm can "feed on" something (animals do)**

After consulting fellow scientists whose mother tongue is English, it turns out that it is completely acceptable to write so in a scientific paper. We have changed "algorithm" to "software" though. (L422)

L685 please check the language of the author contributions for grammar, use of vocabulary …

This part has been re-written properly. (L796)

[Figure]

*Fig.1/ Figure 7.c of manuscript, total deformation for experiment SI3-default, ($N_{EVP}$=100).*

[Figure]

*Fig.2/ Same as Figure 7.c of manuscript, but for SI3-default using $N_{EVP}$=500 instead of $N_{EVP}$=100*

---

## Author Comment (AC6)

Well crafted and relevant paper by the authors, it was very nice to see how well the BBM rheology compares to the satellite data for the LKFs. The PDF analysis is another good results that shows the BBM does a good job in describing deformations.

We thank the reviewer for reviewing our paper and raising interesting questions, and also for suggesting the inclusion of the idealized "cyclone" test-case. We have implemented this test-case and results are now shown and discussed in the new version of the manuscript.

Before moving to a point-by-point reply format, here is some important information regarding the new simulations performed for the new version of the manuscript.

Based on the concerns expressed by both the anonymous reviewer #1 and Dr Plante on our initialization/spinup strategy, we have decided to use a simpler approach. As such, our simulations have been performed again: initialized with the new strategy (described in section 3.2 of the new manuscript), and consequently, all diagnostics, figures, and table values have been processed and generated again, based on these new simulations.

Also based on concerns raised by the reviewers, on what has been wrongly interpreted as unfairness towards the aEVP rheology, we now make it very clear, all over the paper, that what we use as the reference SI3 simulation is the default workhorse setup of SI3 as provided in the current version of NEMO. As such, for instance, experiment "SI3-aEVP" has been renamed to "SI3-default", and it is clearly stated in the paper that a better-tuned "aEVP" would likely perform better (L513-516).

Moreover, in the first version of the manuscript, our developments and simulations were based on the version 4.2.1 of NEMO. At the time, it was the current stable release of NEMO. But during the course of the review process of this paper, a bug related to the ocean-ice drag parameterization has been identified and fixed by the NEMO team. This bug has been judged sufficiently severe by the NEMO team to justify the release of a new stable version: the 4.2.2. Link to release note: https://forge.nemo-ocean.eu/nemo/nemo/-/releases/4.2.2
Consequently we have switched to version 4.2.2 of NEMO for all the simulations presented in the new version of the manuscript.
The bug fix has significantly impacted the values of the simulated deformations and has therefore required a new tuning of the air-ice drag coefficients.

However, I am not convinced that the argument has been conducted in a way that is entirely fair for aEVP for these reasons:

1. If the default number of iterations for aEVP is 100 why has that been changed to 180? That is 80% more. You mention it is for "fairness" but is it fair to make the one you are comparing against slower? As far as I understand it, with 180 iterations your framework is 60% slower than aEVP, does that mean that the comparing to the default number of iterations your framework would be around twice as slow?

Based on your remark and that of RW1, these diagnostics have been performed again with the default value of $N_{EVP}$=100. Also note that in all our new BBM simulations (including those on which the deformation analysis is performed) we now use a number of sub-time-steps of $N_S$=100 for the time-splitting, and not $N_S$=180 as before; because based on sensitivity experiments that we have performed during the course of the review we have come to the conclusion that $N_S$=100 in our BBM implementation is sufficient to give very similar sea ice deformations, both qualitatively and quantitatively. So now, both aEVP and BBM simulations perform 100 subcycles under one advective time-step, in all the results presented. In that regard, we have also added two new figures in the new Appendix C (figures C1 & C2) that compare the cyclone test-case results for aEVP and BBM using more sub-cycles, $N_{EVP}$=1000 and $N_S$=200, respectively.

2. It is also not clear why when coupling with ocean the difference in time decreases. Can you clarify?

Because in regular ocean/sea-ice coupled mode, NEMO couples the ocean and sea-ice modules in a sequential, and not in a parallel, way. This means that the cost of SI3 simply adds up to that of OCE.
We have clarified this part in the discussion section 4.1 (607):
*"This lower value is explained by the fact that by default, the coupling between OCE and SI3 is done sequentially. As such, the cost of SI3 simply adds up to that of OCE, and the cost of OCE is expected to be independent of the mode used (in our case: 113 and 114 cpu h for SI3-default and SI3-BBM, respectively)."*
We also now provide more numbers in the new Table 4, these should help the reader figure out the smaller values obtained when using the coupled setup.

3. You make a comment saying that the resolution does not change with your approach, but you are considering more than double the number of degrees of freedom. When discussing accuracy, I think you should have also compared BBM to a case where aEVP has the same degrees of freedom as BBM.

Yes, and it turns out that it is this higher degree of freedom that is actually responsible for causing our problem with the use of the E-grid, namely the separation of the solutions between the two grids, which necessitates the use of our cross-nudging treatment to prevent it. And this cross-nudging by smoothing and forcing the two solutions to remain consistent with one another does nothing else than "taming" the consequences of this increased degree

of freedom. So in short, this higher degree of freedom has some clear disadvantages and is sort of "destroyed" by the use of the cross-nudging. We now clearly state in our manuscript, that the benefit of not having to interpolate some fields between corner and center points of the grid (using a 4-point-averaging, as done in the aEVP implementation of SI3) comes with the disadvantage of having to smooth them with the cross-nudging (e.g. in the conclusion L640-646).

We think that it would indeed be very interesting to port aEVP on the E-grid in a way similar to what we have done in our BBM implementation, as we expect that this would improve the aEVP results. This, however, is out of scope of this paper that primarily aim is to focus on the BBM rheology, its implementation and the evaluation of its performance compared to observations in a Pan-Arctic realistic simulation.

4. It is great to use satellite data as comparison and the results you show are definitely very good for BBM. I think however that you should also consider a test that has been widely used by other modelers so that your results with BBM can be put into a common context and it would be easier to compare against existing approaches. The test case I am referring to is the one moving anti-cyclone that can be found for instance here: *Simulating Linear Kinematic Features in Viscous-Plastic Sea Ice Models on Quadrilateral and Triangular Grids With Different Variable Staggering*, but has been also used for instance here: *Simulating Sea-Ice Deformation in Viscous-Plastic Sea-Ice Models With CD-Grids* and here: *CD-type discretization for sea ice dynamics in FESOM version2.*

Thank you for this recommendation which enhances the validation of our BBM implementation and helped put BBM into a known context. You will see that this idealized test-case, along with the appropriate references, has been added and is discussed in the new version of the manuscript (new section 3.1, and Figures 5, C1 & C2). It is also used to show the effect of using a higher number of time subcycles in both SI3-default (known as SI3-aEVP in the previous version of our manuscript) and SI3-BBM (Figures C2 & C3).

I recommend the authors address the issues I raised here before I can advise for publication.

---

## Author Comment (AC7)

Review of:

"Implementation of a brittle sea-ice rheology in an Eulerian, finite-difference, C-grid modeling framework: Impact on the simulated deformation of sea-ice in the Arctic" by Laurent Brodeau,Pierre Rampal, Einar Ólason, and Véronique Dansereau.

This manuscript presents the implementation of the Brittle Bringham Maxwell (BBM) rheology in the SI3 community sea ice model, which uses a C-Grid Finite Difference framework. In particular, the authors present their use of the E-Grid discretization scheme to avoid difficulties with the staggered velocity and stress components in the C-Grid. They also present their use of an upper-convected time derivative approach to advect the stress components. The performance of their BBM implementation is then assessed by comparing the sea ice deformation statistics in SI3 simulations using the implemented BBM and the aEVP rheology.

I find that the manuscript is interesting, important and has potential for publication in GMD. In particular, it provides detailed discussions on difficulties implementing the BBM rheology in the SI3 sea ice model, and presents the numerical tools used by the authors to overcome them. This is an important step for the rheology to be more thoroughly investigated and used by the community. However, the methods used to validate the implementation is not adequate, and leaves doubts on the realism of the implemented BBM. In particular (among others listed below):
why is SI3 initialised from neXtSIM fields? This seems to imply that the implemented BBM is not performing well unless the damage is restarted from the other model. This method also makes the comparison between the implemented BBM and the aEVP difficult to interpret, as the BBM benefits from the initial neXtSIM damage, and thus likely has an inherited heterogeneity.
As such, while the implementation itself makes for an interesting publication, significant work needs to be done to address these validation method weaknesses. This may require changing the methods, and thus significant time to be adequately addressed.

For these reasons, I recommend this manuscript to be send back to the authors for major revisions.

Best regards,

Mathieu Plante

We thank Dr Plante for conducting a thorough review of our paper and for raising important points that have helped improve the scientific quality of the manuscript.

Before moving to a point-by-point reply format, here are our answers and comments to the major concern raised.

Based on the major concern expressed by Dr Plante, and also by Reviewer #1, on our initialization/spinup strategy, we have decided to use a new and simpler spinup strategy. As such, our simulations have been performed again, initialized with the new strategy, and

consequently, all diagnostics and figures have been processed and generated again, based on these new simulations.

But before providing the details on the new strategy, let us first rapidly justify the rationale behind our initial choice of initialization/spinup.

In our old ice-reinitialization spinup strategy, we used $A$, $h$ and $d$ taken from a neXtSIM simulation to reinitialize these fields, into SI3, a couple of months before the end of the spinup. We should have better emphasized that the sole purpose of this "mid-way sea-ice re-initialization" was solely to prevent the production segment of our two simulations to be initialized with an initial sea-ice state that has become too unrealistic in terms of sea-ice thickness and extent (due to the unavoidable accumulation of errors / drift during the first phase of the spinup), while conserving a "somewhat" spun-up 3D ocean state.

We thought that using $A$, $h$ and $d$ from the well-documented simulation of Boutin et al. 2023, was a good choice because their simulation was run on the same numerical grid and has been shown to be quite realistic when compared to observations.

However, we now understand that the fact that it is a BBM-driven simulation can be understood as unfair towards the aEVP simulations. However, we are completely convinced that it is not. This is because the main source of heterogeneity in the BBM-driven simulations comes from heterogeneity in the damage parameter, which will be spun up in a matter of days (Bouillon and Rampal, 2015; Rampal et al., 2016). Some ice thickness heterogeneity is inherited from the old spinup strategy though, but in our experience this is much less important than the damage evolution. This is borne out by the results of our revised initialisation strategy (using fields of the GLORYS2v4 reanalysis, performed with LIM using EVP) , which yields essentially the same results as the previous one.

For us, the main weaknesses of the old approach is that the surface properties of the ocean, such as temperature and salinity, are not updated the same way as the sea-ice properties are, creating a (very) temporary inconsistency between the ice-cover and the surface properties of the ocean.

But again, the main purpose of our old approach was to start the production simulation, December 15th 1996, with a sea-ice thickness the most consistent possible with respect to observations, and with a "lightly" spun-up 3D ocean state.

Conceptually, we face the following dilemma for the spinup strategy:
- A: Perform a very short spin-up and conserve a "rather realistic" sea-ice cover and thickness but potentially leave the 3D ocean in an adjustment phase
- B: Perform a long spinup, and risk initiating winter 1996-1997 with a sea-ice cover and thickness that has evolved quite differently from their observed counterparts, due to the imperfections of the model

Based on the main focus of our study, and as briefly discussed in the new version of the manuscript, we think that option A is completely acceptable, and this is what we have chosen for our new "simpler" initialization/spinup strategy.

As detailed in section 3.2 of the new version of the manuscript, we only perform a two-month long spinup, itself initialized October $1^{st}$ 1996 with $A$ and $h$, and the 3D state of the ocean,

taken from the GLORYS2V4 ocean reanalysis (NEMO + LIM using EVP, run on the same horizontal grid as our two Pan-Arctic experiments with data assimilation).

Actions performed :
- New spinup approach chosen
- The 2 simulation re-run following new spin-up approach
- Section about simulation description updated accordingly
- Diagnostics, figures, tables, updated using new simulation outputs
- Our results are not affected at all by this, result section almost not affected

Also based on concerns raised by the reviewers, on what has been wrongly interpreted as unfairness towards the aEVP rheology, we now make it very clear, all over the paper, that what we use as the reference SI3 simulation is the default workhorse setup of SI3 as provided in the current version of NEMO. As such, for instance, experiment "SI3-aEVP" has been renamed to "SI3-default", and it is clearly stated in the paper that a better-tuned "aEVP" would likely perform better (L513-516).

Furthermore, following the suggestion of reviewer #2, we have included a new section (3.1) that presents results of idealized SI3 simulations run on the "cyclone" test-case of Mehlmann et al. 2021, using both our BBM implementation and the default aEVP setup of SI3 (new figure 5).

Another important information regarding the new simulations:
In the first version of the manuscript, our developments and simulations were based on the version 4.2.1 of NEMO. At the time, it was the current stable release of NEMO. But during the course of the review process of this paper, a bug related to the ocean-ice drag parameterization has been identified and fixed by the NEMO team. This bug has been judged sufficiently severe by the NEMO team to justify the release of a new stable version: the 4.2.2. Link to release note: https://forge.nemo-ocean.eu/nemo/nemo/-/releases/4.2.2
Consequently we have switched to version 4.2.2 of NEMO for all the simulations presented in the new version of the manuscript.
The bug fix has significantly impacted the values of the simulated deformations and has therefore required a new tuning of the air-ice drag coefficients.

Major points:
- In the methods, it is indicated that the sea ice model is initialized with fields from neXtSIM simulations. This choice rise doubts on whether this BBM implementation is usable on its own, or if it is only realistic if initialized from neXtSIM fields. Also, and most importantly, this method brings a significant heterogeneity advantage for the BBM in the results, as it is additionally inheriting the damage from neXtSIM. It could very well be that the differences discussed in the result section is only a reflection of this difference. This is an important validation weakness and needs to be addressed.

Please see our reply to this major point above.

- It is difficult to assess the performance of the rheology implementation without any confirmation that the sea ice drift, thickness and concentration are realistic. Also, it is mentioned in the methods (L340-342) that the simulations are tuned to yield similar mean sea ice deformation values. While this method may make sense for the presented metrics, I expect that this tuning comes with significant (and possibly unrealistic) differences in sea ice drift, transport and thickness. Yet, this is not investigated.

Indeed, we do not include a thorough evaluation of the drift or thickness at large scale in the paper, first because it would be a full and dedicated study on its own to do it properly, and second it is not the scope of the present paper.

However, we can tell you that from the numerous simulations we performed during this tuning process, we have not noticed any significant qualitative or quantitative changes on the above-mentioned fields, that would let us revise our conclusions about the overall good performance obtained on the deformation metrics. In this regard, let us remind you that just as all our source code files and scripts, the output (netCDF) files of our simulations are made freely available to the community, see the Code and data availability, allowing anyone to control by themselve the aspects of the fields you mention.

Furthermore, the difference in drag coefficient means that there are larger internal forces in the BBM simulation than in the aEVP simulation, which likely largely impacts the PDF of sea ice deformations. This tuning method is thus also likely advantaging the BBM simulation, and may not be well suited for this validation. It should thus be revised / discussed more In-depth.

Based on your comment, and those of the reviewer #1, we do not "compare" anymore the skills of the BBM and the aEVP simulations in our paper. We realized that attempting such a comparison brings too much confusion on (and distraction from) the main message, which is: "a brittle rheology has been successfully implemented in a staggered-grid Eulerian framework, namely the SI3 model of NEMO, through a new approach based on a E-grid discretization. It allows realistic simulation of sea ice deformation across scales and may offer a new option to the large community of SI3 users to use a brittle rheology for sea ice and explore its potential impacts in coupled simulations."

Moreover, as you will see in the new version of the manuscript, and based on the suggestion of Reviewer #2, we have included the results of the "cyclone" idealized test-case of Mehlmann et al. 2021 for both rheologies. In this idealized experiment, all the relevant parameters are set to the values defined by the authors for this test-case, as such both SI3-defaut (the new name for SI3-aEVP) and SI3-BBM use the exact same $C_b^{(a)}$ of 1.2 10$^{-3}$, the same is true for other parameters used by both rheologies. Please see figure 5 in the new version of the manuscript.

- Results from the aEVP simulation seem particularly bad, even for an EVP model at 10km resolution. Is this using the standard aEVP model parameters in SI3? Please provide more information in that regard. While I understand that tuning the aEVP parameters might be out of the scope of this paper, this will nonetheless likely be picked by the community as

being an unfair comparison, especially that we know some model parameters that are better at representing LKFs (i.e., decreasing the ellipse aspect ratio). This is especially noteworthy given that the BBM parameters are themselves tuned for a better performance (e.g. in section 2.3.3). A similar effort should be done for the aEVP parameters, or at least the authors should indicate and justify the choice of not doing so in the analysis.

The paper has undergone some serious rewriting in this regard, also based on comments from Reviewer #1.

Yet, it (still) looks "bad" as you say, but we can assure you that we use the exact default and recommended setup of SI3, which is the result of years of collaboration between the NEMO development team and the Sea-ice Working Group of NEMO. As such, we cannot do better than moderating some of our conclusions, which is what we have done.

We now make it very clear, all over the paper, that what we use as the reference SI3 simulation is the default workhorse setup of SI3 as provided in the current version of NEMO (4.2.2), as such, for instance, experiment "SI3-aEVP" has been renamed to "SI3-default", and it is clearly stated in the paper that a better-tuned "aEVP" would likely perform better (L513-516).

- It is often implied, as motivation for this implementation, that it is not possible to implement the brittle models with the staggered components in the C-grid, citing Plante at al., (2020) as an example. This is inaccurate and misleading: Plante et al., (2020) in truth demonstrated how to implement the MEB with staggered components without yielding numerical problems. The numerical inaccuracies assessed in Plante et al., (2020) were not related to the staggered stress components, but to the mathematical expression of the prognostic damage equation under large convergence (i.e., their paper demonstrated the same instabilities mentioned at L131 in the current manuscript, with a follow-up, Plante and Tremblay, 2021, offering a solution). The proposed method here should thus not be presented as a solution to a numerical instability problem, but as a novel approach to avoid the treatment of staggered stress components.

We agree, all this was awkwardly introduced in the first version of the manuscript. And in this regard as well, the manuscript has undergone some substantial rewriting.

There is now more information about the solution that you have introduced to prevent instabilities in your 2020 paper in the introduction (L59). We also make it clear, in different parts of the manuscript, that the motivation for the new E-grid-based approach we present is driven by: (i) the requirement to fully advect the stress tensor along with the damage tracer (something important for a model that is used primarily to perform coupled ocean/sea-ice simulations in realistic global or regional configurations); and (ii) the fact that we want to avoid to use a smoothed damage scalar at the corner point of the grid (which requires a damage variable to be both defined at the corner point and also "advectable").

- While the text is mostly clear, some sentences are a bit wordy and tedious. This is throughout the text but most particularly in the introduction and section 3.

Specific comments (some repeating the comments above):

L8-12: This sentence is tedious; I am not sure what you are trying to say. Do you mean results show that the BBM in SI3 is suitable for climate simulations? If so, I think that this is far-fetch, only based on the localisation of sea ice deformations. I don't think that this is appropriate for this manuscript.

The abstract has been largely rewritten based on your comments and those of RW1.

Introduction: This section needs some work: it is a bit "wordy" and it feels more drafty then the rest of the manuscript.

L15: remove "all"

Removed (L14).

L42: "as an upgrade of MEB": this is a non-physical way to put it. The two rheology represent different physics, and we cannot call the BBM as an "upgrade" of the MEB. Please rephrase more objectively.

This sentence has been completely rewritten. (L41)

→ "Recently, Ólason et al. (2022) introduced the Brittle Bingham Maxwell rheology (hereafter BBM) as an effort to address the incomplete treatment of the convergence of highly damaged sea-ice in MEB."

L45: "preserving […] the thickness pattern of the sea-ice cover consistent with observations", preserving here does not work. Rephrase.

This sentence has been rewritten and uses the verb "reproduce" in place of "preserve". (L44)

L46-47: If it is proven, add the associated references. Otherwise, rephrase.

References to Ólason et al., 2022 and Boutin et al., 2023 have been added. (L46)

L64-65: This should be re-worded, as the McGill (and MITgcm) model is capable of producing pan-Arctic simulations (i.e., the models are not limited to the experiments presented in specific papers). The idealised experiments in Plante et al., (2020) were simply not designed to compare against sea ice drift and deformation observations. What you mean here is rather that the other implementations have not yet been assessed in a pan-Arctic context against sea ice drift and deformation observations? This might not be the right focus for this manuscript.

We have completely rewritten this part, it now reads:

"As of today, a few efforts have been made to implement MEB in sea-ice models comparable to SI3 in terms of discretization method and grid, such as the MIT general circulation model (Losch et al., 2010), or LIM, the former sea-ice component of the NEMO modeling system (Rousset et al., 2015). And more recently, Plante et al. (2020) have successfully implemented MEB in the McGill sea-ice model (Tremblay and Mysak, 1997; Lemieux et al., 2008, 2014). Overall, the work of these modeling groups have highlighted some challenging aspects that are specific to the implementation of a brittle rheology in a realistic Eulerian model that uses the finite-difference method on a staggered grid. As suggested by the work of Plante et al. (2020), when discretized on the Arakawa C-grid (Arakawa and Lamb, 1977), the same grid as used by SI3 (Vancoppenolle et al., 2023), brittle rheologies seem to be more prone to

numerical instabilities than their viscous-plastic counterparts. In particular, they report that the stability of their MEB implementation is sensitive to the resort to spatial averaging, an interpolation technique that is traditionally used to relocate certain fields between the staggered points of the grid." (L57)

L69-75: Here it is implied that the problem has no solution yet, although it was overcome in Plante et al., (2020). You should rephrase to reflect that here you propose a different approach, one that avoids interpolation.

Based on your comments and those of RW1 this paragraph has been completely rewritten, and no such statements are made.

L70: "is not well-suited for brittle rheologies": This is too vague and feels subjective. Do you mean that "it makes for a difficult implementation of brittle rheologies, in which any error in the prognostic stress variables are also integrated in the damage parameter"?

Same here, this statement is gone.

L74: Perhaps: we propose a novel approach to address this difficulty ?

We would rather use "new" instead of "novel", as our approach is not really ground-breaking, therefore our new sentence reads:

"In this paper, we propose a new discretization approach adapted to the numerical implementation of a brittle rheology in an Eulerian finite-difference-, C-grid-based sea-ice model." (L69)

Eq. 7: I see that the exponential SIC term is inside the parenthesis, and thus also has the dependency on alpha? This is different from other brittle implementations, and the rationale behind this change should be provided. Note also that this parameter is indicated as beta in Eq. B6, please rectify.

Note: it is now Eq. 6.

Yes, this form is specific to BBM, see Eq. 10 in Ólason et al., 2022. The rationale behind this is to give more realistic ice drift at low concentration.

Thank you for spotting this typo, it has been corrected \beta → \alpha. (L746)

Eq. 13: This formulation (Plante and Tremblay, 2021) miss-matches the comment at L121-122, as the stress state is only going back directly to the yield curve if Δt = Td .

You are right, we were wrong, the new sentence now reads:

"As discussed in Dansereau et al. (2016); Plante and Tremblay (2021), $d_{crit}$ is used to scale overcritical stresses back towards the *Mohr-Coulomb* damage criterion, …" (L123)

L139-141 : "that of a viscous-plastic one such as VP" that of a viscous-plastic (VP) rheology?

Yes, better this way, corrected. (L139)

L143: This is not exact: the ice is always viscoelastic in the MEB and BBM, not only when

damaged. Only, the viscous dissipation acts on a (much) longer time-scale in the undamaged case.

You are right for MEB, but not for BBM. Under compression, BBM is purely elastic until a certain normal stress threshold, $P_{max}$, is reached. And beyond, *i.e.* $\sigma_I < -P_{max}$, then viscosity is activated. Our original sentence, which has been written by Dr Dansereau herself, remains valid. Even if MEB, as opposed to BBM, always includes a viscosity, it is not wrong to state that "it considers unfragmented sea-ice as an elastic and damageable solid", as the viscosity tends towards infinity for no damage. Besides, it is stated, at the beginning of the sentence, that these are "elasto-visco-brittle" rheologies (L143).

L144: Watch for VP vs. EVP (here and through the text). Here I believe that you mean the EVP.

We meant "VP" in a broad sense, referring to viscous-plastic rheologies in general, including EVP, which despite its name is a viscous-plastic rheology, hence the use of the "s" at the end of "frameworks" in "As opposed to the VP frameworks". To prevent further confusion we now use "viscous-plastic" in place of VP when we imply "viscous-plastic".

L150-151: I am not sure that this is a real distinction between MEB and VP models. i.e., regardless of the mechanism to represent "discontinuities" (damage in the brittle models, rate-invariance in the VP), it remains that a continuum model does not resolve discontinuities, which are thus only represented by Finite Difference gradients.

We are not sure to understand the meaning of the reviewer's comment. We thus apologize in advance if this is the case.

We agree that none of the MEB and VP models actually "resolve" discontinuities. Both try to "simulate" them, in their own ways. And these ways are simply different in the two rheologies, because they are built on different physical hypotheses/constitutive (mechanical) models. So we think that this is a distinction between MEB and VP models that is worth mentioning . And we by no means claim that VP models are not  simulating "near-discontinuities", only that they do not originate from the same mechanism.

L158: Rephrase: a rheology is not bound to a numerical scheme. For instance, the MEB in the McGill model also uses an implicit iterative approach. The EVP uses explicit formulation.

We agree, we have rephrased the phrase as follows:

"Finally, note that in their numerical implementation of BBM, Ólason et al. (2022) chose to solve the dynamics explicitly using a time-step sufficiently small to account for the propagation of damage in the ice in a physically realistic manner." (L169)

L161-163: It could be noted that this small time step is also necessary to resolve the elastic waves, otherwise the elastic component (and the damage) is no-longer physically meaningful.

It is mentioned in the new version of the text in the new part that details the choice of the small time step. (below new Eq. 15, L176)

L166: In the case of the stress component, from a scale analysis, the advection is likely very negligible: the time-derivative term is the elastic term, acting at very short time scales, at which the advection is indeed very insignificant, especially at a 10km grid scale.

Yes it is, but the damage is a scalar that can live on for days, if not weeks depending on the temperature conditions. As such, it has to be advected like any other tracer of the model, and based on the strong interdependence between the damage and the internal stresses (through the elasticity, and viscosity of damaged ice) it is more consistent to also advect the stress tensor so that no spatial inconsistencies occurs between the damage and the resulting stress when considering simulations longer than say a couple of days.

We have added some extra discussion about this point in section 2.2 (L159-L168).

L170: "tracers are defined at the point located at the center of each cell"  tracers are defined at the cell centers.

Yes, better indeed, thank you. Corrected (L186).

L183-188 (suggestion): it could also be noted that in the VP formulation, the stress components are only diagnostics, and this staggering of the strain rate components only affects the computing of the Delta parameter, which itself is not prognostic (not integrated). However, in the brittle models, the staggering affects a larger number of prognostic variables (d, sigma), each of which are integrated and inter-dependent. This makes the implementation unforgiving for any incoherence in the treatment of the staggered components.

We have slightly improved this part, using some bits of your suggestion, but it remains mostly similar to our initial version.

L194-195: "in addition to being conceptually debatable in the context of a brittle model, which is expected to simulate very sharp spatial gradients", again, I find this a bit misleading, as this points to a caveat of using a continuum model, not the choice of rheology. You never truly resolve discontinuity in FD continuum approach. This sentence thus suggests that the brittle models should not be used in a continuum context to begin with.

This part has been completely rewritten, and this statement is no longer present.

L197-198: This is false and inacceptable: Plante et al., (2020) actually demonstrated how to implement the MEB with the staggered components without having these errors, while it is written that Plante et al. (2020) implemented the MEB with has all these errors and inaccurate solutions.

We never intended to convey this message, the idea was simply to mention that, similarly to what we experienced, you also experienced checkerboard instabilities as a consequence of the spatial averaging, but BEFORE implementing your method to overcome them. And after having read countless time the section 2.3.2 of your 2020 paper this is still what we understand when you write:

*"Averaging the shear stress components from the neighboring nodes (as in Eq. 37 for the scalars) causes a checkerboard instability in the solution because of the staggered shear stress corrections and memories."*

Our wording was clearly awkward and confusing, apologies for this. So we have completely rewritten this part, in a more accurate way, providing more details on how you overcome the checkerboard instabilities. Hence, the new version of this whole paragraph is:

"On the C-grid, a common way to interpolate a scalar defined at F-points onto T-points is to simply use the average of this scalar on the four surrounding F- points, and conversely to interpolate from T- to F-points. In the aEVP implementation of SI3 (Kimmritz et al., 2016), the problem posed by the staggering of tensor elements is overcome by using this averaging approach to interpolate the square of the shear rate $\dot{\epsilon}_{12}$ from F- to T-points . Later on, the term $P/\Delta$ is also interpolated from T- to F-points in order to estimate $\sigma_{12}$ . In their implementation of MEB, Plante et al. (2020) also use this approach to interpolate the damage tracer at F-points. However, they report that using the same approach to estimate $\sigma_{12}$ , and hence $\sigma_{II}$ , a T-points when performing the Mohr-Coulomb test (Eq.13), results in checkerboard instabilities. The solution they propose to prevent the occurrence of these instabilities is to introduce an additional $\sigma_{12}$ that is defined at T-points. This additional $\sigma_{12}$ is updated at each time step using – as an increment – the average of the four $\sigma_{12}$ increments computed at the surrounding F-points." (L201-210)

L199-202: This is quite vague, subjective yet strongly worded. What do you mean exactly by "consistent"? Do you mean that one needs interpolation but not the other? How much is it an issue?

We have rephrased this sentence, making it less strongly-worded and more accurate ("consistent" is in terms of advection scheme used for the advection).
The new sentence reads:
"Finally, with the C-grid, the implementation of the advection of $\sigma 12$ (F-point) in a way consistent (in terms of the advection scheme used) with that used for $\sigma 11$ and $\sigma 22$ (T-point) is somewhat challenging. That is because the advection of a scalar defined at the F-point, using the same scheme as that used for the advection of scalars at T-points, requires the existence of a u and a v at V- and U-points, respectively." (L118-222)
We have also added the discussion about why we think that despite involving very small terms it is important to advect the stress tensor along with the damage-scalar in section 2.2. See our point above (about L166) on this matter.

L221-222: For consistency, wouldn't it be better to also have A and h defined also at the f-points? So that all tracers defining the E and lambda parameters are treated the same way?

Absolutely! We have considered this option since day 1 of the E-grid idea, but this would require to port the thermodynamics module of SI3 onto the F-points of the grid as well. This is something that may sound straightforward to achieve considering that the thermodynamics is mostly based on column/1D processes, but we can assure you that it is way more tricky to implement given the complexity of the thermodynamics module of SI3…
Nevertheless, this is definitely something that we plan to implement  in the near future, together with the NEMO/SI3 developers, if our implementation is deemed promising by the NEMO team of course…

Eq. 16: I am not sure if this is an error (?): I would expect the stresses to be corrected (and not defined) by the right hand side, which represents a (weighted) difference between the two offset grids. Am I missing something? More precisely, I would expect an equation on the form of sigma_new = sigma_old + weight * Delta-Sigma, but here we have sigma_new = weight* Delta-Sigma. Please clarify.

You are right, our cross-nudging equation was full of typos in the first version of the manuscript. This has been corrected. And more generally, all parts dealing with the cross-nudging have been significantly re-worked and enhanced, based on some of your comments and those of RW1. See new section 2.3.2.

L271: Each time step, to me, seem to refer to the smaller time step, but the thermodynamics is the longer time step. Please clarify. It would also be useful to define the small and large time steps as the "dynamical" and "advective" time steps.

In this sentence, we are referring to the advective time step, we have added "advective". (L316)

We have followed your suggestion, and now "small/big time step" has been replaced by "dynamical/advective time step" throughout the manuscript.

L307: remove "so-called" and "which is".

This sentence no longer exists in the current version of the manuscript.

L333-334: Can you clarify here that this initialization from OCE-neXtSIM is made for both the BBM and the aEVP simulations? Also, please give more insight on why the sea ice model needs to be initialized from neXtSIM. Are results similar if the model is used without this neXtSIM dependency? I would argue that the analysis would be most meaningful and convincing if the simulations were ran without this neXtSIM initialization.

Based on your major concern, the strategy has been completely changed (see our reply at the very beginning of this document). Hence the text is also changed and we hope that it is clear in this new version.

We performed a single 2-month long spinup (October 1st to November 30th 1996) using the SI3-default setup initialized with A, h & the 3D ocean state taken from the GLORYS2v4 reanalysis. Then, both SI3-default and SI3-BBM were started December 1st 1996 with the same restart generated at the end of the spinup. (see new section 3.2)

L333: "and damage (only for SI3-BBM)": this is a very important difference in the method between the simulations, yet this effect is not at all discussed or investigated. It means you initialize the BBM with heterogeneity imported from neXtSIM, but less so for the aEVP case.

Same here, everything has been changed, see our previous and introductory reply.

Personally we tend to believe that it might be the opposite: that the heterogeneities imported from neXtSIM in our old spinup strategy, mainly in terms of ice thickness, could have actually helped to trigger the formation of LKFs in the aEVP run. The BBM simulation produces new LKFs in a matter of a few hours, regardless of the aspect/state of the initial sea-ice cover, *e.g.* including if concentration and thickness are set to constant over the entire Arctic basin.

L342: This tuning of the sea ice deformation could be interesting, as opposed to tuning the sea ice drift, but we need to understand how else it impact the simulations. How is it impacting the sea ice drift and thicknesses? Are they very different between the simulations? Are they all realistic, or is this tuning made at the cost of the large-scale dynamics? This should be shown and discussed.

Please see our previous and upcoming replies on this drag coefficient matter. The focus of the paper is on the numerical implementation of BBM into SI3 with an assessment of simulated sea-ice deformations against those obtained from satellite data. The Journal is called "Geoscientific Model Development" and we believe that, as such, the work presented in this paper is very relevant and substantial . Plus we are no longer comparing the BBM to the aEVP simulation anymore, because apparently, too many remarks from the reviewers seem to imply that we have purposely made aEVP look bad.

L354-355: How does your method compares with the SIREx methods?

We believe our method is very similar to that used for SIREx (Bouchat et al. 2022). However, the scripts used by Bouchat et al. to compute deformations and scaling are still not available, and so despite having asked the lead author of that paper to share them with us a very long time ago. We thus cannot claim full equivalence between our method and theirs, which is why we refer to the method presented in the paper of Ólason et al. 2022 from which three of us are the authors. Besides, a lot of details regarding the approach and methods we use is given in the manuscript, now in section 2.5, and all our scripts are available online so that anyone can use/check/improve our method.

L368: This to me looks like a particularly bad result even for the aEVP at 10km: we indeed do not expect the EVP to produce much LKF at this resolution, but this result is particularly smooth... Is this using the default aEVP parameters in SI3, or can it be a result of the drag coefficient tuning? This should be discussed, and add some notes here to show awareness that the aEVP results could be improved by simple modifications to the model parameters, and justify the choice of not doing so for the comparison.

See our previous reply to the similar point made in the major concerns section.

What's new in this current version of the paper:
- We stress that aEVP could be better tuned
- We use "SI3-default" in place of the old "SI3-aEVP"
- All aEVP-related parameters are shown in new Table 1
- Total deformation simulated by aEVP at different resolutions is assessed via the Mehlmann cyclone test-case, which produces results very similar to those of Mehlmann et al 2021, which tend to suggest that, after all, the aEVP of SI3 is not so badly tuned.
- Same test-case results are also shown in Annex C with aEVP using 1000 iterations instead of 100 in the rest of the paper.
- Based on your remarks and those of RW1 we have also performed a full Pan-Arctic simulation with aEVP using 1000 iterations instead of 100 (figures included at the end of this document).

- And we have also performed the same aEVP simulation with an air(-ice) drag coefficient of 2 10-3 in place of 1.4 10-3 as used in the default setup (see figure 3 at the end of this document).

Section 3.4.2: This feels a bit strange as you demonstrate just above that the aEVP simulation does not have a tail. So, it is unclear what is the purpose of this analysis, as it only repeats what is already demonstrated in section 3.4.1.

This section has been heavily rewritten, and we barely mention the results for the aEVP driven simulation. Note: it's now section 3.3.2.

But we would be criticized for not including SI3-default in this diagnostic.

Again our main interest is to see how SI3-BBM behaves with respect to RGPS, SI3-default only serves the purpose of showing what to expect with the workhorse setup of NEMO.

L398: "a transition" : Please clarify what transition this refers to.

This refers to the transition when the moment $q$ of a random variable's distribution diverges/is not finite. Here by "divergence" we mean that the law of large number does not hold, or in other words, that we cannot find a $\mu$ such that $\sum_{i} X_i/n \to \mu$

Usually, this happens for $q > \beta+1$, with $\beta$ being the exponent of the power-law tail of the pdf of that random variable. In order to satisfy the curiosity of an eventual reader we've added the reference to a classic book on statistics theory from the 50's, in which this point is discussed (Savage, 1954).

L403: "Interestingly", do you mean "In particular"? It sounds as if this result is not expected.

We have changed to "In particular" (L543).

L408: These cda values should be provided in the method section, at L342. Also, this is a surprisingly wide difference between the simulations, and thus suggest that the internal stresses would be much smaller in the aEVP simulations. This surely impacts the production of large deformation rates and LKFs. This needs to be investigated as to show how much this impacts the difference in representing the tail of the sea ice deformation PDFs.

Note: these values of $C_D^{(a)}$ have changed with the NEMO bug fix mentioned in the intro, but this does not change the fact that BBM requires a larger $C_D^{(a)}$ to be in agreement with RGPS at the 10 km scale.

We have added these values in the method section just as you advise (L490-495).

In order to address your point, we have performed a clone of simulation SI3-default using $C_D^{(a)} = 2 \ 10^{-3}$, the same value as used in SI3-BBM, instead of $1.4 \ 10^{-3}$. As shown in Figure 3 at the end of this document, if we increase $C_D^{(a)}$ as such in the SI3-default simulation, apart from obtaining amplified deformations everywhere (which will yield a PDF of the total deformation way above that of RGPS, not shown), almost no additional LKFs are created (we just see one about north of the Komsomolets Island).

Also, note that in the new figure 5 (manuscript) showing the solution in cyclone test-case,

both SI3-BBM and SI3-default use the same $C_D^{(a)} = 1.2\ 10^{-3}$ as recommended by Melhmann et al. 2021.

**L417-426: What is the purpose of this analysis? It is unclear yet what the structure functions tell us, in terms of model performance (Bouchat et al., 2022).**

We think the structure function is still an insightful diagnostic to evaluate to which degree a sea ice model is capable of localizing e.g. the simulated sea ice deformation, and more specifically how the degree of localization of deformation rates depends on their actual intensity (i.e. small deformation events being less localized than the larger ones). The higher the curvature of the structure function, the higher the degree of multifractality (the smaller the fractal dimension of the deformations distribution over space), hence more localized are the largest deformation events compared to the smaller ones. This approach is very much used to characterize e.g. the patterns/distribution of atmospheric precipitations that look typically very heterogeneously distributed over space.

Yet, the question whether this metric can be used (solely) to discriminate between models may be subject for debate. And if this is what the reviewer meant with his question, then we would tend to agree.

**L429-431 Again, this is a strong comment that is not demonstrated. It also does not reflect that there have been successful implementations of the MEB on the staggered C-grid.**

This sentence is no longer present.

**L472-474: A note could be added here that the fully-converged EVP likely influences the production of LKFs.**

Well it turns out that, at least with the SI3 implementation of aEVP, it does not. As mentioned in a previous reply, we have performed a clone of simulation SI3-default with $N_{EPV}$=500 instead of the default $N_{EPV}$=100 (Figure 2, at the end of this document) and you will see that it barely changes anything. Same with the new figure X included in the new Appendix C that compares the cyclone test-case solution with $N_{EPV}$=1000 instead of the default $N_{EPV}$=100. Again, in the new version of the manuscript we stress that aEVP could be better tuned (L513-516), but we were unable to get different results by changing the number of iterations.

**L483-485: This investigation is interesting and should be shown.**

We think our paper is already very dense. We have spent a lot of energy adding some extra material like the cyclone test-case, we have performed new sensitivity experiments with SI3-default to back our honesty. We are sorry but will not carry out this investigation in this paper.

**L487-489: This has not been assessed here, there are no comparisons with neXtSIM.**

We refer to the exact same figure as in Ólason et al. 2022, so we think it is fine to mention it this way.

**Conclusion: This section includes several strongly worded but subjective statements that do not serve well the actual contribution of the manuscript. In the context of a scientific**

manuscript, such subjective statements have a history of distracting readers from the actual analysis and to raise doubt on the transparency. This should be revised to be more nuanced and to keep the focus on the presented analysis.

About our transparency: we would like to bring to your attention the fact that contrary to a lot of recent papers published in this field of sea-ice rheology, we are openly sharing all our source codes, scripts, and the simulated data we have produced…

Having said that, we agree on the rest of your comment and you will see that our conclusion has undergone a serious rewriting as well and is more moderate than this old version.

L499-501: This is not accurate and too strongly worded. As the authors acknowledge in the introduction, the MEB has been implemented in the McGill sea ice model, and the MITgcm, which are both capable of producing pan-Arctic simulations.

We have removed this statement and rephrased as follows instead:

"We have shown that our implementation, which features a prognostic ice damage tracer and a prognostic stress tensor, is able to realistically simulate sea-ice deformation statistics on a pan-Arctic scale when compared to satellite observations." (L633-635)

We indeed probably expressed our point inaccurately, and we apologize. The important part of our statement was "... a realistic solution on a pan-Arctic scale…". Our message was certainly not to contradict what we ourselves wrote in the introduction about an existing implementation of MEB into the McGill model, and even less to pretend that this model is not capable of producing pan-Arctic simulations! It would actually be very interesting to see such simulations and their evaluation with respect to sea ice drift, thickness-volume and deformation in a publication...

As for the MITgcm though, we are indeed aware of the recent work of the reviewer together with colleagues from AWI/Germany to implement the MEB in this model, but in this case (i) there is no published publication presenting this implementation as far as we know, and (ii) there is no paper presenting pan-Arctic simulations from this model with this rheology switched on either (some of us have seen a single poster with preliminary results, but again of idealized simulations only). And so no comparison to observations at the Pan-Arctic scale that we had a chance to see.

L501-502: "has proven to be poorly fitted for brittle rheologies." Again, it is not demonstrated. This is a rather subjective statements not reflecting that the MEB was successfully implemented on staggered C-grids in other models.

This was clearly an awkward statement that has now disappeared from the manuscript.

L520-523: This last paragraph needs to be removed. It has not even been assessed here: there are no comparison with neXtSIM, only against aEVP. It is particularly blind to the listed numerical sensitivities associated with this implementation, which likely strongly affect the results (e.g., just looking at Figure 5).

PAragraph removed.

L571: If I understand well, you get the stress state as a function of the (d, u, v, h, A) from the

previous iteration (i.e. from the constitutive equation), but then apply the cross-nudging before computing the new damage. One issue I see with this is that the propagation of damage is supposed stem from the stress concentration around the previous damage. By applying the cross-nudging between these steps, I fear that the cross-nudging would affects the development of damage and the orientation of its propagation. As this method is a main contribution of this paper, I believe it would be worth providing some analysis and discussion on this regard. Also, to me, the cross-nudging is not very different, in terms of numerical effect, than using interpolation: you are smoothing between two solutions instead of between staggered components. Not that it is an issue to me, but as the text highlights that interpolating is not appropriate, it then raises the question: how is this cross-nudging more appropriate than interpolating?

It is a good point, yet we are not sure to fully understand your reasoning.

We favor the option to apply the cross-nudging (CN) right before the Mohr-Coulomb (MC) test for the exact same reason we think it is important to advect the stress tensor: the strong interdependence, and hence spatial correlation, between the stresses and the damage.

If we apply the CN right after the MC test, then we may propagate to neighbor points (via the averaging process at play in the CN) stress values that have been corrected without propagating the associated increase in damage (we don't want to have to apply a CN on the damage!). And we want the damage and the stresses to remain fully spatially consistent.

But as suggested by sensitivity experiments we have performed, comparing the two options, in the end, the contribution of the cross-nudging correction at a given sub-time-step is so weak that it does not really matter if it is done before or after the MC test. It has been impossible to notice significant differences between simulated fields obtained using one or the other options.

You are right, this point is worth mentioning in the paper, so we have added some discussion about this aspect in the text (L296-301).

Then, to reply to your second point, we now clearly state in the conclusion (L646-644), that the use of the cross-nudging, required by the use of the E-grid, introduces some smoothing of the solution, and is therefore in contradiction with one of the reason/motivation for choosing the E-grid in the first place: namely to avoid averaging some fields. However, we also recall the reader, that despite the resort to this cross-nudging, the use of the E-grid allows to have a native prognostic damage tracer at the corner points that is never smoothed (beneficial for the estimate of $\sigma_{12}$), and also allows to advect $\sigma_{12}$ using the same advection scheme a that used to advect $\sigma_{11}$ and $\sigma_{22}$.

References:

Bouchat A, Hutter NC, Dupont JCF, Dukhovskoy DS, Garric G, Lee YJ, Lemieux JF, Lique C, Losch M, Maslowski W, Myers PG, Ólason E, Rampal P, Rasmussen T, Talandier C, Tremblay B and Wang Q (2022) Sea Ice Rheology Experiment (SIREx), Part I: Scaling and statistical properties of sea-ice deformation fields. J. Geophys. Res., 127(4), e2021JC01766 (doi: 10.1029/2021JC017667)

Hutter NC, Bouchat A, Dupont F, Dukhovskoy DS, Koldunov NV, Lee YJ, Lemieux JF, Lique C, Losch M, Maslowski W, Myers PG, Ólason E, Rampal P, Rasmussen T, Talandier C,

Tremblay B and Wang Q (2022) Sea Ice Rheology Experiment (SIREx), Part II: Evaluating simulated linear kinematic features in high-resolution sea-ice simulations. J. Geophys. Res., 127(4), e2021JC017666 (doi: 10.1029/2021JC017666)

Plante M, Tremblay LB, Losch M and Lemieux JF (2020) Landfast sea ice material properties derived from ice bridge simulations using the Maxwell elasto-brittle rheology. The Cryosphere, 14(6), 2137–2157

Plante M and Tremblay LB (2021): A generalized stress correction scheme for the Maxwell elasto-brittle rheology: impact on the fracture angles and deformations. The Cryosphere, 15 (12)

[Figure]

*Fig.1/ Figure 7.c of manuscript, total deformation for experiment SI3-default, ($N_{EVP}$=100, and $C_D^{(a)} = 1.4 \ 10^{-3}$).*

[Figure]

*Fig.2/ Same as Figure 7.c of manuscript, but for SI3-default using $N_{EVP}$=500 instead of $N_{EVP}$=100.*

[Figure]

*Fig.3/ Same as Figure 7.c of manuscript, but for SI3-default using $C_D^{(a)}$=2 $10^{-3}$ instead of $C_D^{(a)}$=1.4 $10^{-3}$.*

---

## Referee Report (RR1)

**2[nd] Review of: "*Implementation of a brittle sea-ice rheology in an Eulerian, finite-difference, C-grid modeling framework: Impact on the simulated deformation of sea-ice in the Arctic*" by Laurent Brodeau, Pierre Rampal, Einar Ólason, and Véronique Dansereau.**

This manuscript presents the implementation of the Brittle Bringham Maxwell (BBM) rheology in the SI3 community sea ice model, which uses a C-Grid Finite Difference framework. In particular, the authors present their use of the E-Grid discretization scheme to avoid difficulties with the staggered velocity and stress components in the C-Grid. The performance of their BBM implementation is then assessed first using a benchmark simulation from Mehlmann et al., 2021, then in pan-Arctic context, both compared to the same SI3 simulations repeated using the standard SI3 rheology configuration (the aEVP rheology). Results demonstrate the adequacy of the BBM implementation and its good performance in representing LKFs.

I find that the manuscript is interesting, important and has potential for publication in GMD. In particular, it provides detailed discussions on difficulties implementing the BBM rheology in the SI3 sea ice model, and presents the numerical tools used by the authors to overcome them. This is an important step for the rheology to be more thoroughly investigated and used by the community.

I commend the work made by the authors to clarify the manuscript and genuinely address all of my (and other reviewers) comments. Many useful explanations have been added to the implementation description, and some changes to the methods largely clarify the analysis and allow for a more detailed presentation and interpretation of the results. I find this version of the manuscript to be very clear, useful and interesting, making for a strong manuscript that will open the door for a wider use of the BBM rheology in the sea ice modelling community.

I therefore recommend its publication, after addressing the following minor edits/clarifications.

Best regards,

Mathieu Plante

Minor comments:

- Title: perhaps indicating E-augmented C-grid?

- L287: "for the term of the divergence of the stress tensor" --> either : for the stress divergence term, or : for the rheology term?

- L288-293: Please clarify, I have difficulty understanding what you try to say. Are you referring to the 1/h form of the last terms in Eq. 16? Also, I am not sure what errors you are referring to.

- L294-295: Please clarify. Also, I may be missing something but according to the equation, the nudging intensity is modulated by gamma_ns/Ns …  So why would it be less sensitive to Ns than to gamma_cn?

- L296: remove coma after (Eq.9). Also, I would introduce (CN) earlier, at L280, and use it thereafter.

- L296 "before any potential upcoming correction is applied following the Mohr-Coulomb test" :: Rewrite, as this sounds as if the nudging is made between the MC test and the correction. I believe, according to your answers to my last review, that you rather apply the nudging before applying the MC test (i.e. before calculating dcrit)?
  Perhaps something along the lines of: "Due to the strong damage-stress interdependence, we apply the CN after solving the constitutive equation but before computing dcrit and applying the stress correction".

- L297-298: "then we may propagate […] ":: Too many comas, this sentence needs to be clarified. I get that you mean that it would introduce a discrepancy between the damage and the corrected stress values. This is very helpful, thanks for adding this explanation to the manuscript.

- L303: remove come after gamma_cn

- L323-324: This precision may not be needed, and confuses me as "the sum of Ns successive displacement vectors" sounds too much like making a Lagrangian track out of the Eulerian vectors. But I do not think it is the case here: I assume that the mean is applied on the Eulerian U,V, hence defined at a specific grid location and not building into a track.

- L325: I am not sure, but I would remove the dash between order and moment.

- L342: "yields the so-called" -> change to "is the"

- Add point at end of L395.

- Section 2.5: I find a bit awkward the miss-match between the subsections (2.5.1--2.5.4) and the use of "as a first step", "as a second step", "a final step". I think it would be clearer if you distinguished between the trajectory selection (L388-395, which could be 2.5.1), from the quadrangle selection (L403-409), instead of describing both as steps in an owerall "selection process".

- L411-412: repetition of quadrangles: → "based on their position #1 and #2 using […]"

- L413-414: "Similarly to […]." → This sentence is confusing, and may not be needed, as the following sentence is clear enough by itself.

- L420: no need for indent (keep in 1 paragraph)

- L442: "somewhat very different" → not clear. I suggest going directly into the differences for conciseness.

- L498: remove shown and put "figure 7" in parentheses.

- Section 3.3.1. Looking at the curves, it looks to me that in terms of power law, the standard simulations have a similar slope in the tail of the PDFs, although with under-represented large deformations. This is worth mentioning, although I understand that it is not the focus. E.g., perhaps something on the lines of "Our results suggest a propensity for SI3-default to exhibit somewhat similar power-law behaviour (similar exponents, except in convergence) but yet to underestimate the extreme values of the deformation rates. […]").?

---

## Author Response (AR2)

**2nd Review of: "*Implementation of a brittle sea-ice rheology in an Eulerian, finite-difference, C-grid modeling framework: Impact on the simulated deformation of sea-ice in the Arctic*" by Laurent Brodeau, Pierre Rampal, Einar Ólason, and Véronique Dansereau.**

This manuscript presents the implementation of the Brittle Bringham Maxwell (BBM) rheology in the SI3 community sea ice model, which uses a C-Grid Finite Difference framework. In particular, the authors present their use of the E-Grid discretization scheme to avoid difficulties with the staggered velocity and stress components in the C-Grid. The performance of their BBM implementation is then assessed first using a benchmark simulation from Mehlmann et al., 2021, then in pan-Arctic context, both compared to the same SI3 simulations repeated using the standard SI3 rheology configuration (the aEVP rheology). Results demonstrate the adequacy of the BBM implementation and its good performance in representing LKFs.

I find that the manuscript is interesting, important and has potential for publication in GMD. In particular, it provides detailed discussions on difficulties implementing the BBM rheology in the SI3 sea ice model, and presents the numerical tools used by the authors to overcome them. This is an important step for the rheology to be more thoroughly investigated and used by the community.

I commend the work made by the authors to clarify the manuscript and genuinely address all of my (and other reviewers) comments. Many useful explanations have been added to the implementation description, and some changes to the methods largely clarify the analysis and allow for a more detailed presentation and interpretation of the results. I find this version of the manuscript to be very clear, useful and interesting, making for a strong manuscript that will open the door for a wider use of the BBM rheology in the sea ice modelling community.

I therefore recommend its publication, after addressing the following minor edits/clarifications.

Best regards,

Mathieu Plante

We appreciate Dr Plante's efforts in conducting a second review of our manuscript and for providing constructive suggestions, as well as identifying areas for improvement. Please find below our point-by-point responses to your comments.

Best regards, Laurent Brodeau

Minor comments:

- Title: perhaps indicating E-augmented C-grid?
We find that the title is already long and a bit "painful" to read, so that we would rather keep the original title. Yet, our main argument for preferring "C-grid" to "E-augmented C-grid" in the title has to do with the message we would like the title to convey. We want to make it clear, to potential readers, that the implementation we present in our paper is mainly relevant for sea-ice models that use the C-grid.

It is also clearly stated, already from the abstract (L5) or the introduction (L70), that our implementation involves augmenting the C-grid into an E-grid.

- L287: "for the term of the divergence of the stress tensor" --> either : for the stress divergence term, or : for the rheology term?
Based on the next comment of the reviewer, this sentence has been removed as it was only introducing confusion.

- L288-293: Please clarify, I have difficulty understanding what you try to say. Are you referring to the 1/h form of the last terms in Eq. 16? Also, I am not sure what errors you are referring to.
We agree with the reviewer that this part was confusing. Based on the fact that it does not bring any useful information on the cross nudging approach, we have decided to remove it. The useful information, which is that the cross-nudging treatment is performed on the vertically-integrated stresses is already stated (L277), and we have completed this statement by explicitly adding the expressions of the vertically-integrated tensors in the text (L278).

L294-295: Please clarify. Also, I may be missing something but according to the equation, the nudging intensity is modulated by gamma_ns/Ns ... So why would it be less sensitive to Ns than to gamma_cn?
As stated earlier in the text (L283), during the course of one advective time step $\Delta T$, the cross-nudging is applied every other dynamical time steps, so $N_s/2$ times. Let us imagine 2 configurations: in the first one the user chooses to use $N_s=100$, and in the second $N_s=200$. The user wants to achieve the same level of cross-nudging in the two experiments, so uses the same $\gamma_{CN}$ in both experiments. In the first case, each stress tensor is going to be corrected 50 times during 1 $\Delta T$ with a nudging intensity of $\gamma_{CN}/100$, in the second case it is going to be corrected 100 times with a nudging intensity of $\gamma_{CN}/200$. This, in the end, will achieve a somewhat similar level of correction during the course of 1 $\Delta T$, hence our statement:
"*The form of the term that modulates the nudging intensity, i.e. $\gamma_{cn}/Ns$ , ensures that the level of cross-nudging undergone by the two tensors under one $\Delta T$ is primarily controlled by $\gamma_{cn}$ and remains somewhat independent of the choice of $N_s$.*"
We don't think that we can write our original sentence in a clearer way, so it is left unchanged. (L283-L285)

L296: remove coma after (Eq.9). Also, I would introduce (CN) earlier, at L280, and use it thereafter.
We agree, coma removed (L287) and "CN" introduced earlier (L277).

L296 "before any potential upcoming correction is applied following the Mohr-Coulomb test" ::
Rewrite, as this sounds as if the nudging is made between the MC test and the correction. I believe, according to your answers to my last review, that you rather apply the nudging before applying the MC test (i.e. before calculating dcrit)?
Perhaps something along the lines of: "Due to the strong damage-stress interdependence, we

apply the CN after solving the constitutive equation but before computing dcrit and applying the stress correction".

We agree that this sentence was confusing and we thank the reviewer for his suggestion. The new sentence now reads:

"Due to the strong damage-stress interdependence (section 2.2), the CN is applied on $\boldsymbol{\sigma}^{(i)}$ rather than $\boldsymbol{\sigma}$ , *i.e.* after solving the constitutive equation (Eq. 9) but before computing $d_{crit}$ and applying the stress correction (Eq. 12). " (L286-L287)

L297-298: "then we may propagate [...] ":: Too many comas, this sentence needs to be clarified. I get that you mean that it would introduce a discrepancy between the damage and the corrected stress values. This is very helpful, thanks for adding this explanation to the manuscript.

This part was clearly painful to follow. Here are the two new sentences we have written instead (L287-L290):

"Applying the CN after the stress correction stage (rather than before) may result in the use of a mix of (i) stress values that have been corrected through a local increase in damage and (ii) uncorrected stress values (with no increase in damage). This may lead to spatial inconsistencies between the post-CN stresses and the damage field."

L303: remove come after gamma_cn

Done (L292).

L323-324: This precision may not be needed, and confuses me as "the sum of Ns successive displacement vectors" sounds too much like making a Lagrangian track out of the Eulerian vectors. But I do not think it is the case here: I assume that the mean is applied on the Eulerian U,V, hence defined at a specific grid location and not building into a track.

Yes. We agree that this sentence was confusing and completely unnecessary, we have removed it (L312).

L325: I am not sure, but I would remove the dash between order and moment.

The reviewer is right, following a small investigation this advection scheme is generally referred to as "second-order moments", we have corrected accordingly in the manuscript (L313).

L342: "yields the so-called" -> change to "is the"

Done (L330).

Add point at end of L395.

Done (L383).

Section 2.5: I find a bit awkward the miss-match between the subsections (2.5.1--2.5.4) and the use of "as a first step", "as a second step", "a final step". I think it would be clearer if you distinguished between the trajectory selection (L388-395, which could be 2.5.1), from the quadrangle selection (L403-409), instead of describing both as steps in an owerall "selection process".

We agree. We have removed all the "as a first step", "as a second step", "a final step", and instead we have followed the recommendation of the reviewer to include a new "subsubsection" (now 2.5.1) (L375).

L411-412: repetition of quadrangles: → "based on their position #1 and #2 using [...]"
Corrected accordingly (L399).

L413-414: "Similarly to [...]." → This sentence is confusing, and may not be needed, as the following sentence is clear enough by itself.
We agree, the first sentence has therefore been removed, and what the reviewer refers to as the "following" sentence has been changed from:
"Instead, we assign the time that corresponds to the center of the time interval defined by position #1 and position #2 of each quadrangle."
to:
"The time location (date) assigned to a given deformation rate corresponds to the center of the time interval defined by position
**1 and position #2 of each quadrangle." (L401-L402)**

L420: no need for indent (keep in 1 paragraph)
Done. (L406)

L442: "somewhat very different" → not clear. I suggest going directly into the differences for conciseness.
Yes, this sentence was not particularly useful, we removed it and get directly to the differences as suggested by the reviewer:
"In the solutions obtained with BBM, we note the presence of a circular network of LKFs..."  (L429)

L498: remove shown and put "figure 7" in parentheses.
Yes, done (L486).

Section 3.3.1. Looking at the curves, it looks to me that in terms of power law, the standard simulations have a similar slope in the tail of the PDFs, although with under-represented large deformations. This is worth mentioning, although I understand that it is not the focus. e.g., perhaps something on the lines of "Our results suggest a propensity for SI3-default to exhibit somewhat similar power-law behaviour (similar exponents, except in convergence) but yet to underestimate the extreme values of the deformation rates. [...]").?
Yes, it is indeed worth mentioning. We have added the following sentence, largely inspired from your suggestion, at the end of the paragraph:
"Interestingly, the underestimation of extreme deformation values set aside, SI3-default exhibits a power-law behavior similar to that of both observations and SI3-BBM, with similar exponents, except in convergence." (L502)